# Convergence Theory of Decentralized Diffusion Models via Pseudo-Non-Markov Analysis

## Abstract

Diffusion probabilistic models (DPMs) have demonstrated remarkable success in generative tasks, supported by a solid foundation of convergence analysis. Recently, decentralized DPMs have been proposed to enhance data security and enable cross-institutional collaboration. However, their unique decentralized structure renders existing analysis techniques inapplicable, leaving their theoretical convergence properties an open question. In this paper, we introduce a novel pseudo-non-Markovian method to analyze the convergence of both standard and decentralized DPMs within the context of the denoising diffusion probabilistic model (DDPM) sampler. Our key technical insight is to reframe the analysis of the backward transition. While the transition from $x_t$ to $x_s$ ($s < t$) is Markovian, we analyze its conditional form given the initial data $x_0$. This conditional transition becomes non-Markovian but gains a tractable analytical expression, allowing for a direct analysis of the discretization error on the Cartesian product space of $x_t \times x_s \times x_0$. We show that this method is readily extensible to the decentralized setting. To the best of our knowledge, our convergence theory represents the first of its kind applicable to the decentralized scenario.

## 1 Introduction

The diffusion probabilistic models (DPMs) (Sohl-Dickstein et al., 2015; Ho et al., 2020; Song & Ermon, 2019; Song et al., 2020b), generating samples of data distribution from initial noise by learning a reverse diffusion process, have been proven to be an effective technique for modeling data distribution, especially in images generation (Nichol et al., 2022b; Dhariwal & Nichol, 2021; Saharia et al., 2022; Ramesh et al., 2022; Rombach et al., 2022; Ho et al., 2022a), image super-resolution (Li et al., 2022), audio generation (Nichol et al., 2022a; Kong et al., 2021; Popov et al., 2021), video generation (Ho et al., 2022b; Yang et al., 2024), 3D generation (Poole et al., 2022), and motion planning (Carvalho et al., 2023).

Diffusion models generate data by relying on a pair of forward and backward diffusion stochastic differential equations (SDEs) (Song et al., 2020b). The forward diffusion SDE corrupts data into noise, while the backward SDE reverses this process. A (Stein) score is required in the backward diffusion SDE; thus, the training process of diffusion models can be regarded as using a neural network to match the score $\nabla_{\boldsymbol{x}} \log q_t(\boldsymbol{x})$ at different noise levels. Given the complexity of this

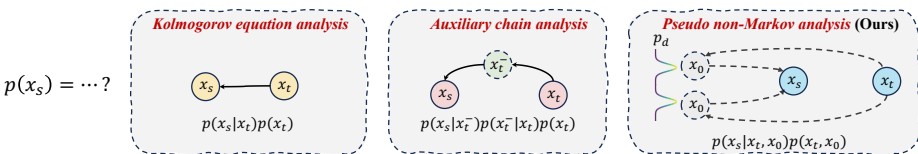

Figure 1: Comparison of analysis methods. **(a)** Kolmogorov equation analysis relies on single-chain SDE steps, which are absent in decentralized DPMs. **(b)** Auxiliary chain analysis is difficult to generalize from its typical ratio-based auxiliary states to the decentralized setting. **(c)** In contrast, our pseudo-non-Markovian approach derives an analytical form with respect to the initial data, leveraging Gaussian tail properties for analysis on the Cartesian product space of $x_t \times x_s \times x_0$. Note that in the proofs, we use $y$ to represent the initial data points for simplicity.

Table 1: Comparison of different convergence analysis methods for the standard DDPM samplers of DPMs and decentralized DPMs. In some works, a better convergence rate is reported for high-order samplers; here, we only show the rates on the standard DDPM sampler. Convergence rates are shown using the total variation (TV) distance, with logarithmic factors neglected. Here, $d$ denotes the data dimension; $|\mathcal{D}|$ denotes the maximum time interval, whose order is consistent with the $\frac{1}{T}$ notation in (Li & Yan, 2024) and the $h$ notation in (Lee et al., 2022); $\bar{y}$ denotes the true value of the $y$-prediction for DPMs; and $y_\theta$ denotes the $y$-prediction value learned by the network.

| Analysis methods | Convergence rate on DPMs | Assumptions on $p_d$ | Assumptions on $y_\theta$ | Convergence rate on decentralized DPMs |
|---|---|---|---|---|
| Kolmogorov Equation (Lee et al., 2022) | $\mathcal{O}(d^{\frac{13}{18}} L^{\frac{4}{9}} C_{LS}^{\frac{5}{9}} |\mathcal{D}|^{\frac{1}{2}})$ | $\nabla \log p_d$ is L-Lipschitz; $p_d$ is $C_{LS}$ log-Sobolev; $\int \|x\|_2^2 \mathrm{d}p_d(x) < \infty$ | $y_\theta$ is L-Lipschitz; $y_\theta \approx \bar{y}$ in $L^2(p(x_t))$ | Not extensible ☹ |
| Kolmogorov Equation (Chen et al., 2022) | $\mathcal{O}(L d^{\frac{1}{2}} |\mathcal{D}|^{\frac{1}{2}})$ | $\nabla \log p_d$ is L-Lipschitz; $\int \|x\|_2^2 \mathrm{d}p_d(x) < \infty$ | $y_\theta \approx \bar{y}$ in $L^2(p(x_t))$ | Not extensible ☹ |
| Kolmogorov Equation (Lee et al., 2023) | $\mathcal{O}(d^{\frac{37}{46}} L^{\frac{10}{23}} |\mathcal{D}|^{\frac{1}{2}})$ | bounded support; $\nabla \log p_d$ is L-Smooth | $y_\theta \approx \bar{y}$ in $L^2(p(x_t))$ | Not extensible ☹ |
| Kolmogorov Equation (Chen et al., 2023a) | $\mathcal{O}(d |\mathcal{D}|^{\frac{1}{2}})$ | $\int \|x\|_2^2 \mathrm{d}p_d(x) < \infty$ | $y_\theta \approx \bar{y}$ in $L^2(p(x_t))$ | Not extensible ☹ |
| Kolmogorov Equation (Benton et al., 2024) | $\mathcal{O}(d^{\frac{1}{2}} |\mathcal{D}|^{\frac{1}{2}})$ | $\int \|x\|_2^2 \mathrm{d}p_d(x) < \infty$ | $y_\theta \approx \bar{y}$ in $L^2(p(x_t))$ | Not extensible ☹ |
| Auxiliary Chain (Li & Yan, 2025) | $\mathcal{O}(d |\mathcal{D}|)$ | $\int \|x\|_2^2 \mathrm{d}p_d(x) < \infty$ | $y_\theta \approx \bar{y}$ in $L^2(p(x_t))$ | Not extensible ☹ |
| Auxiliary Chain (Li & Jiao, 2025) | $\mathcal{O}(d^{\frac{1}{2}} L^{\frac{1}{2}} |\mathcal{D}|)$ | $\nabla \log p_d$ is L-Lipschitz; $\int \|x\|_2^2 \mathrm{d}p_d(x) < \infty$ | $y_\theta$ is L-Lipschitz; $y_\theta \approx \bar{y}$ in $L^2(p(x_t))$ | Not extensible ☹ |
| Pseudo Non-Markov **(This paper)** | $\mathcal{O}(d |\mathcal{D}|)$ | $\int \|x\|_2^2 \mathrm{d}p_d(x) < \infty$ | $y_\theta$ is bounded; $y_\theta \approx \bar{y}$ in $L^2(p(x_t))$ | $\mathcal{O}(d |\mathcal{D}|)$ ☺ |

process, proving convergence to the desired data distribution is not an easy task, especially when additional modifications are made to the diffusion process.

Significant research advancements have been made to validate the convergence of diffusion models. Lee et al. (2022) were the first to provide polynomial convergence guarantees for diffusion models. The scope of this convergence was further expanded to a broader range of data distributions (Chen et al., 2022; Lee et al., 2023). De Bortoli (2022) demonstrated convergence when the data is only supported on a lower-dimensional manifold. Chen et al. (2024); Benton et al. (2023) focused on the development of convergence for deterministic sampling. Additionally, Li et al. (2023) was able to achieve a superior error bound by making additional assumptions on the Jacobian of the score functions. However, the aforementioned convergence relies heavily on tools from SDEs, such as the Kolmogorov equations (Lee et al., 2022) and the Girsanov theorem (Chen et al., 2022). This reliance makes it challenging to adapt these methods to cases where the reverse process is not derived from a SDE. More recent works (Li & Yan, 2025; Li & Jiao, 2024) also explore the convergence rate of DPMs by constructing intricately designed auxiliary ratio-based intermediate chains.

Recently, decentralized DPMs (McAllister et al., 2025; Chen et al., 2025; Dong et al., 2024) have been proposed through the introduction of a modified multi-cluster diffusion process. This innovation enables data privacy across various institutions while maintaining generative performance. However, for the decentralized DPMs, previous convergence theories that built on Kolmogorov equations or auxiliary chains cannot be applied, see details in Appendix E.2. To our best knowledge, there is currently no theory to guarantee the convergence of this practically promising framework.

In this paper, we propose an intuitive pseudo-non-Markovian convergence theory for DPMs. Our key insight is to analyze the reverse process transition from a state $x_t$ to $x_s$ by first conditioning on the initial data $x_0$, which yields a tractable, non-Markovian conditional transition with an analytical form. We then analyze the local discretization errors by partitioning the Cartesian product space of the states $(x_t, x_s, x_0)$. Our theory also incorporates errors from network approximation and the forward process. Notably, while conceptually simple, our framework establishes a convergence rate comparable to state-of-the-art results. More importantly, the pseudo-non-Markovian method can be seamlessly extended to decentralized DPMs, providing the first convergence guarantee for this empirically valuable framework.

**Our contributions.** Compared to previous efforts in DPMs convergence theory, our main contributions are as follows:

- In Theorem 3.7, we propose a novel pseudo-non-Markov method to establish a convergence rate for DPMs in terms of the total variation (TV) distance. Under mild finite-moment assumptions on initial data, we establish a $\mathcal{O}(d|\mathcal{D}|)$ TV convergence rate for DPMs, which is comparable to the SOTA. This approach is intuitive, which obviates the need for resorting to the complex Kolmogorov equation theory and intricate Markov chain construction.

- In Theorem 5.2, we extend our pseudo-non-Markov convergence theory to the decentralized DPM, where the transition kernel is redefined as a mixture of Gaussians. This is achieved by the inherent extensible nature of our pseudo-non-Markov method. We point out that the decentralized DPMs would introduce an additional classifier approximation error, but retain the same forward error, score network approximation error, and discretization error as the classical DPMs.

- We observe that the classifier approximation error in decentralized DPMs diverges as $T$ increases, which we attribute to the zero-order training of the cluster classifier. Theoretically, we propose mitigating this divergence by incorporating extra high-order objectives for the decentralization classifier to ensure convergence.

**Organization.** Section 2 reviews the preliminaries of standard and decentralized DPMs. We then present our pseudo-non-Markovian convergence theory for standard DPMs in Section 3, followed by a proof sketch in Section 4. Finally, Section 5 extends this analysis to establish the first convergence guarantee for decentralized DPMs.

## 2 PRELIMINARIES

**Notation**. The Euclidean norm over $\mathbb{R}^d$ is denoted by $\|\cdot\|$, and the Euclidean inner product is denoted by $\langle\cdot|\cdot\rangle$. Throughout, we simply write $\int g$ to denote the integral with respect to the Lebesgue measure: $\int g(x)\mathrm{d}x$. When the integral is with respect to a different measure $\mu$, we explicitly write $\int g\mathrm{d}\mu$. When clear from context, we sometimes abuse notation by identifying a measure $\mu$ with its Lebesgue density. We use $\overline{A}$ to denote the closure and $A^c$ to denote the complement of a set $A$. We denote a standard $d$-dimensional Gaussian distribution as $\gamma^d$. Detailed notations table can be found in Appendix A.1

### 2.1 DIFFUSION PROBABILISTIC MODELS AND TRANSITION KERNELS

Given a data distribution $p_d(x)$, a diffusion model is designed to generate samples according to the data distribution. Diffusion probabilistic models define two Markov chains, including the forward process and reverse process. The forward process is typically hand-designed with a Gaussian transition kernel to perturb data to noise and can be expressed as (Zhang et al., 2024):

$$p(x_t, t|x_s, s) = \mathcal{N}(x_t; \alpha_{t|s}x_s, \sigma_{t|s}^2 I), \tag{1}$$

where $\alpha_t$ and $\sigma_t$ are positive scalar functions of $t$ satisfying $\alpha_t^2 + \sigma_t^2 = 1$, and $\alpha_t$ decreases monotonically from 1 to 0 over time $t$ (Kingma et al., 2021). The choice for $\alpha_t$ and $\sigma_t$ is referred to as the noise schedule of a DM. $t, s$ are two timesteps and $0 \leq s < t \leq 1$, and $\sigma_{t|s} = \sqrt{1 - \alpha_{t|s}^2}$, $\alpha_{t|s} = \alpha_t/\alpha_s$, $\sigma_{t|s}^2 = 1 - \alpha_{t|s}^2$.

Hence, the conditional distribution of $x_t$ given $x_0$ can be derived as:

$$p(x_t, t|x_0, 0) = \mathcal{N}(x_t; \alpha_t x_0, \sigma_t^2 I). \tag{2}$$

Taking the initial condition $p(x_0, 0) = p_d(x_0)$ into account, the single time marginal distribution of $x_t$ is:

$$p(x_t, t) = \int_y (2\pi\sigma_t^2)^{-\frac{d}{2}} \exp\left(-\frac{\|x_t - \alpha_t y\|^2}{2\sigma_t^2}\right) p_d(y)\, \mathrm{d}y. \tag{3}$$

The reverse process reverses the forward one with a learned kernel. Based on Eq. (1) and (3), the ground truth reversed transition kernel can be derived with Bayes' rule:

$$
\begin{aligned}
p(x_s, s|x_t, t) &= p(x_t, t|x_s, s)\frac{p(x_s, s)}{p(x_t, t)} \\
&= (2\pi\sigma_{s|t})^{-\frac{d}{2}} \int_y w(x_t, t, y) \exp\left(-\frac{1}{2\sigma_{s|t}^2}\left\|x_s - \frac{\alpha_{t|s}\sigma_s^2}{\sigma_t^2}x_t - \frac{\alpha_s\sigma_{t|s}^2}{\sigma_t^2}y\right\|^2\right)\, \mathrm{d}y,
\end{aligned}
\tag{4}
$$

where $\sigma_{s|t} = \sigma_{t|s}\frac{\sigma_s}{\sigma_t}$, $w(x_t, t, y) = \frac{u(x_t,t,y)}{\int_{y'} u(x_t,t,y')\,\mathrm{d}y'}$ while $u(x_t, t, y) = \exp\left(-\frac{||x_t - \alpha_t y||^2}{2\sigma_t^2}\right)p_d(y)$. Here $w(x_t, t, y)$ is exactly the conditional distribution of $y$ given $x_t, t$.

Existing methods typically approximate the learnable reversed transition kernel as a single Gaussian distribution (Ho et al., 2020). The transition kernel can be expressed as (Zhang et al., 2024):

$$\tilde{p}(x_s, s|x_t, t) = (2\pi\sigma_{s|t})^{-\frac{d}{2}} \exp\left(-\frac{1}{2\sigma_{s|t}^2}\left\|x_s - \frac{\alpha_{t|s}\sigma_s^2}{\sigma_t^2}x_t - \frac{\alpha_s\sigma_{t|s}^2}{\sigma_t^2}\bar{y}(x_t, t)\right\|^2\right), \quad (5)$$

where $\bar{y}(x_t, t) = \int_y w(x_t, t, y)y\,\mathrm{d}y$. Subsequently, the approximated single-time marginal distribution with the accurate $\bar{y}(x_t, t)$ is:

$$\tilde{p}(x_{t_i}) = \int_{\mathbb{R}^d} \cdots \int_{\mathbb{R}^d} \tilde{p}(x_{t_i}|x_{t_{i+1}}) \cdots \tilde{p}(x_{t_{T-1}}|x_{t_T})\tilde{p}(x_{t_T})\,\mathrm{d}x_{t_{i+1}} \cdots \mathrm{d}x_{t_T}. \quad (6)$$

The mean of this Gaussian distribution is related to $\bar{y}(x_t, t)$, which is estimated by a neural network $y_\theta(x_t, t)$ in $x$-prediction methods, while the variance is isotropic and depends only on the timestep $s$ and $t$. Another commonly used parameterization is $\epsilon$-prediction, which uses a noise prediction network to estimate noise $\epsilon(x_t, t)$ (Salimans & Ho, 2022). Despite their difference from $x$-prediction, these two parameterizations are equivalent, as demonstrated by the relationship.

$$x_t = \alpha_t \bar{y}(x_t, t) + \sigma_t \epsilon(x_t, t). \quad (7)$$

By substituting $\bar{y}(x_t, t)$ in Eq. (6) and Eq. (5) with the network prediction $y_\theta(x_t, t)$, we obtain $p_\theta(x_{t_i})$ and $p_\theta(x_{t_i}|x_{t_{i+1}})$. In this paper, we utilize the $x$-prediction for simplicity in our proofs. For clarity, we refer to $p(x_t)$ and $p(x_s \mid x_t)$ instead of $p(x_t, t)$ and $p(x_s, s \mid x_t, t)$ when there is no ambiguity.

## 2.2 PRELIMINARIES ON DECENTRALIZED DPMs

Based on differing motivations, McAllister et al. (2025), and concurrently, Chen et al. (2025); Dong et al. (2024), propose to partition data into clusters and utilize cluster-specific diffusion models to match the noised data distribution. The primary intuition driving McAllister et al. (2025) is two-fold: to ensure data security across various institutions and to distribute the computational burden across multiple nodes, particularly in large-scale training scenarios. In contrast, Chen et al. (2025); Dong et al. (2024) cluster the data with the distinct insight that this approach would mitigate the discretization error by formulating the reverse transition kernels as a Mixture-of-Gaussian (MoG) distribution, rather than relying on a simple naive single Gaussian parameterization.

In the decentralized scenario, the training data is divided into $L$ classes: $\{y_i \in \mathbb{R}^d | i = 1, 2, \ldots, N\}$ $= \bigcup_{l=1}^L \{y_i^l \in \mathbb{R}^d | i = 1, 2, \ldots, N_l\}$. For a data point $x_0$, we use a one-hot vector $L(x_0)$ to denote its cluster label. We also define the underlying cluster partition on the data distribution as $\{p_d^l\}_{l=1}^L$. Note that these $p_d^l$ are measures rather than probability measures; their sum constitutes a probability measure. The cluster division way can be arbitrary in theory, and in practice can be decided by institution (McAllister et al., 2025), k-means, or dataset class label (Chen et al., 2025; Dong et al., 2024). Later, we will show that any method of data division results in convergence, because the subsequent proof does not rely on how the data is partitioned.

The underlying true kernel is then approximated by integrating the cluster-specific kernels. The decentralized DPM (Dong et al., 2024) defines a new MoG approximation to reverse transition as:

$$\hat{p}(x_s|x_t) = \sum_{l=1}^L a^l(x_t, t)\hat{p}^l(x_s|x_t), \quad (8)$$

where $\hat{p}^l$ is the reverse transition kernel on label $l$, $a^l$ denotes the time-aware class probability:

$$\hat{p}^l(x_s|x_t) = \int_y (2\pi\sigma_{s|t})^{-\frac{d}{2}} u^l(x_t, t, y) \exp\{-\frac{1}{2\sigma_{s|t}^2}||x_s - \frac{\alpha_{t|s}\sigma_s^2}{\sigma_t^2}x_t - \frac{\alpha_s\sigma_{t|s}^2}{\sigma_t^2}\bar{y}^l(x_t, t)||^2\}p_d^l(y)\mathrm{d}y, \quad (9)$$

$$w(x_t, t, y) = \frac{\exp\left(-\frac{||x_t - \alpha_t y||^2}{2\sigma_t^2}\right)}{\sum_l \left[\int_{y'} \exp\left(-\frac{||x_t - \alpha_t y'||^2}{2\sigma_t^2}\right)p_d^l(y')\mathrm{d}y'\right]}, \quad a^l(x_t, t) = \int_y w(x_t, t, y)p_d^l(y)\mathrm{d}y \quad (10)$$

$$u^l(x_t, t, y) = \frac{w(x_t, t, y)}{a^l(x_t, t)}, \quad \bar{y}^l(x_t, t) = \int_y u^l(x_t, t, y) y p_d^l(y) \mathrm{d}y \tag{11}$$

The marginal distribution of this approximation $\hat{p}(x_t)$ is defined in the same way as in Eq. (6). In practice, we use L neural networks to approximate $\bar{y}^l(x_t, t)$, which are parameterized as conditional network $y_\theta(x_t, t, l)$ (also denoted as $y_\theta^l(x_t, t)$). Additionally, a neural network $a_\phi(x_t, t)$ is necessary to approximate $a(x_t, t) \stackrel{def}{=} (a^1(x_t, t), \cdots, a^L(x_t, t))^T \in \mathbb{R}^L$. $a_\phi(x_t, t)$ is trained via the objective $\mathbb{E}_{x_0 \sim p_{data}, t \sim \mathrm{U}(1,T), x_t \sim p(x_t|x_0)} \|a_\phi(x_t, t) - L(x_0)\|^2$.

# 3 CONVERGENCE THEORY OF DPMS

In this section, we will present our assumptions on the initial data, noise schedule, and network, and then introduce our convergence analysis result.

## 3.1 ASSUMPTIONS

To start with, we outline some assumptions regarding the initial distribution, the neural network approximation errors, and the noise scheduler, which will be referenced throughout this paper:

**Assumption 3.1.** *The data distribution $p_d(x)$ has finite second moment:*

$$\mathbb{E}_{x \sim p_d} \left[\|x\|^2\right] = \int \|x\|^2 p_d(x) dx = M < \infty. \tag{12}$$

**Remark 3.2.** *The Assumption 3.1 is fulfilled in common image, audio, and video datasets, which guarantees that any marginal distribution has a finite moment.*

By the Fubini-Tonelli theorem, the Assumption 3.1 allows for the interchange of the order of integration, enabling a shift between double integrals and iterated integrals across our paper. For the sake of simplicity, we will not explicitly mention this in the remaining part of the paper.

**Assumption 3.3.** *For all $t$, $y_\theta(\cdot, t)$ and $\bar{y}(\cdot, t)$ are close in $L^2(p)$, that is, $\int_{\mathbb{R}^d} p(x_t) \|y_\theta(x_t, t) - \bar{y}(x_t, t)\|^2 \mathrm{d}x_t < \varepsilon_y^2 < 1$. And $y_\theta(\cdot, t)$ are universally bounded by a constant $C_{y_\theta}$.*

**Remark 3.4.** *This assumption has been adopted by previous studies (Chen et al., 2022; Lee et al., 2023) and is confirmed by Oko et al. (2023). This is consistent with the training objectives of DPMs.*

**Assumption 3.5.** *$\alpha_t$ is a predefined function that decreases monotonically from 1 to 0, with its derivatives bounded; specifically, $0 > \frac{\mathrm{d}\alpha_t}{\mathrm{d}t} \geq -C_\alpha$ for some positive constant $C_\alpha$.*

**Remark 3.6.** *As $\alpha_t$ is designed manually, we assume that its derivative is bounded, which is satisfied by all commonly used DPM schedulers like VE (Song & Ermon, 2019), VP (Ho et al., 2020), sub-VP (Song et al., 2020b), and EDM (Karras et al., 2022).*

Our theory relies on the finite moment of the target distribution, $l_2$-accurate and finite score estimates, and a well-behaved noise scheduler. In contrast to prior works, it does not require the log-Sobolev (Lee et al., 2022), log-concavity (Gao & Zhu, 2024), or Lipschitz assumptions (Chen et al., 2022; 2023c;d), or finite Jacobian error assumption in (Li et al., 2023; 2024).

## 3.2 MAIN RESULTS

Considering that the sampling process occurs in discrete steps, we introduce the notation for time discretization as $\mathcal{D} = \{0 < t_{\min} = t_0 < t_1 < \cdots < t_T = t_{\max} < 1\}$. We also define $\Delta t_i = t_{i+1} - t_i$ and denote the maximum $\Delta t_i$ as $|\mathcal{D}|$. Based on these assumptions, we derive the following convergence result as the main theorem of our paper:

Figure 2: The illustration of our analysis scheme. (a) The discretization error is introduced because of the gap between the forward transition and the backward transition, which is analysed in Lemma 4.1 and Lemma 4.2. (b) The network approximation error is introduced by imperfect score matching, which is tackled in Lemma 4.3. (c) The forward error is introduced by the convergence gap of the forward process to the standard Gaussian, which is detailed in Appendix A.6. (d) The singularity interval error near $t = 0$ is investigated in Appendix A.7.

**Theorem 3.7.** *(Main Theorem)* *There exist $C_1, C_2, C_3 > 0, \delta > 0, k \geq 1$ depending on $t_{min}$, such that for all time discretizations $\mathcal{D}$ with $|\mathcal{D}| < \delta$, the following inequality holds:*

$$TV(p(x_{t_{min}}), p_\theta(x_{t_{min}})) \leq \underbrace{C_1 d|\mathcal{D}| \log^k \frac{1}{|\mathcal{D}|}}_{\text{Discretization Error}} + \underbrace{C_2 \varepsilon_y}_{\text{Network Approximation Error}} + \underbrace{C_3 \exp\left(-\frac{1}{|\mathcal{D}|}\right) \sqrt{KL(p_d, \gamma^d)}}_{\text{Forward Error}}$$

$$(13)$$

The proof overview can be seen in Section 4. In this theorem, we establish an $\mathcal{O}(d/T)$ convergence rate (log is omitted) for the DDPM sampler in TV distance under mild assumptions. This result is comparable to the most advanced convergence result for DDPM in (Li & Yan, 2025).

**Convergence on Singularity Endpoints**   Notice that the diffusion model has an inherent singularity property when $t_{\min} \to 0$ (Zhang et al., 2024). It's not feasible to compute the TV distance for this singularity endpoint. We adapt the idea from Theorem 2.1 in prior work (Lee et al., 2023), which applies the Wasserstein distance to the left interval.

For the detailed results on the singularity error and discussion of the singularity issue at the right endpoint, we defer them to Appendix A.7.

## 4   PROOF OF THE MAIN THEOREM VIA PSEUDO NON-MARKOV ANALYSIS

In this section, we introduce a novel approach to demonstrate that the distribution generated by the conventional diffusion model closely matches the actual data distribution, obviating the need for Kolmogorov equations or complex auxiliary sequences. Instead, we adopt the pseudo-non-Markov analysis to formulate the analytical form of the local discretization error directly.

From a high-level intuition, we decompose the total variation (TV) distance into two components: the discretization error between $p(x_s)$ and $\tilde{p}(x_s)$, and the approximation error between $\tilde{p}(x_s)$ and $p_\theta(x_s)$. The discretization error arises from the discrepancy in the forms of the forward and backward diffusion processes, which is inherent to the design of DPMs. The approximation error stems from the matching error of the learned score network. Note that there will be an additional forward error stemming from the convergence gap of the forward process to the standard Gaussian.

### 4.1   BOUNDING THE DISCRETIZATION ERROR VIA PSEUDO-NON-MARKOV ANALYSIS

For the discretization error part, we first quantify the local discretization error and sum it up.

$$2\text{TV}(p(x_s), \tilde{p}(x_s)) \leq \underbrace{\int_{\mathbb{R}^d} \left| \int_{\mathbb{R}^d} p(x_s|x_t)p(x_t) - \tilde{p}(x_s|x_t)p(x_t) \, dx_t \right| dx_s}_{\text{Local Discretization Error}} + 2\text{TV}(p(x_t), \tilde{p}(x_t)) \qquad (14)$$

The core challenge in the convergence theory consistently resides in bounding the discretization error in Eq. (14), which represents an integral of the difference between $p(x_s|x_t)p(x_t)$ and $\tilde{p}(x_s|x_t)p(x_t)$

over the Cartesian product space $(x_s \times x_t) \in (\mathbb{R}^d)^2$. Direct knowledge of these terms is unavailable. However, our pseudo non-Markov method conditions these transitions on a specific initial data point $y$, and we can derive their analytical conditional forms, as follows:

$$p(x_s|x_t)p(x_t) = \int p(x_s|x_t, y)p(x_t, y)\,\mathrm{d}y$$

$$= \int (2\pi\sigma_{s|t}^2)^{-\frac{d}{2}} \exp\left(-\frac{1}{2\sigma_{s|t}^2}\left|x_s - \frac{\alpha_{t|s}\sigma_s^2}{\sigma_t^2}x_t - \frac{\alpha_s\sigma_{t|s}^2}{\sigma_t^2}y\right|^2\right)(2\pi\sigma_t^2)^{-\frac{d}{2}}\exp\left(-\frac{1}{2\sigma_t^2}|x_t - \alpha_t y|^2\right)p_d(y)\,\mathrm{d}y \tag{15}$$

Similarly, for the single-Gaussian reverse kernel, its pseudo-non-Markov formulation writes:

$$\tilde{p}(x_s|x_t)p(x_t) = \int \tilde{p}(x_s|x_t, y)p(x_t, y)\,\mathrm{d}y$$

$$= \int (2\pi\sigma_{s|t}^2)^{-\frac{d}{2}} \exp\left(-\frac{1}{2\sigma_{s|t}^2}\left|x_s - \frac{\alpha_{t|s}\sigma_s^2}{\sigma_t^2}x_t - \frac{\alpha_s\sigma_{t|s}^2}{\sigma_t^2}\bar{y}(x_t, t)\right|^2\right)(2\pi\sigma_t^2)^{-\frac{d}{2}}\exp\left(-\frac{1}{2\sigma_t^2}|x_t - \alpha_t y|^2\right)p_d(y)\,\mathrm{d}y \tag{16}$$

Thus, the integral space is transformed to the Cartesian product space of $(x_s \times x_t \times y) \in (\mathbb{R}^d)^3$. Then, we divide the space into light-tail areas and the Gaussian peak area. Define a partition $\mathbb{P} = \{\mathbb{P}_1, \mathbb{P}_2, \mathbb{P}_3, \mathbb{P}_4\}$ on this Cartesian product space $(x_s \times x_t \times y)$ with

$$\begin{aligned}
\mathbb{P}_1 &= \{|u_{ts}| > \sigma_{t|s}B\} \cap \{|\epsilon_t| > \sigma_t B \text{ or } |\bar{\epsilon}_t| > \sigma_t B\}, \\
\mathbb{P}_2 &= \{|u_{ts}| > \sigma_{t|s}B\} \cap \{|\epsilon_t| \le \sigma_t B \text{ and } |\bar{\epsilon}_t| \le \sigma_t B\}, \\
\mathbb{P}_3 &= \{|u_{ts}| \le \sigma_{t|s}B\} \cap \{|\epsilon_t| > \sigma_t B \text{ or } |\bar{\epsilon}_t| > \sigma_t B\}, \\
\mathbb{P}_4 &= \{|u_{ts}| \le \sigma_{t|s}B\} \cap \{|\epsilon_t| \le \sigma_t B \text{ and } |\bar{\epsilon}_t| \le \sigma_t B\}.
\end{aligned} \tag{17}$$

where $B = \sqrt{-Cd\log\sqrt{\alpha_s}\sigma_{t|s}}$ and define $u_{ts}(x_t, x_s) = x_t - \alpha_{t|s}x_s$, $\epsilon_t(x_t, y) = x_t - \alpha_t y$, $\epsilon_s(x_s, y) = x_s - \alpha_s y$, $\bar{\epsilon}_t(x_t) = x_t - \alpha_t\bar{y}(x_t, t)$. Note that the union of these four regions forms the whole space. The motivation is as follows. In $\mathbb{P}_1, \mathbb{P}_2, \mathbb{P}_3$, the probability of the distribution itself on these regions will be small due to the light-tail property of the Gaussian kernel. Only in the region near the peak of the Gaussian, that is, $\mathbb{P}_4$, do we need to delve into the Taylor expansion form of the difference to establish the bound. In the following Lemmas, we establish the bounds of the discretization error on each partition.

**Lemma 4.1.** *For all adjacent timesteps $s < t$, there exist $\delta > 0$ and $C_1, C_2, C_3 > 0$, the integral of local discretization error on the $\mathbb{P}_1, \mathbb{P}_2, \mathbb{P}_3$, is bounded by $C_1(t-s)^2, C_2(t-s)^2, C_3(t-s)^2$.*

**Lemma 4.2.** *For all adjacent timesteps $s < t$, there exist $\delta > 0$ and $C > 0, k \ge 1$, the integral of local discretization error from $s$ to $t$ on the $\mathbb{P}_4$ is bounded by $Cd(t-s)^2\log^k\frac{1}{t-s}$.*

Considering these partitions jointly, we conclude that the local discretization error is of the order $\mathcal{O}(d(t-s)^2)$, which contributes to the $\mathcal{O}(d(t-s))$ term in the global error.

## 4.2 BOUNDING THE APPROXIMATION ERROR VIA PINSKER'S INEQUALITY

For the approximation error, we adopt a different approach. We generally follow the intuition of Li & Yan (2024), which get the global approximation error in terms of the KL divergence first, and use Pinsker's inequality to quantify the global approximation error in terms of the TV distance.

$$\mathrm{TV}(\tilde{p}(x_s), p_\theta(x_s)) \le \sqrt{\frac{1}{2}\mathrm{KL}(\tilde{p}(x_s), p_\theta(x_s))} \tag{18}$$

In a similar derivation to Eq. (14), we can transfer the global approximation error into an iterative summation of local approximation error.

$$\mathrm{KL}(\tilde{p}(x_s), p_\theta(x_s)) = \underbrace{\mathrm{KL}\left(\tilde{p}(x_s|x_t)\tilde{p}(x_t), p_\theta(x_s|x_t)\tilde{p}(x_t)\right)}_{\text{Local Approximation Error (KL)}} + \mathrm{KL}(\tilde{p}(x_t), p_\theta(x_t)) \tag{19}$$

In the following lemma, we show that the local approximation error, measured in terms of KL divergence, is of the order $\mathcal{O}((t-s)\varepsilon_y^2)$.

**Lemma 4.3.** *For all adjacent timesteps $s < t$, there exist $\delta > 0$ and $C > 0$, the inequality $KL\left(\tilde{p}(x_s|x_t)\tilde{p}(x_t), p_\theta(x_s|x_t)\tilde{p}(x_t)\right) \leq C(t-s)\varepsilon_y^2$ holds.*

Then the global approximation error, measured in terms of KL divergence, is $\mathcal{O}(\varepsilon_y^2)$. By extension, the global approximation error in terms of the total variation (TV) distance will be $\mathcal{O}(\varepsilon_y)$.

**Bounding the Forward Error**   The convergence error stems from the finite horizon of the forward process, which is the gap between $p(x_T)$ and $\tilde{p}(x_T) = \gamma^d$. We defer its derivation to Appendix A.6.

## 5   EXTENSION TO THE DECENTRALIZED DPMS

Similar to Assumption 3.1, we also need the assumption that $y_\theta^l(x_t, t)$ and $a_\phi(x_t, t)$ approximate $\bar{y}^l(x_t, t)$ and $a^l(x_t, t)$ in $L^2(p)$ to derive the error bound of the decentralized DPM.

**Assumption 5.1.** *For all $t \in [t_{min}, 1]$ and $1 \leq l \leq L$, $y_\theta^l$ and $a_\phi^l$ are close to $\bar{y}^l$ and $a^l$ in $L^2(p)$ respectively, that is, $\int_{\mathbb{R}^d} p(x_t)||y_\theta^l(x_t, t) - \bar{y}^l(x_t, t)||^2 \, dx_t < \varepsilon_{yl}^2 < 1$ and $\int_{\mathbb{R}^d} p(x_t)(a_\phi^l(x_t, t) - a^l(x_t, t))^2 \, dx_t < \varepsilon_{al}^2 < 1$. Let's denote $\varepsilon_y = \max_l \varepsilon_{yl}$ and $\epsilon_a = \max_l \varepsilon_{al}$. We also need the y-prediction network output to be upper-bounded.*

Then, $\hat{p}_{\theta,\phi}(x_s|x_t)$ is defined by $y_\theta^l$s and $a_\phi^l$s. $\hat{p}_\theta(x_t)$ is defined in a manner consistent with Eq. (6).

To estimate the error boundary of the decentralized DPMs, we employ a strategy that transforms it into a weighted single Gaussian. Taking into account that $p(x_s|x_t)$ has this equivalent form:

$$p(x_s|x_t) = \sum_{l=1}^{L} a^l(x_t, t)p^l(x_s|x_t), \tag{20}$$

where

$$p^l(x_s|x_t) = \int_y (2\pi\sigma_{s|t})^{-\frac{d}{2}} u^l(x_t, t, y) \exp\left(-\frac{1}{2\sigma_{s|t}^2}\left\|x_s - \frac{\alpha_{t|s}\sigma_s^2}{\sigma_t^2}x_t - \frac{\alpha_s\sigma_{t|s}^2}{\sigma_t^2}y\right\|^2\right) p_d^l(y)dy, \tag{21}$$

### 5.1   CONVERGENCE THEORY FOR DECENTRALIZED DPMS

Here, we leverage our local space partition analysis framework to rigorously establish the convergence rate for decentralized DPMs. Specifically, we:

**Theorem 5.2.** *There exist $C_1, C_2, C_3, C_4 > 0, k \geq 1$ depending on $t_{min}$, such that for all time discretizations $\mathcal{D}$ with $|\mathcal{D}| < \delta$, the inequality $TV(p(x_{t_{min}}), \hat{p}_{\theta,\phi}(x_{t_{min}})) \leq C_1 d|\mathcal{D}| \log^k \frac{1}{|\mathcal{D}|} + C_2\varepsilon_y + C_3\frac{L}{|\mathcal{D}|}\varepsilon_a + C_4 \exp\left(-\frac{1}{|\mathcal{D}|}\right)\sqrt{KL(p_d, \gamma^d)}$ holds.*

The proof idea of Theorem 5.2 still follows the idea of Theorem 3.7, the approach of first decomposing it into three components: the discretization error, the score approximation error introduced by the score network, and the class approximation error induced by the classifier network $a_\phi$.

$$\mathrm{TV}(p(x_s), \hat{p}_{\theta,\phi}(x_s)) \leq \underbrace{\mathrm{TV}(p(x_s), p_\theta(x_s))}_{\text{Global Class Approximation Error}} + \underbrace{\mathrm{TV}(p_\phi(x_s), \hat{p}_\phi(x_s))}_{\text{Global Discretization Error}} + \underbrace{\mathrm{TV}(\hat{p}_\phi(x_s), \hat{p}_{\phi,\theta}(x_s))}_{\text{Global Score Approximation Error}} \tag{22}$$

Notice that the forward error will be contained in the end step of the Global Discretization Error, and can be handled in the same way as in the Theorem 3.7. Notice that the analysis on forward error in the decentralized setting is totally the same as in Theorem 3.7.

**Class approximation error**   Here, we first quantify how much extra error will be introduced due to the decentralized classifier.

$$2\mathrm{TV}(p(x_s), p_\phi(x_s)) \leq L\varepsilon_a + 2\mathrm{TV}(p(x_t), p_\phi(x_t)). \tag{23}$$

It is noted that the local approximation error here is $L\varepsilon_a$, which will eventually sum up to the $C_3 T L\varepsilon_a$ term in Theorem 5.2.

**Discretization error**    For the discretization error, we first investigate its local form and bound it utilizing the sum of errors in each cluster.

$$2\text{TV}(p_\phi(x_s), \hat{p}_\phi(x_s))$$

$$\leq \sum_{l=1}^{L} a^l(x_t, t) \int_{\mathbb{R}^d} \int_{\mathbb{R}^d} \left| p_\phi^l(x_s|x_t)p_\phi(x_t) - \hat{p}_\phi^l(x_s|x_t)p_\phi(x_t) \right| \mathrm{d}x_t\, \mathrm{d}x_s + 2\text{TV}(p_\phi(x_t), \hat{p}_\phi(x_t)) \tag{24}$$

For every cluster $l$, we can bound the single-cluster local discretization error $\iint \left| p_\phi^l(x_s|x_t)p_\phi(x_t) - \hat{p}_\phi^l(x_s|x_t)p_\phi(x_t) \right| \mathrm{d}x_t\, \mathrm{d}x_s$ via the method in Section 4.1 with $\mathcal{O}((t-s)^2 \log^k \frac{1}{t-s})$. Considering that $\sum_{l=1}^{L} a^l = 1$, we can also bound the whole-cluster local discretization error with $\mathcal{O}((t-s)^2 \log^k \frac{1}{t-s})$. Of course, the global discretization error will be $\mathcal{O}((t-s) \log^k \frac{1}{t-s})$ and result in the $C_1 |\mathcal{D}| \log^k \frac{1}{|\mathcal{D}|}$ of Theorem 5.2.

**Score approximation error**    For the score approximation error part, we also use Pinsker's inequality first.

$$\text{TV}(\hat{p}_\phi(x_s), \hat{p}_{\phi,\theta}(x_s)) \leq \sqrt{\frac{1}{2}\text{KL}(\hat{p}_\phi(x_s), \hat{p}_{\phi,\theta}(x_s))} \tag{25}$$

For the KL term, we first divide them into different decentralized clusters.

$$\text{KL}(\hat{p}_\phi(x_s), \hat{p}_{\phi,\theta}(x_s))$$

$$\leq \sum_{l=1}^{L} a^l(x_t, t)\text{KL}\left( \hat{p}_\phi^l(x_s|x_t)\hat{p}_\phi(x_t), \hat{p}_{\phi,\theta}^l(x_s|x_t)\hat{p}_\phi(x_t) \right) + \text{KL}(\hat{p}_\phi(x_s), \hat{p}_{\phi,\theta}(x_s)) \tag{26}$$

For every cluster $l$, we can bound the single-cluster local score approximation error $\text{KL}\left( \hat{p}_\phi^l(x_s|x_t)\hat{p}_\phi(x_t), \hat{p}_{\phi,\theta}^l(x_s|x_t)\hat{p}_\phi(x_t) \right)$ via Lemma 4.3. Therefore, the whole-cluster local score approximation error can be bounded by $\mathcal{O}((t-s)\varepsilon_y^2)$. Then the global score approximation error in terms of KL is $\mathcal{O}(\varepsilon_y^2)$ and results in the $C_2 \varepsilon_y$ term in Theorem 5.2.

## 5.2  Handling the Divergent Classifier Approximation Error

The local approximation error introduced by $a_\phi$ is zero-order locally, which will sum up to a divergent $C_3 T L \varepsilon_a$ term in the global error as in Theorem 5.2. We identify that this is inherent in the current sampling method of decentralized DPMs as shown in Algorithm 3. To mitigate this divergence, we propose to theoretically use Ito's chain rule to characterize the behavior of $a_\phi$ based on its relation to $x_t$, but not directly predict $a_\phi$. The first derivative of the underlying true $a$ is as follows:

$$\mathrm{d}a(x_t, t) = \left( \frac{\partial a(x_t, t)}{\partial t} + \frac{\partial a^T(x_t, t)}{\partial x_t}\left( f_t x_t - g_t^2 s_t(x_t) \right) + \frac{1}{2}g_t^2 \Delta_{x_t} a(x_t, t) \right) dt + \frac{\partial a^T(x_t, t)}{\partial x_t} g_t dB_t \tag{27}$$

Notably, the $\frac{\partial a(x_t, t)}{\partial t}$, $\frac{\partial a(x_t, t)}{\partial x_t}$ and $\Delta_{x_t} a(x_t, t)$ all have analytical from which can be used as an enhanced high-order training objectives in the training of $a_\phi$, which is a technique used in classical DPMs (Lu et al., 2022a). Detailed formulation is deferred to Appendix C.

## 6  Conclusion

In this paper, we developed a novel convergence theory for DDPM using an intuitive pseudo-non-Markov analysis, determining an $O(d/T)$ convergence rate under the finite moments assumption. Moreover, we extend this method to establish the convergence rate of decentralized diffusion models, presenting the first convergence theory applicable to decentralized scenarios. Our work provides a new theoretical foundation and analysis approach for diffusion models. Looking ahead, the pseudo-non-Markov framework established in this study serves as a foundation for future research. It opens up new avenues for exploring the convergence properties of more complex and specialized diffusion models. Detailed proofs, more discussions, and experiments are deferred to the Appendix sections.

## ETHICS STATEMENT

As a nearly pure theoretical work, we feel our work is less likely to suffer from ethical issues. The authors have read and comply with the ICLR Code of Ethics. This research does not involve human subjects or personally identifiable information. We do not foresee harmful or dual-use implications from the proposed methods. There are no conflicts of interest or undisclosed sponsorship.

## REPRODUCIBILITY STATEMENT

For theoretical results, the derivations of the claims are included in the Appendix. We promise that we will open-source our empirical validation codes to promote the technological progress of the community.

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

# Appendix

## CONTENTS

## A  DETAILED DERIVATIONS

### A.1  THE NOTATION TABLE

Here, we list several notations of the theory in Table 2.

| Symbol | Meaning |
|---|---|
| $\gamma^d$ | The standard $d$-dimensional Gaussian distribution |
| $p_d \equiv p_0$ | The data distribution |
| $T$ | The number of sampling timesteps |
| $|\mathcal{D}|$ | The maximum stepsize |
| $\alpha_{t\|s}$ | The state coefficient from $s$ to $t$, where $\alpha_{t\|s} = \alpha_t/\alpha_s$ |
| $\sigma_{t\|s}$ | The variance coefficient from $s$ to $t$, where $\sigma_{t\|s}^2 = 1 - \alpha_{t\|s}^2$ |
| $u(x_t, t, y)$ | The density of $x_t$ noised by a data point $y$ at time $t$ |
| $w(x_t, t, y)$ | The probability of a state $x_t$ is noised by $y$ at time $t$ compared to other data |
| $p(x_s\|x_t), s < t$ | The true backward transition probability |
| $\hat{p}(x_s\|x_t), s < t$ | The single-Gaussian backward transition probability, true score |
| $p_\theta(x_s\|x_t), s < t$ | The single-Gaussian backward transition probability, network learned score |
| $\bigcup_{l=1}^{L} \{y_i^l \in \mathbb{R}^d \| i = 1, 2, \ldots, N_l\}$ | A $L$ clusters partition of the dataset |
| $L$ | The number of clusters |
| $L(x_0)$ | The cluster label of data point $x_0$ |
| $N_l$ | The number of data points in a cluster $l$ |
| $p^l(x_s\|x_t), s < t$ | The true backward transition, on the $l$ cluster |
| $\hat{p}^l(x_s\|x_t), s < t$ | The single-Gaussian backward transition, on the $l$ cluster, true score |
| $p_\theta^l(x_s\|x_t), s < t$ | The single-Gaussian backward transition, on the $l$ cluster, network learned score |

Table 2: The main symbols and their corresponding meanings used throughout the paper.

## A.2 DETAILS OF EQ. (14)

Here, we derive a decomposition that yields a recursive bound, showing that the error at step $s$ is the sum of the expected local discretization error from the transition approximation and the previous step error.

$$
\begin{aligned}
2\text{TV}(p(x_s), \tilde{p}(x_s)) &= \int_{\mathbb{R}^d} |p(x_s) - \tilde{p}(x_s)| \, \mathrm{d}x_s \\
&= \int_{\mathbb{R}^d} \left| \int_{\mathbb{R}^d} p(x_s|x_t)p(x_t) \, \mathrm{d}x_t - \int_{\mathbb{R}^d} \tilde{p}(x_s|x_t)\tilde{p}(x_t) \, \mathrm{d}x_t \right| \mathrm{d}x_s \\
&\leq \int_{\mathbb{R}^d} \left| \int_{\mathbb{R}^d} p(x_s|x_t)p(x_t) \, \mathrm{d}x_t - \int_{\mathbb{R}^d} \tilde{p}(x_s|x_t)p(x_t) \, \mathrm{d}x_t \right| \mathrm{d}x_s \\
&\quad + \int_{\mathbb{R}^d} \left| \int_{\mathbb{R}^d} \tilde{p}(x_s|x_t)p(x_t) \, \mathrm{d}x_t - \int_{\mathbb{R}^d} \tilde{p}(x_s|x_t)\tilde{p}(x_t) \, \mathrm{d}x_t \right| \mathrm{d}x_s \\
&= \int_{\mathbb{R}^d} \left| \int_{\mathbb{R}^d} p(x_s|x_t)p(x_t) - \tilde{p}(x_s|x_t)p(x_t) \, \mathrm{d}x_t \right| \mathrm{d}x_s \\
&\quad + \int_{\mathbb{R}^d} \underbrace{\left| \int_{\mathbb{R}^d} \tilde{p}(x_s|x_t) \, \mathrm{d}x_s \right|}_{=1} |p(x_t) - \tilde{p}(x_t)| \, \mathrm{d}x_t \\
&= \underbrace{\int_{\mathbb{R}^d} \left| \int_{\mathbb{R}^d} p(x_s|x_t)p(x_t) - \tilde{p}(x_s|x_t)p(x_t) \, \mathrm{d}x_t \right| \mathrm{d}x_s}_{\text{Local Discretization Error}} + 2\text{TV}(p(x_t), \tilde{p}(x_t))
\end{aligned}
\tag{28}
$$

The derivation begins by applying the law of total probability to express the marginal distributions $p(x_s)$ and $\tilde{p}(x_s)$ via the intermediate variable $x_t$. The core bounding step uses the triangle inequality to decompose the total difference into two terms by introducing a bridging distribution.

The first term (local error) quantifies the difference between the true and approximated conditional transitions, $|p(x_s|x_t) - \tilde{p}(x_s|x_t)|$, averaged over the true distribution $p(x_t)$. The second term isolates the difference in the distributions from the previous step, $|p(x_t) - \tilde{p}(x_t)|$. It simplifies because the integral of the conditional density $\int_{\mathbb{R}^d} \tilde{p}(x_s|x_t) \, \mathrm{d}x_s$ equals 1.

## A.3 DETAILS OF EQ. (23)

Here, we derive a recursive bound on the TV distance between the true distribution $p(x_s)$ and the score approximated distribution $p_\phi(x_s)$, which results from modeling the transition probability using

a mixture of kernels (or weighted transitions).

$$2\text{TV}(p(x_s), p_\phi(x_s))$$

$$= \iint |p(x_s|x_t)p(x_t) - p_\phi(x_s|x_t)p(x_t)| \, dx_t \, dx_s + 2\text{TV}(p(x_t), p_\phi(x_t))$$

$$= \iint \left| \sum_{l=1}^{L} a^l(x_t,t)p^l(x_s|x_t)p(x_t) - \sum_{l=1}^{L} a_\phi^l(x_t,t)p^l(x_s|x_t)p(x_t) \right| \, dx_t \, dx_s + 2\text{TV}(p(x_t), p_\phi(x_t)) \quad (29)$$

$$\leq \sum_{l=1}^{L} \iint \left| a^l(x_t,t) - a_\phi^l(x_t,t) \right| \left| p^l(x_s|x_t)p(x_t) \right| \, dx_t \, dx_s + 2\text{TV}(p(x_t), p_\phi(x_t)),$$

$$\leq L\varepsilon_a + 2\text{TV}(p(x_t), p_\phi(x_t)).$$

The second step introduces the class-cluster represented conditional distribution, where $p(x_s|x_t)$ is modeled as a weighted sum of $L$ basis kernels:

$$p(x_s|x_t) = \sum_{l=1}^{L} a^l(x_t,t)p^l(x_s|x_t).$$

The first inequality is achieved by using the triangle inequality for sums, factoring out the shared terms $p^l(x_s|x_t)$ and $p(x_t)$, and then using Fubini's theorem to separate the integrals. Since $p^l(x_s|x_t)$ is a probability density function, $\int_{\mathbb{R}^d} p^l(x_s|x_t) \, dx_s = 1$. This leaves the term $\left| a^l(x_t,t) - a_\phi^l(x_t,t) \right|$ averaged over $p(x_t)$.

The final inequality uses the definition of $\varepsilon_a$. The constant $L$ (the number of clusters) then appears as a multiplicative factor, completing the recursive bound: the error at step $s$ is bounded by the sum of the previous error from $x_t$ and the local error.

### A.4 DETAILS OF EQ. (24)

Here, we first divide the discretization error into different decentralized clusters.

$$2\text{TV}(p_\phi(x_s), \hat{p}_\phi(x_s))$$

$$= \iint |p_\phi(x_s|x_t)p_\phi(x_t) - \hat{p}_\phi(x_s|x_t)p_\phi(x_t)| \, dx_t \, dx_s + 2\text{TV}(p_\phi(x_t), \hat{p}_\phi(x_t))$$

$$= \iint \left| \sum_{l=1}^{L} a^l(x_t,t)p_\phi^l(x_s|x_t)p_\phi(x_t) - \sum_{l=1}^{L} a^l(x_t,t)\hat{p}_\phi^l(x_s|x_t)p_\phi(x_t) \right| \, dx_t \, dx_s + 2\text{TV}(p_\phi(x_t), \hat{p}_\phi(x_t))$$

$$\leq \sum_{l=1}^{L} a^l(x_t,t) \int_{\mathbb{R}^d} \int_{\mathbb{R}^d} \left| p_\phi^l(x_s|x_t)p_\phi(x_t) - \hat{p}_\phi^l(x_s|x_t)p_\phi(x_t) \right| \, dx_t \, dx_s + 2\text{TV}(p_\phi(x_t), \hat{p}_\phi(x_t))$$

$$(30)$$

In the second step, the terms $p_\phi(x_s|x_t)$ and $\hat{p}_\phi(x_s|x_t)$ are explicitly written as a mixture of cluster-specific kernels using the shared weight function $a^l(x_t,t)$ for both transitions.

The inequality is derived by applying the triangle inequality to the sum within the integral, factoring out the shared, non-negative terms $a^l(x_t,t)$ and $p_\phi(x_t)$, and then rearranging the order of integration using Fubini's Theorem. This leaves the bound as a weighted sum of the total variation distances between the individual kernel perturbations, $2\text{TV}(p_\phi^l(x_s|x_t), \hat{p}_\phi^l(x_s|x_t))$, averaged over the previous distribution $p_\phi(x_t)$.

## A.5 Details of Eq. (26)

Here, we divide the KL form score approximation error into different decentralized clusters.

$$
\begin{aligned}
&\mathrm{KL}(\hat{p}_\phi(x_s), \hat{p}_{\phi,\theta}(x_s)) \\
=&\mathrm{KL}\left(\hat{p}_\phi(x_s|x_t)\hat{p}_\phi(x_t), \hat{p}_{\phi,\theta}(x_s|x_t)\hat{p}_\phi(x_t)\right) + \mathrm{KL}(\hat{p}_\phi(x_s), \hat{p}_{\phi,\theta}(x_s)) \\
=&\mathrm{KL}\left(\sum_{l=1}^{L} a^l(x_t, t)\hat{p}_\phi^l(x_s|x_t)\hat{p}_\phi(x_t), \sum_{l=1}^{L} a^l(x_t, t)\hat{p}_{\phi,\theta}^l(x_s|x_t)\hat{p}_\phi(x_t)\right) + \mathrm{KL}(\hat{p}_\phi(x_s), \hat{p}_{\phi,\theta}(x_s)) \\
\leq&\sum_{l=1}^{L} a^l(x_t, t)\mathrm{KL}\left(\hat{p}_\phi^l(x_s|x_t)\hat{p}_\phi(x_t), \hat{p}_{\phi,\theta}^l(x_s|x_t)\hat{p}_\phi(x_t)\right) + \mathrm{KL}(\hat{p}_\phi(x_s), \hat{p}_{\phi,\theta}(x_s))
\end{aligned}
\tag{31}
$$

The derivation uses two main mathematical techniques: the Chain Rule for KL Divergence and the convexity property of KL Divergence.

The first line applies the Chain Rule for KL Divergence, which states that $\mathrm{KL}(P(X,Y)||Q(X,Y)) = \mathrm{KL}(P(X)||Q(X)) + \mathbb{E}_{P(X)}[\mathrm{KL}(P(Y|X)||Q(Y|X))]$.

In the subsequent equality, the conditional distributions are substituted with the assumed cluster mixture kernel form, where the weights $a^l(x_t, t)$ are identical for both $\hat{p}_\phi$ and $\hat{p}_{\phi,\theta}$.

The final step is an inequality ($\leq$) that uses the convexity of the KL Divergence (specifically, a form of Jensen's inequality). For a mixture distribution $\sum w_i P_i$, the divergence is bounded by $\mathrm{KL}(\sum w_i P_i || \sum w_i Q_i) \leq \sum w_i \mathrm{KL}(P_i||Q_i)$.

## A.6 Derivation of the forward error

The convergence error stems from the finite horizon of the forward process, which is the gap between $p(x_T)$ and $\tilde{p}(x_T)$.

Following the concept of Chen et al. (2022), we use the convergence (Bakry et al., 2013, Theorem 5.2.1) of the OU process in the KL divergence to prove that

$$
\mathrm{TV}(p(x_T), \tilde{p}(x_T)) \lesssim \exp\left(-\frac{1}{|\mathcal{D}|}\right)\sqrt{\mathrm{KL}(p_d, \gamma^d)}
\tag{32}
$$

Applying this result to the global convergence analysis will result in the third term of Theorem 3.7.

## A.7 Convergence on the singularity points.

**The left interval** Notice that the diffusion model has an inherent singularity property when $t_{\min} \to 0$ (Zhang et al., 2024). This implies that many quantities associated with diffusion models become ill-defined in the vicinity of $t = 0$. It's not feasible to compute the TV distance for this singularity endpoint. We adapt the idea from Theorem 2.1 in prior work (Lee et al., 2023), which applies the Wasserstein distance to the left interval.

**Proposition A.1.** *Given* $0 < t_{min} < 1$*, the 2-Wasserstein distance*

$$
W_2(p(x_0), p(x_{t_{min}})) < \sqrt{2dC_\alpha t_{min}}.
\tag{33}
$$

*Proof.*

$$
\begin{aligned}
W_2(p(x_0), p(x_{t_{min}})) &\leq \frac{1}{N}\sum_{i=1}^{N} W_2(\delta(x - y_i), \mathcal{N}(y_i, \sigma_{t_{min}}^2 I)) \\
&= \frac{1}{N}\sum_{i=1}^{N} \sqrt{d}\sigma_{t_{min}} \\
&\leq \frac{1}{N}\sum_{i=1}^{N} \sqrt{2dC_\alpha t_{min}} = \sqrt{2dC_\alpha t_{min}}.
\end{aligned}
\tag{34}
$$

$\square$

As Proposition A.1 ensures the convergence of diffusion probabilistic models (DPMs) in the vicinity of complex singularity points, the remaining sections of this paper will exclusively focus on convergence within the domain $t > t_{\min}$, rather than at the singularity point $t = 0$. This approach aligns with common practices in the literature, as exemplified by Li & Yan (2025).

We also found that, with an extra compact support assumption, we can indeed bound the TV distance between $p_0$ and $p_{t_{\min}}$ as shown in the following Proposition:

**Proposition A.2.** *Suppose the data distribution: $p_0(x)$ has a compact support $U$ (bounded by $M$ ) and $|p_0(x) - p_0(y)| \leqslant c_1|x - y|^2 \quad \forall x, y \in R$ for some $c_1 > 0$. We have that $TV(p_0, p_{t_{\min}}) \sim \mathcal{O}(dt_{\min}) \simeq \mathcal{O}(d\frac{1}{|\mathcal{D}|})$.*

*Proof.* For ease of notation, in this proof we denote that $\alpha = \alpha_{\min}, \sigma = \sigma_{t_{\min}}$. We first write out the analytical form of TV $(p_0, p_{t_{\min}})$:

$$2\text{TV}(p_0, p_{t_{\min}}) = \int \left| p_0(x) - \int \left(2\pi\sigma^2\right)^{-\frac{d}{2}} e^{-\frac{1}{2\sigma^2}|x - \alpha y|^2} p_0(y) dy \right| dx \qquad (35)$$

Then we use a trick to divide it into two parts:

$$2\text{TV}(p_0, p_{t_{\min}}) = \int \left| p_0(x) - \int \left(2\pi\sigma^2\right)^{-\frac{d}{2}} e^{-\frac{1}{2\sigma^2}|x + \alpha y|^2} \left(p_0(x) - p_0(y) - p_0(x)\right) dy \right| dx$$

$$\leq \int \left| p_0(x) - \int \left(2\pi\sigma^2\right)^{-\frac{d}{2}} e^{-\frac{1}{2\sigma^2}|x + \alpha y|^2} \left(p_0(x)\right) dy \right| dx \qquad (36)$$

$$+ \int \left| \int \left(2\pi\sigma^2\right)^{-\frac{d}{2}} e^{-\frac{1}{2\sigma^2}|x + \alpha y|^2} \left(p_0(x) - p_0(y)\right) dy \right| dx$$

First, we bound the first integral:

$$\int \left(2\pi\sigma^2\right)^{-\frac{d}{2}} e^{-\frac{1}{\sigma^2}(x - \alpha y)^2} p_0(x) dy = \alpha^{-d} \int \left(2\pi\frac{\sigma^2}{\alpha^2}\right)^{-\frac{d}{2}} e^{-\frac{\alpha^2}{2\sigma^2}\left|y - \frac{x}{\alpha}\right|^2} p_0(x) dy = \alpha^{-d} p_0(x) \qquad (37)$$

Thus, the first term will be:

$$\int \left| p_0(x) - \alpha^d p_0(x) \right| dx = \left| 1 - \alpha^d \right| = \frac{1}{1 - (1 - \alpha^d)} - 1 \leq 2\left(1 - \alpha^d\right) = 2\left(1 - \left(1 - \sigma^2\right)^{\frac{d}{2}}\right) \qquad (38)$$

where the result can be bounded by $\sigma^2 \sim \frac{1}{|\mathcal{D}|}$.

For the second integral, we deduce:

$$\int_U \left| \int_U \left(2\pi\sigma^2\right)^{-\frac{d}{2}} e^{-\frac{1}{2\sigma^2}(x - d)^2} \left(p_0(x) - p_0(y)\right) dy \right| dx$$

$$\leqslant \int_U |x - \alpha y + \alpha y - y|^2 \left(2\pi\sigma^2\right)^{-\frac{d}{2}} e^{-\frac{1}{2\sigma^2}|x - \alpha y|^2} dy dx$$

$$\leqslant \iint_U \psi \left(x - \alpha y|^2 + (\alpha - 1)^2|y|^2\right) \left(2\pi r^2\right)^{-\frac{d}{2}} e^{-\frac{1}{2\sigma^2}|x - \alpha y|^2} dy dx \qquad (39)$$

$$\leqslant 4d\sigma^2 + (\alpha - 1)^2 M^2 \int_U dy = \varphi d\sigma^2 + (\alpha - 1)^2 M \text{ Volumn }(V)$$

This part is also bounded by $\frac{1}{|\mathcal{D}|}$. $\qquad\qquad\square$

**The right interval** For the right interval, that is near $t_{\max}$, we improve the error bound from $(1 - s)^{\frac{1}{2}}$ (Zhang et al., 2024), to $(1 - s)^2$.

**Proposition A.3.** *For all $0 < s < 1$, there are constants $C_1, C_2 > 0$, such that $KL(p(x_s)||p_\theta(x_s)) \leq C_1(1 - s)^2 + C_2(1 - s)^2\varepsilon_y$.*

*proof.*

First, we consider the difference between $p(x_s)$ and $\tilde{p}(x_s)$. Since $p(x_1) = \tilde{p}(x_1) \sim \mathcal{N}(0, I)$,

$$
\begin{aligned}
&\mathrm{KL}(p(x_s)||\tilde{p}(x_s)) \\
&= \int_{\mathbb{R}^d} \int_{\mathbb{R}^d} p(x_s|x_1)p(x_1) \, \mathrm{d}x_1 \log \frac{\int_{\mathbb{R}^d} p(x_s|x_1)p(x_1) \, \mathrm{d}x_1}{\int_{\mathbb{R}^d} \tilde{p}(x_s|x_1)\tilde{p}(x_1) \, \mathrm{d}x_1} \, \mathrm{d}x_s \\
&\leq \int_{\mathbb{R}^d} \int_{\mathbb{R}^d} p(x_s|x_1)p(x_1) \log \frac{p(x_s|x_1)p(x_1)}{\tilde{p}(x_s|x_1)\tilde{p}(x_1)} \, \mathrm{d}x_1 \, \mathrm{d}x_s \\
&= \int_{\mathbb{R}^d} \int_{\mathbb{R}^d} p(x_s|x_1)p(x_1) \log \frac{p(x_s|x_1)}{\tilde{p}(x_s|x_1)} \, \mathrm{d}x_1 \, \mathrm{d}x_s \\
&= \int_{\mathbb{R}^d} (2\pi\sigma_s^2)^{-\frac{d}{2}} \int_{\mathbb{R}^d} \sum_i w_i(x_1, 1) \exp\left(-\frac{||x_s - \alpha_s y_i||^2}{2\sigma_s^2}\right) \log[ \\
&\qquad \frac{\sum_i w_i(x_1, 1) \exp\left(-\frac{||x_s - \alpha_s y_i||^2}{2\sigma_s^2}\right)}{\exp\left(-\frac{||x_s - \alpha_s \bar{y}(x_1, 1)||^2}{2\sigma_s^2}\right)}] \, \mathrm{d}x_s p(x_1) \, \mathrm{d}x_1 \\
&\leq \int_{\mathbb{R}^d} (2\pi\sigma_s^2)^{-\frac{d}{2}} \int_{\mathbb{R}^d} \sum_i w_i(x_1, 1) \exp\left(-\frac{||x_s - \alpha_s y_i||^2}{2\sigma_s^2}\right) \log[ \\
&\qquad \frac{\exp\left(-\frac{||x_s - \alpha_s y_i||^2}{2\sigma_s^2}\right)}{\exp\left(-\frac{||x_s - \alpha_s \bar{y}(x_1, 1)||^2}{2\sigma_s^2}\right)}] \, \mathrm{d}x_s p(x_1) \, \mathrm{d}x_1 \\
&\leq \int_{\mathbb{R}^d} (2\pi\sigma_s^2)^{-\frac{d}{2}} \int_{\mathbb{R}^d} \sum_i w_i(x_1, 1) \exp\left(-\frac{||x_s - \alpha_s y_i||^2}{2\sigma_s^2}\right) \\
&\qquad \cdot \frac{\alpha_s}{\sigma_s^2}(x_s - \alpha_s y_i)^T(y_i - \bar{y}(x_1, 1)) + \frac{\alpha_s^2}{2\sigma_s^2}||y_i - \bar{y}(x_1, 1)||^2 \, \mathrm{d}x_s p(x_1) \, \mathrm{d}x_1 \\
&\leq \int_{\mathbb{R}^d} \frac{\alpha_s^2}{2\sigma_s^2} M^2 p(x_1) \, \mathrm{d}x_1 \leq \frac{M^2}{2\sigma_{t_{min}}^2}\alpha_s^2 \leq \frac{C_\alpha^2 M^2}{2\sigma_{t_{min}}^2}(1-s)^2.
\end{aligned}
\tag{40}
$$

The method used to prove the previous Proposition cannot be applied to prove the main Proposition because there is a $\sigma_{s|t}^2$ in the denominator. This results in an error bound of $\sigma_{s|t}^2$, which does not allow for global convergence.

## B  CONVERGENCE PROOFS FOR DPM VIA PSEUDO-NON-MARKOV ANALYSIS

### B.1  TECHNICAL TOOLS

Here, we will first quantify the relation between the noise schedule transition step $\sigma_{t|s}^2$ and the time step gap $t - s$. Here, we show that these two terms exhibit the same convergence order.

**Lemma B.1.** $\sigma_{t|s}^2 \sim \mathcal{O}(t - s)$

*Proof.*

$$\frac{\alpha_s + \alpha_t}{\alpha_s^2}(\alpha_s - \alpha_t) = \frac{\alpha_s^2 - \alpha_t^2}{\alpha_s^2}$$

$$= 1 - \frac{\left(\alpha_s + \int_s^t \frac{d\alpha_z}{dz} dz\right)^2}{\alpha_s^2}$$

$$\leq 1 - \frac{[\alpha_s - C_\alpha(t-s)]^2}{\alpha_s^2} \tag{41}$$

$$= \frac{2C_\alpha(t-s)\alpha_s}{\alpha_s^2} + \frac{C_\alpha^2(t-s)^2}{\alpha_s^2}$$

$$\leq \frac{2C_\alpha}{\alpha_{t_{\max}}}(t-s) + O\left((t-s)^2\right)$$

$$= O\left(t-s\right)$$

$\square$

For completeness, we need to clearly show the convexity of the function in the form of $e^{a|x|^2}$. Later, it will be used in derivations of the Gaussian transition kernel.

**Lemma B.2.** *When $a > 0$, $e^{a|x|^2}$ is convex in $x$.*

*Proof.* $\nabla_x^2(e^{a|x|^2}) = e^{a|x|^2}(4a^2 xx^T + 2aI) \geq 0$ $\square$

As a corollary of Lemma B.2, we have the following convexity inequality in an integral form:

$$e^{a|\int(x_t - \alpha_t y)p(y|x_t) \, dy|^2} \leq \int e^{a|x_t - \alpha_t y|^2} p(y|x_t) \, dy \tag{42}$$

For completeness, we present the famous Markov inequality here.

**Lemma B.3.** *The exponential Markov inequality gives:*

$$P[\alpha X \geqslant \epsilon] \leqslant \frac{E\left[e^{\alpha X}\right]}{e^\varepsilon}, \epsilon > 0 \tag{43}$$

*The proof is omitted.*

Here, we present a lemma that will be used to tackle the peak region analysis of the cross-product space:

**Lemma B.4.** *If $|y| < |x|$, then $|x - y|^2 \leq (1 - \frac{|y|}{|x|})^2|x|^2$.*

*Proof.*

$$|x - y|^2 = |x - \frac{1}{|x|^2}y^T xx - (y - \frac{1}{|x|^2}y^T xx)|^2$$

$$= |x - \frac{1}{|x|^2}y^T xx|^2 + |(y - \frac{1}{|x|^2}y^T xx)|^2$$

$$\leq |x - \frac{1}{|x|^2}y^T xx|^2 \tag{44}$$

$$= (1 - \frac{y^T x}{|x|^2})^2|x|^2$$

$$\leq (1 - \frac{|y|}{|x|})^2|x|^2$$

$\square$

We derive the analytical form of the first-order derivatives of the time-aware diffused probability $v(x_t, t, y)$.

**Lemma B.5.** *Let* $v(x_t, t, y) \stackrel{def}{=} \exp\left(-\frac{|x_t - \alpha_t y|^2}{2\sigma_t^2}\right) = (2\pi\sigma_t^2)^{\frac{d}{2}} p(x_t|y)$, *we have* $\nabla v(x_t, t, y) = -v(x_t, t, y)\left(\frac{x_t - \alpha_t y}{\sigma_t^2}\right)$

*Proof.*

$$
\begin{aligned}
\nabla v(x_t, t, y) &= \nabla \exp\left(-\frac{|x_t - \alpha_t y|^2}{2\sigma_t^2}\right) \\
&= -\exp\left(-\frac{|x_t - \alpha_t y|^2}{2\sigma_t^2}\right)\left(\frac{x_t - \alpha_t y}{\sigma_t^2}\right) \qquad (45) \\
&= -v(x_t, t, y)\left(\frac{x_t - \alpha_t y}{\sigma_t^2}\right)
\end{aligned}
$$

$\square$

Last, we derive the analytical form of the second-order derivatives of $v(x_t, t, y)$.

**Lemma B.6.** $\nabla \bar{\epsilon}(x_t, t) = I - \iint p(y|x_t)p(y'|x_t)\frac{\alpha_t^2}{2\sigma_t^2}(y - y')(y - y')^T \, \mathrm{d}y \, \mathrm{d}y'$ *and also* $\nabla \bar{\epsilon}(x_t, t) = I - \int p(y|x_t)\frac{(x_t - \alpha_t y)(x_t - \alpha_t y)^T}{\sigma_t^2} \, \mathrm{d}y + \iint p(y|x_t)p(y'|x_t)\frac{(x_t - \alpha_t y)(x_t - \alpha_t y')^T}{\sigma_t^2} \, \mathrm{d}y \, \mathrm{d}y'$.

*Proof.*

$$
\begin{aligned}
\nabla \bar{\epsilon}(x_t, t) &= \nabla \frac{\int v(x_t, t, y)p(y)(x_t - \alpha_t y) \, \mathrm{d}y}{\int v(x_t, t, y')p(y') \, \mathrm{d}y'} \\
&= I + \left(\int v(x_t, t, y')p(y') \, \mathrm{d}y'\right)^2 (x_t - \alpha_t y)\left[-\int v(x_t, t, y)\frac{x_t - \alpha_t y}{\sigma_t^2}p(y) \, \mathrm{d}y \int v(x_t, t, y')p(y') \, \mathrm{d}y' \right. \\
&\quad \left. + \int v(x_t, y)p(y) \, \mathrm{d}y \int v(x_t, t, y')\frac{x_t - \alpha_t y'}{\sigma_t^2}p(y') \, \mathrm{d}y'\right] \\
&= I - \int p(y|x_t)\frac{(x_t - \alpha_t y)(x_t - \alpha_t y)^T}{\sigma_t^2} \, \mathrm{d}y + \iint p(y|x_t)p(y'|x_t)\frac{(x_t - \alpha_t y)(x_t - \alpha_t y')^T}{\sigma_t^2} \, \mathrm{d}y \, \mathrm{d}y' \\
&= I - \iint p(y|x_t)p(y'|x_t)\frac{\alpha_t^2}{2\sigma_t^2}(y - y')(y - y')^T \, \mathrm{d}y \, \mathrm{d}y'
\end{aligned}
$$

$$(46)$$

$\square$

## B.2 TECHNICAL LEMMAS

Remembers that we defined $u_{ts} = x_t - \alpha_{t|s}x_s$, $\epsilon_t = x_t - \alpha_t y$, $\epsilon_s = x_s - \alpha_s y$, $\bar{\epsilon}_t = x_t - \alpha_t \bar{y}(x_t, t)$.
We first establish a rearrangement transition formula, which is valid for arbitrary $y'$:

$$
(2\pi\sigma_{s|t}^2)^{-\frac{d}{2}} \exp\left(-\frac{1}{2\sigma_{s|t}^2}|x_s - \frac{\alpha_{t|s}\sigma_s^2}{\sigma_t^2}x_t - \frac{\alpha_s\sigma_{t|s}^2}{\sigma_t^2}y|^2\right)(2\pi\sigma_t^2)^{-\frac{d}{2}}\exp\left(-\frac{1}{2\sigma_t^2}|x_t - \alpha_t y'|^2\right)
$$

$$
=(2\pi\sigma_{s|t}^2)^{-\frac{d}{2}}\exp\left(-\frac{1}{2\sigma_{s|t}^2}|x_s - \alpha_{s|t}x_t + \frac{\alpha_{s|t}\sigma_{t|s}^2}{\sigma_t^2}(x_t - \alpha_t y)|^2\right)(2\pi\sigma_t^2)^{-\frac{d}{2}}\exp\left(-\frac{1}{2\sigma_t^2}|x_t - \alpha_t y'|^2\right)
$$

$$
=(2\pi\sigma_{t|s}^2)^{-\frac{d}{2}}\exp\left(-\frac{1}{2\sigma_{t|s}^2}|x_t - \alpha_{t|s}x_s + \frac{\alpha_t\sigma_{t|s}^2}{\sigma_t^2}(y - y')|^2\right)(2\pi\sigma_s^2)^{-\frac{d}{2}}\exp\left(-\frac{1}{2\sigma_s^2}|x_s - \alpha_s y + \frac{\alpha_{t|s}\alpha_t\sigma_s^2}{\sigma_t^2}(y - y')|^2\right)
$$

$$
=(2\pi\sigma_{t|s}^2)^{-\frac{d}{2}}\exp\left(-\frac{1}{2\sigma_{t|s}^2}|x_t - \alpha_{t|s}x_s + \frac{\alpha_t\sigma_{t|s}^2}{\sigma_t^2}(y - y')|^2\right)(2\pi\sigma_s^2)^{-\frac{d}{2}}\exp\left(-\frac{1}{2\sigma_s^2}|x_s - \frac{\alpha_s\sigma_{t|s}^2}{\sigma_t^2}y - \frac{\alpha_t^2\sigma_s^2}{\alpha_s\sigma_t^2}y'|^2\right)
$$

$$
=(2\pi\sigma_{t|s}^2)^{-\frac{d}{2}}\exp\left(-\frac{1}{2\sigma_{t|s}^2}|x_t - \alpha_{t|s}x_s + \frac{\alpha_t\sigma_{t|s}^2}{\sigma_t^2}(y - y')|^2\right)(2\pi\sigma_s^2)^{-\frac{d}{2}}\exp\left(-\frac{1}{2\sigma_s^2}|x_s - \alpha_s y' - \frac{\alpha_s\sigma_{t|s}^2}{\sigma_t^2}(y - y')|^2\right)
$$

$$\tag{47}$$

Now we start to delve into the structure of the conditional transition term

$$
p(x_s|x_t)p(x_t)
$$

$$
\overset{(15)}{=} \int (2\pi\sigma_{s|t}^2)^{-\frac{d}{2}}\exp\left(-\frac{1}{2\sigma_{s|t}^2}|x_s - \frac{\alpha_{t|s}\sigma_s^2}{\sigma_t^2}x_t - \frac{\alpha_s\sigma_{t|s}^2}{\sigma_t^2}y|^2\right)(2\pi\sigma_t^2)^{-\frac{d}{2}}\exp\left(-\frac{1}{2\sigma_t^2}|x_t - \alpha_t y|^2\right)p(y)\,\mathrm{d}y
$$

$$
\overset{(1)}{=} \int (2\pi\sigma_{s|t}^2)^{-\frac{d}{2}}\exp\left(-\frac{1}{2\sigma_{s|t}^2}|x_s - \alpha_{s|t}x_t + \frac{\alpha_{s|t}\sigma_{t|s}^2}{\sigma_t^2}(x_t - \alpha_t y)|^2\right)(2\pi\sigma_t^2)^{-\frac{d}{2}}\exp\left(-\frac{1}{2\sigma_t^2}|x_t - \alpha_t y|^2\right)p(y)\,\mathrm{d}y
$$

$$
\overset{(2)}{=} \int (2\pi\sigma_{s|t}^2)^{-\frac{d}{2}}\exp\left(-\frac{1}{2\sigma_{s|t}^2}|-\alpha_{s|t}u_{ts} + \frac{\alpha_{s|t}\sigma_{t|s}^2}{\sigma_t^2}\epsilon_t|^2\right)(2\pi\sigma_t^2)^{-\frac{d}{2}}\exp\left(-\frac{1}{2\sigma_t^2}|\epsilon_t|^2\right)p(y)\,\mathrm{d}y
$$

$$
\overset{(3)}{=} \int (2\pi\sigma_{t|s}^2)^{-\frac{d}{2}}\exp\left(-\frac{1}{\sigma_{t|s}^2}|u_{ts} + \frac{\sigma_{t|s}^2}{\sigma_t^2}\epsilon_t|^2\right)(2\pi\sigma_s^2)^{-\frac{d}{2}}\exp\left(-\frac{1}{2\sigma_s^2}\left|\epsilon_s - \frac{\alpha_{s|t}\sigma_{t|s}^2}{\sigma_t^2}\epsilon_t\right|^2\right)
$$

$$
\cdot \exp\left(\left(\frac{\alpha_{s|t}^2}{\sigma_t^2}u_{ts}^T\epsilon_t - \frac{\alpha_{s|t}^2\sigma_{t|s}^2}{2\sigma_t^2\sigma_s^2}\epsilon_t^2\right)\right)p(y)\,\mathrm{d}y
$$

$$\tag{48}$$

where (1) divide the $-\frac{\alpha_{s|t}\sigma_s^2}{\sigma_t^2}x_t$ apart; (2) is due to the definition of $u_{ts}$ and $\epsilon_t$; (3) is due to the rearrangement across different terms in exponents, we can simplify the calculation by leveraging the Eq. (47) with $y' = 2y - \frac{x_t}{\alpha_t}$.

Similarly, we have:

$$
\tilde{p}(x_s|x_t)p(x_t)
$$

$$
= \int \tilde{p}(x_s|x_t)p(x_t, y)\,\mathrm{d}y
$$

$$
= \int (2\pi\sigma_{s|t}^2)^{-\frac{d}{2}}\exp\left(-\frac{1}{2\sigma_{s|t}^2}|x_s - \frac{\alpha_{t|s}\sigma_s^2}{\sigma_t^2}x_t - \frac{\alpha_s\sigma_{t|s}^2}{\sigma_t^2}\bar{y}(x_t, t)|^2\right)(2\pi\sigma_t^2)^{-\frac{d}{2}}\exp\left(-\frac{1}{2\sigma_t^2}|x_t - \alpha_t y|^2\right)p(y)\,\mathrm{d}y
$$

$$
= \int (2\pi\sigma_{s|t}^2)^{-\frac{d}{2}}\exp\left(-\frac{1}{2\sigma_{s|t}^2}|x_s - \alpha_{s|t}x_t + \frac{\alpha_{s|t}\sigma_{t|s}^2}{\sigma_t^2}(x_t - \alpha_t\bar{y}(x_t, t))|^2\right)(2\pi\sigma_t^2)^{-\frac{d}{2}}\exp\left(-\frac{1}{2\sigma_t^2}|x_t - \alpha_t y|^2\right)p(y)\,\mathrm{d}y
$$

$$
= \int (2\pi\sigma_{t|s}^2)^{-\frac{d}{2}}\exp\left(-\frac{1}{\sigma_{t|s}^2}|u_{ts} + \frac{\sigma_{t|s}^2}{\sigma_t^2}\epsilon_t|^2\right)(2\pi\sigma_s^2)^{-\frac{d}{2}}\exp\left(-\frac{1}{2\sigma_s^2}\left|\epsilon_s - \frac{\alpha_{s|t}\sigma_{t|s}^2}{\sigma_t^2}\epsilon_t\right|^2\right)
$$

$$
\cdot \exp\left(\left(\frac{\alpha_{s|t}^2}{\sigma_t^2}u_{ts}^T\bar{\epsilon}_t - \frac{\alpha_{s|t}^2\sigma_{t|s}^2}{2\sigma_t^2\sigma_s^2}\bar{\epsilon}_t^2\right)\right)p(y)\,\mathrm{d}y
$$

$$\tag{49}$$

Let us denote that $A_1 = \{(x_t, y) : |x_t - \alpha_{t|s}x_s| \geq \sigma_{t|s}B\}$, $A_2 = \{(x_t, y) : \int |x_t - \alpha_t y'|^4 p(y'|x_t) \, \mathrm{d}y' \geq (\sigma_t B)^4\}$, $A_3 = \{(x_t, y)|x_t - \alpha_t y| \geq \sigma_t B\}$, $A_4 = \{(x_t, y) : |\bar{\epsilon}_t| \geq \sigma_t B\}$. We found that $A_1$ and $A_3$ exactly correspond to the conditions in $\mathbb{P}$, and we show that $A_4 \subset A_2$ because of the vector-function Jensen's inequality.

**Lemma B.7.** *We show that $A_4$ (on the measure space of $x_t$) is bounded at the order of $\sigma_{t|s}^5$.*

*Proof.*

$$P\left(|x_t - \alpha_t\bar{y}(x_t, t)| \geq \sigma_t B\right)$$

$$=P\left(\frac{|x_t - \alpha_t\bar{y}(x_t, t)|}{4\sigma_t^2} \geq \frac{B^2}{4}\right)$$

$$\overset{(1)}{\leq} e^{-\frac{B^2}{4}} \int \exp\left(\frac{|x_t - \alpha_t\bar{y}(x_t, t)|^2}{4\sigma_t^2}\right) p(x_t) \, \mathrm{d}x_t$$

$$\overset{(2)}{\leq} e^{-\frac{B^2}{4}} \iint \exp\left(\frac{|x_t - \alpha_t y|^2}{4\sigma_t^2}\right) p(y|x_t)p(x_t) \, \mathrm{d}y \, \mathrm{d}x_t$$

$$\overset{(3)}{=} e^{-\frac{B^2}{4}} \iint \exp\left(\frac{|x_t - \alpha_t y|^2}{4\sigma_t^2}\right) p(x_t|y)p(y) \, \mathrm{d}y \, \mathrm{d}x_t \tag{50}$$

$$\overset{(4)}{=} e^{-\frac{B^2}{4}} \iint \exp\left(\frac{|x_t - \alpha_t y|^2}{4\sigma_t^2}\right) (2\pi\sigma_t^2)^{-\frac{d}{2}} \exp\left(-\frac{|x_t - \alpha_t y|^2}{2\sigma_t^2}\right) p(y) \, \mathrm{d}y \, \mathrm{d}x_t$$

$$\overset{(5)}{=} e^{-\frac{B^2}{4}} \iint (2\pi\sigma_t^2)^{-\frac{d}{2}} \exp\left(-\frac{|x_t - \alpha_t y|^2}{4\sigma_t^2}\right) p(y) \, \mathrm{d}y \, \mathrm{d}x_t$$

$$\overset{(6)}{=} e^{\frac{Cd\log\left(\sqrt{\alpha_s}\sigma_{t|s}\right)}{4} + \frac{d}{2}\log 2}$$

$$\overset{(7)}{\leq} e^{\frac{Cd\log\left(\sqrt{\alpha_s}\sigma_{t|s}\right)}{8}} = \sqrt{\alpha_s}\sigma_{t|s}^{\frac{Cd}{8}} \overset{(8)}{\leq} \sigma_{t|s}^5$$

where (1) make use of the Lemma B.3; (2) utilize the definition of $\bar{y}$ and the result of Eq. (42); (3) apply Bayes's formula; (4) employ the analytical formula of $p(x_t|y)$; (5) combines the exponent terms; (6) is due to simplifying a Gaussian density; (7) is obtained by setting $C \geq -\frac{4\log 2}{\log\sqrt{\alpha_s}\sigma_{t|s}}$; (8) is attained by setting $C \geq \frac{40}{d\log\alpha_s}$. $\square$

We further extend the Lemma B.7 to an arbitrary exponential order form:

$$\int_{A_4} |x_t - \alpha_t\bar{y}|^n p(x_t, y) \, \mathrm{d}y \, \mathrm{d}x_t$$

$$\overset{(1)}{\leq} e^{-\frac{B^2}{5}} \int \exp\left(\frac{|x_t - \alpha_t\bar{y}(x_t, t)|^2}{5\sigma_t^2} + \log|x_t - \alpha_t\bar{y}|^n\right) p(x_t, y) \, \mathrm{d}x_t \, \mathrm{d}y \tag{51}$$

$$\overset{(2)}{\leq} e^{-\frac{B^2}{5}} \int \exp\left(\frac{|x_t - \alpha_t\bar{y}(x_t, t)|^2}{4\sigma_t^2}\right) p(x_t, y) \, \mathrm{d}x_t \, \mathrm{d}y$$

$$\overset{(3)}{\leq} \sigma_{t|s}^4,$$

where (1) makes use of the Lemma B.3 in the $x_t$ integral part and leaves $y$ unchanged; (2) is motivated by the faster growth of a polynomial than a logarithm, which can be guaranteed with

$$C \geq -\frac{e^{-W_{-1}(\max(-\frac{1}{10n\sigma_t^2}, e^{-1}))}}{\sigma_t^2 d\log\sigma_{t|s}} \Rightarrow \sigma_{t|s} \leq \exp\left(-\frac{e^{-W_{-1}(\max(-\frac{1}{10n\sigma_t^2}, e^{-1}))}}{\sigma_t^2 Cd}\right) \Rightarrow |x_t - \alpha_t\bar{y}|^2 \geq$$

$$\sigma_t Be^{-W_{-1}(\max(-\frac{1}{10n\sigma_t^2}, e^{-1}))} \Rightarrow \frac{|x_t - \alpha_t\bar{y}(x_t, t)|^2}{5\sigma_t^2} + \log|x_t - \alpha_t\bar{y}|^n \leq \frac{|x_t - \alpha_t\bar{y}(x_t, t)|^2}{4\sigma_t^2}, \text{ for } \frac{x^2}{20\sigma_t^2} \geq$$

$$\frac{n}{2}\log x^2 \Rightarrow \frac{1}{x^2}\log\frac{1}{x^2} \geq -\frac{1}{10n\sigma_t^2} \Rightarrow x^2 \geq e^{-W_{-1}(\max(-\frac{1}{10n\sigma_t^2}, e^{-1}))}; (3) \text{ is achieved in the same way}$$

as Lemma B.7 with $C \geq -\frac{20\log 2}{3\log\sqrt{\alpha_s}\sigma_{t|s}}$.

Then we start to analyze the probability of $A_2$. First, we prove an easier version, changing the order in $A_2$ to 1.

$$
\begin{aligned}
& P(\int_{\mathbb{R}^d} |x_t - \alpha_t y'| p(y'|x_t) \, \mathrm{d}y' \geq \sigma_t B) \\
& \leq e^{-\frac{B^2}{4}} \int \exp\left( \frac{|\int_{\mathbb{R}^d} |x_t - \alpha_t y'| p(y'|x_t) \, \mathrm{d}y'|^2}{4\sigma_t^2} \right) p(x_t) \, \mathrm{d}x_t \\
& \leq e^{-\frac{B^2}{4}} \iint \exp\left( \frac{|x_t - \alpha_t y'|^2}{4\sigma_t^2} \right) p(y'|x_t) p(x_t) \, \mathrm{d}y' \, \mathrm{d}x_t \\
& \leq \sigma_{t|s}^4
\end{aligned}
\tag{52}
$$

The proof technique is the same as the previous ones. However, establishing the bounds of the total probability of $A_2$ proves to be more challenging. To address this, we must utilize the local convexity concept.

**Lemma B.8.** *Let* $D \stackrel{def}{=} \{x_t, y : \int |x_t - \alpha_t y'|^n p(y'|x_t) \, \mathrm{d}y' \geq (\sigma_t B)^n\}$, *we have* $P(D) \leq \sigma_{t|s}^4$ *with sufficient large $C$.*

*Proof.* For a fixed $n$ and $x \geq 0$, consider the function $f(x) = e^{x^{\frac{1}{n}}}$. We wanna to establish the convexity inequality for this function on a part of its domain. $f'(x) = e^{x^{\frac{1}{n}}} \frac{1}{n} x^{\frac{1}{n}-1} = \frac{1}{n} x^{\frac{1}{n}-1} f(x) \geq 0$, $f''(x) = \frac{1}{n^2} x^{\frac{1}{n}-2}(x^{\frac{1}{n}} - (n-1))f(x)$. Thus when $x \geq (n-1)^n$, $f''(x) >= 0$ and when $x < (n-1)^n$, $f''(x) < 0$. As a result, $f(x) \geq 1$ is increasing and is convex in $[(n-1)^n, +\infty]$ and is concave in $[0, (n-1)^n]$. Now we want to establish a bounding box of this concave region, and find domains where the curve line will go below the concave box. Since it is increasing in $[0, (n-1)^n]$, it is easy to verify $f([0, (n-1)^n]) \subset [0, (n-1)^n] \times [0, e^{n-1}] \stackrel{def}{=} D_c$. Since it is convex in $[(n-1)^n, +\infty]$, for all $y_0, y \in [(n-1)^n, +\infty]$, $f(y) \geq f(y_0) + f'(y_0)(y - y_0)$. Note that $y_0 \geq (n-1)^n$, and if $y_0 \geq 2(n-1)^n$,

$$
\begin{aligned}
& y_0 \geq 2(n-1)^n \geq (2n-1)(n-1)^{n-1} \\
& \Rightarrow (n-1)^{1-n}(y_0 - (n-1)^n) \geq n \\
& \Rightarrow y_0^{\frac{1}{n}-1}(y_0 - (n-1)^n) \geq n \\
& \Rightarrow 1 + \frac{1}{n} y_0^{\frac{1}{n}-1}((n-1)^n - y_0) \leq 0 \\
& \Rightarrow f(y_0) + f'(y_0)((n-1)^n - y_0) \leq 0.
\end{aligned}
\tag{53}
$$

Thus the line $f(y_0) + f'(y_0)(y - y_0)$ do not intersect with $f([0, (n-1)^n])$. Thus $f(y) \geq f(y_0) + f'(y_0)(y - y_0)$ for all $y \in [0, (n-1)^n]$. As a result, $f(y) \geq f(y_0) + f'(y_0)(y - y_0)$ for all $y \in [0, +\infty]$.

Thus if $\sigma_t B \geq 2^{\frac{1}{n}}(n-1)$, then we have $y_0 \geq (\sigma_t B)^n$ indicates $f(y) \geq f(y_0) + f'(y_0)(y - y_0)$.

Consider the random variable $\xi = |\epsilon_t|^n \in \mathbb{R}$ with its distribution $p(\xi)$. If $E\xi \geq (\sigma_t B)^n$, which is equivalent to $\int |x_t - \alpha_t y|^n p(y|x_t) \, \mathrm{d}y \geq (\sigma_t B)^n$, we have $f(\xi) \geq f(E\xi) + f'(E\xi)(\xi - E\xi)$ and $Ef(\xi) \geq f(E\xi)$, which means $\int f(\xi) p(\xi) \, \mathrm{d}\xi \geq f(\int \xi p(\xi) \, \mathrm{d}\xi)$ and $\int e^{|\epsilon_t|} p(y|x_t) \, \mathrm{d}y \geq e^{[\int |\epsilon_t|^n p(y|x_t) \, \mathrm{d}y]^{\frac{1}{n}}} \geq e^{\sigma_t B}$.

With the same derivation to $f(x) = e^{x^{\frac{2}{n}}}$, we have if $\sigma_t B$ is large enough, we have $\int e^{|\epsilon_t|^2} p(y|x_t) \, \mathrm{d}y \geq e^{[\int |\epsilon_t|^n p(y|x_t) \, \mathrm{d}y]^{\frac{2}{n}}} \geq e^{\sigma_t^2 B^2}$, or $\int e^{\frac{|\epsilon_t|^2}{4\sigma_t^2}} p(y|x_t) \, \mathrm{d}y \geq e^{\frac{B^2}{4}}$

$$P(D) = \int_D p(x_t)\,\mathrm{d}(x_t)$$

$$\overset{(1)}{\le} e^{-\frac{B^2}{4}} \int e^{\frac{1}{4\sigma_t^2}\left[\int |\frac{\epsilon_t}{2\sigma_t}|^n p(y|x_t)\,\mathrm{d}y\right]^{\frac{2}{n}}} p(x_t)\,\mathrm{d}x_t$$

$$\overset{(2)}{\le} e^{-\frac{B^2}{4}} \iint e^{\frac{|\epsilon_t|^2}{4\sigma_t^2}} p(x_t, y)\,\mathrm{d}y\,\mathrm{d}x_t$$

$$= e^{-\frac{B^2}{4}}(2\pi 2\sigma_t^2)^{-\frac{d}{2}} 2^{\frac{d}{2}} \iint e^{-\frac{|\epsilon_t|^2}{4\sigma_t^2}} p(y)\,\mathrm{d}x_t\,\mathrm{d}y$$

$$= e^{\frac{Cd \log \sigma_{t|s}}{4} + \frac{d}{2}\log 2}$$

$$\le e^{\frac{Cd \log \sigma_{t|s}}{8}}$$

$$= \sigma_{t|s}^{\frac{Cd}{4}} \le \sigma_{t|s}^4. \tag{54}$$

Here, (1) makes the use of the previous Markov's inequality; (2) makes the use of the above locally convex inequality of $f(x) = e^{x^{\frac{2}{n}}}$; and the subsequent derivations leverage the same technique as before.

Let $n = 4$, we have $P(A_2) \le \sigma_{t|s}^4$. $\qquad\square$

Then we are going to prove that the probability of $A_3$ is small.

$$P(|x_t - \alpha_t y| \ge \sigma_t B)$$

$$\le e^{-\frac{B^2}{4}} \int \exp\left(\frac{|x_t - \alpha_t y|^2}{4\sigma_t^2}\right) p(x_t, y)\,\mathrm{d}y\,\mathrm{d}x_t$$

$$= e^{-\frac{B^2}{4}} \iint (2\pi\sigma_t^2)^{-\frac{d}{2}} \exp\left(-\frac{|x_t - \alpha_t y|^2}{4\sigma_t^2}\right) p(y)\,\mathrm{d}y\,\mathrm{d}x_t$$

$$= e^{\frac{Cd \log \sigma_{t|s}}{4} + \frac{d}{2}\log 2} \le \sigma_{t|s}^4. \tag{55}$$

The bound of the expectation of $|x_t - \alpha_t y|^n$ on $A_3$ is as follows:

$$\int_{A_3} |x_t - \alpha_t y|^n p(x_t, y)\,\mathrm{d}y\,\mathrm{d}x_t$$

$$\le e^{-\frac{B^2}{5}} \int \exp\left(\frac{|x_t - \alpha_t y|^2}{5\sigma_t^2} + \log|x_t - \alpha_t y|^n\right) p(x_t, y)\,\mathrm{d}y\,\mathrm{d}x_t$$

$$\le e^{-\frac{B^2}{5}} \int \exp\left(\frac{|x_t - \alpha_t y|^2}{4\sigma_t^2}\right) p(x_t, y)\,\mathrm{d}y\,\mathrm{d}x_t \tag{56}$$

$$\le \sigma_{t|s}^4 e^{-\frac{B^2}{10}}$$

$$\le \sigma_{t|s}^4$$

Note that the proof on $A_3$ here is very similar to that of $A_4$, except that here we don't need the application of Jensen's inequality to take the integral to the outside. Similarly, we have $\int_{A_3} \epsilon(x_t, y) p(x_t, y)\,\mathrm{d}y\,\mathrm{d}x_t \le \sigma_{t|s}^4 e^{-\frac{B^2}{10}}$.

**Lemma B.9.** *The integral of $p(x_s|x_t)p(x_t)$ on $A_1$ is bounded by $4\sqrt{\alpha_s}\sigma_{t|s}^4$.*

*Proof.*

$$\int_{\{|x_t-\alpha_{t|s}x_s|\geq\sigma_{t|s}B\}} p(x_s|x_t)p(x_t)\,\mathrm{d}x_s\,\mathrm{d}x_t$$

$$=\int_{\{|x_t-\alpha_{t|s}x_s|\geq\sigma_{t|s}B\}} (2\pi\sigma_{s|t}^2)^{-\frac{d}{2}}\exp\left(-\frac{1}{2\sigma_{s|t}^2}\left|x_s-\frac{\alpha_{t|s}\sigma_s^2}{\sigma_t^2}x_t-\frac{\alpha_s\sigma_{t|s}^2}{\sigma_t^2}y\right|^2\right)(2\pi\sigma_t^2)^{-\frac{d}{2}}$$

$$\cdot\exp\left(-\frac{1}{2\sigma_t^2}|x_t-\alpha_t y|^2\right)p(y)\,\mathrm{d}y\,\mathrm{d}x_s\,\mathrm{d}x_t$$

$$\overset{(1)}{=}\int_{\{|x_t-\alpha_{t|s}x_s|\geq\sigma_{t|s}B\}} (2\pi\sigma_{t|s}^2)^{-\frac{d}{2}}\exp\left(-\frac{1}{2\sigma_{t|s}^2}|x_t-\alpha_{t|s}x_s|^2\right)(2\pi\sigma_s^2)^{-\frac{d}{2}}\exp\left(-\frac{1}{2\sigma_s^2}|x_s-\alpha_s y|^2\right)p(y)\,\mathrm{d}y\,\mathrm{d}x_s\,\mathrm{d}x_t$$

$$=\int_{\{|u_{ts}|\geq\sigma_{t|s}B\}} (2\pi\sigma_{t|s}^2)^{-\frac{d}{2}}\exp\left(-\frac{|u_{ts}|^2}{2\sigma_{t|s}^2}\right)(2\pi\sigma_s^2)^{-\frac{d}{2}}\exp\left(-\frac{1}{2\sigma_s^2}|x_s-\alpha_s y|^2\right)p(y)\,\mathrm{d}y\,\mathrm{d}x_s\,\mathrm{d}x_t$$

$$\overset{(2)}{\leq}4\exp\left(-\frac{B^2}{8d}\right)\int(2\pi\sigma_s^2)^{-\frac{d}{2}}\exp\left(-\frac{1}{2\sigma_s^2}|x_s-\alpha_s y|^2\right)p(y)\,\mathrm{d}y\,\mathrm{d}x_s$$

$$=4\exp\left(-\frac{B^2}{8d}\right)=4\sqrt{\alpha_s}\sigma_{t|s}^{\frac{C}{8}}\overset{(3)}{\leq}4\sqrt{\alpha_s}\sigma_{t|s}^4$$

$$\tag{57}$$

where (1) makes the use of the conclusion of Eq. (48); (2) leverages the estimation theorem on the tail probability of Gaussian as discussed in (Rhee & Talagrand, 1986); and we need $C\geq 32$ in (3). □

In the realm of $(x_t,y)\in A_2^c\cap A_3^c$, we have:

$$\frac{\alpha_t\sigma_{t|s}^2}{\sigma_t^2}|\bar{y}-y|$$

$$=\frac{\sigma_{t|s}^2}{\sigma_t^2}|(x_t-\alpha_t y)-(x_t-\alpha_t\bar{y})|$$

$$\overset{(1)}{<}\frac{\sigma_{t|s}^2}{\sigma_t^2}2\sigma_t B$$

$$=\frac{2}{\sigma_t}\sigma_{t|s}^2\sqrt{-Cd\log\alpha_s\sigma_{t|s}}\leq\frac{\alpha_s\sigma_{t|s}}{\sigma_t}$$

$$\tag{58}$$

where (1) is given by the definition of $A_2$ and $A_3$; (2) is achieved by setting $\sigma_{t|s}\leq e^{\frac{1}{2}W_{-1}(-\frac{1}{2Cd})}$.

### B.3 PROOF OF LEMMA 4.1

Obviously, the Lemma 4.1 can be divided into the following three parts.

**Lemma B.10.** *For all $t_{min}\leq s<t\leq 1$, there exist $\delta>0$ and $C>0$, the integral of local discretization error from $s$ to $t$ on the $\mathbb{P}_1$ is bounded by $C(t-s)^2$.*

**Lemma B.11.** *For all $t_{min}\leq s<t\leq 1$, there exist $\delta>0$ and $C>0$, the integral of local discretization error from $s$ to $t$ on the $\mathbb{P}_2$ is bounded by $C(t-s)^2$.*

**Lemma B.12.** *For all $t_{min}\leq s<t\leq 1$, there exist $\delta>0$ and $C>0$, the integral of local discretization error from $s$ to $t$ on the $\mathbb{P}_3$ is bounded by $C(t-s)^2$.*

### B.3.1 PROOF OF LEMMA B.10

$$\int_{\mathbb{P}_1} |p(x_s|x_t)p(x_t) - \tilde{p}(x_s|x_t)p(x_t)| \, \mathrm{d}x_s \, \mathrm{d}x_t$$

$$= \int_{A_1 \cap (A_2 \cup A_3)} |p(x_s|x_t)p(x_t) - \tilde{p}(x_s|x_t)p(x_t)| \, \mathrm{d}x_s \, \mathrm{d}x_t \tag{59}$$

$$\leq \underbrace{\int_{A_1 \cap (A_2 \cup A_3)} p(x_s|x_t)p(x_t) \, \mathrm{d}x_s \, \mathrm{d}x_t}_{\text{First part}} + \underbrace{\int_{A_1 \cap (A_2 \cup A_3)} \tilde{p}(x_s|x_t)p(x_t) \, \mathrm{d}x_s \, \mathrm{d}x_t}_{\text{Second part}}$$

For the first part, we have:

$$\int_{A_1 \cap (A_2 \cup A_3)} p(x_s|x_t)p \, \mathrm{d}x_s \, \mathrm{d}x_t$$

$$\leq \int_{A_1} p(x_s|x_t)p \, \mathrm{d}x_s \, \mathrm{d}x_t \tag{60}$$

$$\overset{\text{Lemma B.9}}{\leq} 4\sqrt{\alpha_s}\sigma_{t|s}^4$$

For the second part, we have:

$$\int_{A_1 \cap (A_2 \cup A_3)} \tilde{p}(x_s|x_t)p(x_t) \, \mathrm{d}x_s \, \mathrm{d}x_t$$

$$= \int_{A_1 \cap (A_2 \cup A_3)} \tilde{p}(x_s|x_t, y)p(x_t, y) \, \mathrm{d}x_s \, \mathrm{d}x_t \, \mathrm{d}y$$

$$\overset{(1)}{=} \int_{A_1 \cap (A_2 \cup A_3)} p(x_t, y) \, \mathrm{d}x_t \, \mathrm{d}y \tag{61}$$

$$\overset{(2)}{\leq} \int_{A_2} + \int_{A_3} p(x_t, y) \, \mathrm{d}x_t \, \mathrm{d}y \overset{(3)}{\leq} 2\sigma_{t|s}^4$$

where (1) integrates over the Gaussian form $x_s$; (2) broaden the integral region; (3) uses the conclusion of Lemma B.8 and Eq. (55).

Thus, we have $\int_{\mathbb{P}_1} p(x_s|x_t)p(x_t) - \tilde{p}(x_s|x_t)p(x_t) \, \mathrm{d}x_s \, \mathrm{d}x_t \leq (4\sqrt{\alpha_s} + 2)\sigma_{t|s}^4$

### B.3.2 PROOF OF LEMMA B.11

$$\int_{\mathbb{P}_2} p(x_s|x_t)p(x_t) - \tilde{p}(x_s|x_t)p(x_t) \, \mathrm{d}x_s \, \mathrm{d}x_t$$

$$= \int_{A_1 \cap A_2^c \cap A_3^c} p(x_s|x_t)p(x_t) - \tilde{p}(x_s|x_t)p(x_t) \, \mathrm{d}x_s \, \mathrm{d}x_t \tag{62}$$

$$\leq \underbrace{\int_{A_1 \cap A_2^c \cap A_3^c} p(x_s|x_t)p(x_t) \, \mathrm{d}x_s \, \mathrm{d}x_t}_{\text{First part}} + \underbrace{\int_{A_1 \cap A_2^c \cap A_3^c} \tilde{p}(x_s|x_t)p(x_t) \, \mathrm{d}x_s \, \mathrm{d}x_t}_{\text{Second part}}$$

For the first part, we have:

$$\int_{A_1 \cap A_2^c \cap A_3^c} p(x_s|x_t)p \, \mathrm{d}x_s \, \mathrm{d}x_t$$

$$\leq \int_{A_1} p(x_s|x_t)p \, \mathrm{d}x_s \, \mathrm{d}x_t \tag{63}$$

$$\overset{\text{Lemma B.9}}{\leq} 4\sqrt{\alpha_s}\sigma_{t|s}^4$$

For the second part, we have:

$$\int_{A_1 \cap A_2^c \cap A_3^c} \tilde{p}(x_s|x_t)p(x_t)\, \mathrm{d}x_s\, \mathrm{d}x_t$$

$$\overset{(1)}{=} \int_{A_1 \cap A_2^c \cap A_3^c} (2\pi\sigma_{t|s}^2)^{-\frac{d}{2}} \exp\left(-\frac{1}{2\sigma_{t|s}^2}|x_t - \alpha_{t|s}x_s + \frac{\alpha_t \sigma_{t|s}^2}{\sigma_t^2}(\bar{y}-y)|^2\right)(2\pi\sigma_s^2)^{-\frac{d}{2}}$$

$$\cdot \exp\left(-\frac{1}{2\sigma_s^2}|x_s - \alpha_s y - \frac{\alpha_s \sigma_{t|s}^2}{\sigma_t^2}(\bar{y}-y)|^2\right)\mathrm{d}y\, \mathrm{d}x_s\, \mathrm{d}x_t$$

$$\overset{(2)}{\leq} \int_{A_1 \cap A_2^c \cap A_3^c} (2\pi\sigma_{t|s}^2)^{-\frac{d}{2}} \exp\left(-\frac{1}{2\sigma_{t|s}^2}(1 - \frac{\alpha_t \sigma_{t|s}^2}{\sigma_t^2}\frac{|\bar{y}-y|}{|x_t - \alpha_{t|s}x_s|})|x_t - \alpha_{t|s}x_s|^2\right)$$

$$\cdot (2\pi\sigma_s^2)^{-\frac{d}{2}} \exp\left(-\frac{1}{2\sigma_s^2}|x_s - \alpha_s y - \frac{\alpha_s \sigma_{t|s}^2}{\sigma_t^2}(\bar{y}-y)|^2\right)\mathrm{d}y\, \mathrm{d}x_s\, \mathrm{d}x_t$$

$$\overset{(3)}{=} \int_{A_1 \cap A_2^c \cap A_3^c} (2\pi\sigma_{t|s}^2)^{-\frac{d}{2}} \exp\left(-\frac{1}{2\sigma_{t|s}^2}(1 - \frac{\alpha_t \sigma_{t|s}^2}{\sigma_t^2}\frac{|\bar{y}-y|}{|x_t - \alpha_{t|s}x_s|})|x_t - \alpha_{t|s}x_s|^2\right)$$

$$\cdot (2\pi\sigma_s^2)^{-\frac{d}{2}} \exp\left(-\frac{1}{2\sigma_s^2}|x_s - \alpha_s y - \frac{\alpha_s \sigma_{t|s}^2}{\sigma_t^2}(\bar{y}-y)|^2\right)\mathbb{1}\{|x_s - \alpha_s y| \leq \frac{2}{\sigma_t}\sigma_{t|s}^2 B\}\,\mathrm{d}y\, \mathrm{d}x_s\, \mathrm{d}x_t$$

$$+ \int_{A_1 \cap A_2^c \cap A_3^c} (2\pi\sigma_{t|s}^2)^{-\frac{d}{2}} \exp\left(-\frac{1}{2\sigma_{t|s}^2}(1 - \frac{\alpha_t \sigma_{t|s}^2}{\sigma_t^2}\frac{|\bar{y}-y|}{|x_t - \alpha_{t|s}x_s|})|x_t - \alpha_{t|s}x_s|^2\right)$$

$$\cdot (2\pi\sigma_s^2)^{-\frac{d}{2}} \exp\left(-\frac{1}{2\sigma_s^2}|x_s - \alpha_s y - \frac{\alpha_s \sigma_{t|s}^2}{\sigma_t^2}(\bar{y}-y)|^2\right)\mathbb{1}\{|x_s - \alpha_s y| > \frac{2}{\sigma_t}\sigma_{t|s}^2 B\}\,\mathrm{d}y\, \mathrm{d}x_s\, \mathrm{d}x_t$$

$$\overset{(4)}{\leq} \int_{A_1 \cap A_2^c \cap A_3^c} (2\pi\sigma_{t|s}^2)^{-\frac{d}{2}} \exp\left(-\frac{1}{2\sigma_{t|s}^2}(1 - \frac{\alpha_t \sigma_{t|s}^2}{\sigma_t^2}\frac{|\bar{y}-y|}{|x_t - \alpha_{t|s}x_s|})|x_t - \alpha_{t|s}x_s|^2\right)$$

$$\cdot (2\pi\sigma_s^2)^{-\frac{d}{2}} \mathbb{1}\{|x_s - \alpha_s y| \leq \frac{2}{\sigma_t}\sigma_{t|s}^2 B\}\,\mathrm{d}y\, \mathrm{d}x_s\, \mathrm{d}x_t$$

$$+ \int_{A_1 \cap A_2^c \cap A_3^c} (2\pi\sigma_{t|s}^2)^{-\frac{d}{2}} \exp\left(-\frac{1}{2\sigma_{t|s}^2}(1 - \frac{\alpha_t \sigma_{t|s}^2}{\sigma_t^2}\frac{|\bar{y}-y|}{|x_t - \alpha_{t|s}x_s|})|x_t - \alpha_{t|s}x_s|^2\right)$$

$$\cdot (2\pi\sigma_s^2)^{-\frac{d}{2}} \exp\left(-\frac{1}{2\sigma_s^2}(1 - \frac{\alpha_s \sigma_{t|s}^2}{\sigma_t^2}\frac{|\bar{y}-y|}{|x_s - \alpha_s y|})^2|x_s - \alpha_s y|^2\right)\mathbb{1}\{|x_s - \alpha_s y| > \frac{2}{\sigma_t}\sigma_{t|s}^2 B\}\,\mathrm{d}y\, \mathrm{d}x_s\, \mathrm{d}x_t$$

$$\overset{(5)}{\leq} \int_{A_1 \cap A_2^c \cap A_3^c} (2\pi\sigma_{t|s}^2)^{-\frac{d}{2}} \exp\left(-\frac{1}{2\sigma_{t|s}^2}(1 - \frac{\alpha_t \sigma_{t|s}^2}{\sigma_t^2}\frac{\frac{2}{\sigma_t}\sigma_{t|s}^2 B}{\sigma_{t|s}B})|x_t - \alpha_{t|s}x_s|^2\right)$$

$$\cdot (2\pi\sigma_s^2)^{-\frac{d}{2}} \mathbb{1}\{|x_s - \alpha_s y| \leq \frac{2}{\sigma_t}\sigma_{t|s}^2 B\}\,\mathrm{d}y\, \mathrm{d}x_s\, \mathrm{d}x_t$$

$$+ \int_{A_1 \cap A_2^c \cap A_3^c} (2\pi\sigma_{t|s}^2)^{-\frac{d}{2}} \exp\left(-\frac{1}{2\sigma_{t|s}^2}(1 - \frac{\alpha_t \sigma_{t|s}^2}{\sigma_t^2}\frac{\frac{2}{\sigma_t}\sigma_{t|s}^2 B}{\sigma_{t|s}B})|x_t - \alpha_{t|s}x_s|^2\right)$$

$$\cdot (2\pi\sigma_s^2)^{-\frac{d}{2}} \exp\left(-\frac{1}{2\sigma_s^2}(1 - \frac{\alpha_s \sigma_{t|s}^2}{\sigma_t^2}\frac{\frac{2}{\sigma_t}\sigma_{t|s}^2 B}{\frac{2}{\sigma_t}\sigma_{t|s}^2 B})^2|x_s - \alpha_s y|^2\right)\mathbb{1}\{|x_s - \alpha_s y| > \frac{2}{\sigma_t}\sigma_{t|s}^2 B\}\,\mathrm{d}y\, \mathrm{d}x_s\, \mathrm{d}x_t$$

$$\overset{(6)}{\leq} \underbrace{(1 - \frac{2\alpha_t \sigma_{t|s}^3}{\sigma_t^3})^d}_{\leq 1} 4\sigma_{t|s}^{\frac{C}{2}}(\frac{2}{\sigma_t}\sigma_{t|s}^2 B)^d + \underbrace{(1 - \frac{2\alpha_t \sigma_{t|s}^3}{\sigma_t^3})^d}_{\leq 1} 4\sigma_{t|s}^{\frac{C}{2}} \underbrace{(1 - \frac{\alpha_s \sigma_{t|s}^2}{\sigma_t^2})^d}_{\leq 1}$$

$$\leq 8\sigma_{t|s}^4.$$

$$(64)$$

where (1) uses Eq. (47) but replace $y$ with $\bar{y}$; (2) uses Lemma B.4 with $x = x_t - \alpha_{t|s}x_s$ and $y = \frac{\alpha_s \sigma_{t|s}^2}{\sigma_t^2}(\bar{y}-y)$; (3) is the division of the space via the condition of $|x_s - \alpha_s y| \leq \frac{2}{\sigma_t}\sigma_{t|s}^2 B$; (4) is because that in the region of $|x_s - \alpha_s y| \leq \frac{2}{\sigma_t}\sigma_{t|s}^2 B$, we have the inside of the exponent $\exp\left(-\frac{1}{2\sigma_s^2}(1 - \frac{\alpha_s \sigma_{t|s}^2}{\sigma_t^2}\frac{|\bar{y}-y|}{|x_s - \alpha_s y|})^2|x_s - \alpha_s y|^2\right)$ is smaller than 1; (4) also uses Lemma B.4 on the

$x_s$ part; (5) uses the bound on $|\bar{y} - y|$ and $|x_t - \alpha_{t|s}x_s|$ in Eq. (58) and the definition of $A_3$; (6) sorts Gaussian form out and only leaves the coefficients.

Thus, we have $\int_{\mathbb{P}_2} p(x_s|x_t)p(x_t) - \tilde{p}(x_s|x_t)p(x_t)\,\mathrm{d}x_s\,\mathrm{d}x_t \leq (4\sqrt{\alpha_s} + 8)\sigma_{t|s}^4$.

### B.3.3 PROOF OF LEMMA B.12

$$
\int \left| \iint_{A_1^c \cap (A_2 \cup A_3)} p(x_s|x_t, y)p(x_t, y) - \tilde{p}(x_s|x_t, y)p(x_t, y)\,\mathrm{d}y\,\mathrm{d}x_t \right| \mathrm{d}x_s
$$

$$
\overset{(1)}{\leq} \iint_{A_1^c \cap (A_2 \cup A_3)} \left| \int p(x_s|x_t, y)\,\mathrm{d}x_s \right| \mathrm{d}y\,\mathrm{d}x_t + \iint_{A_1^c \cap (A_2 \cup A_3)} \left| \int \tilde{p}(x_s|x_t, y)\,\mathrm{d}x_s \right| \mathrm{d}y\,\mathrm{d}x_t \tag{65}
$$

$$
\overset{(2)}{\leq} \iint_{A_2 \cup A_3} 2p(x_t, y)\,\mathrm{d}y\,\mathrm{d}x_t
$$

$$
\overset{(3)}{\leq} 2\sigma_{t|s}^4
$$

Here, (1) uses a triangle inequality; (2) relaxes the constraint on $x_s$ to the $\mathbb{R}^d$ and thus makes it a full Gaussian, which integrates to 1; (3) uses the conclusion of Lemma B.8 and Eq. (55).

### B.4 PROOF OF LEMMA 4.2

Then we focus on $(x_t, y) \in A_1^c \cap A_2^c \cap A_3^c$, where $|x_t - \alpha_{t|s}x_s| < \sigma_{t|s}B, |x_t - \alpha_t\bar{y}| < \sigma_t B$, $|x_t - \alpha_t y| \leq \sigma_t B$. In this region, we have the following bounds on $|\bar{y} - y|$ and $|y - y'|$ according to the definition of the $A_2$ and $A_3$.

$$
\alpha_t|\bar{y} - y| = |(x_t - \alpha_t y) - (x_t - \alpha_t\bar{y})| \leq |x_t - \alpha_t y| + |x_t - \alpha_t\bar{y}| \leq 2\sigma_t B \tag{66}
$$

$$
\alpha_t|y - y'| = |(x_t - \alpha_t y') - (x_t - \alpha_t y)| \leq |x_t - \alpha_t y'| + |x_t - \alpha_t y| \leq 2\sigma_t B \tag{67}
$$

Here, we want to analyze by conducting Taylor's expansion of $\bar{y}$ on $x_t$ at the point of $\alpha_{t|s}x_s$.

First, we want to control the operator norm of $\nabla\bar{\epsilon}(x_t, t)$.

**Lemma B.13.** $\|\nabla\bar{\epsilon}(x_t, t)\| \leq (1 + 2(\sigma_t B)^2)$

*Proof.* We will prove that, for arbitrary $\xi \in \mathbb{R}^d$ and $|\xi| = n$, we have $|\nabla\bar{\epsilon}(x_t, t) \cdot \xi \otimes \xi| \leq n^2(1 + 2(\sigma_t B)^2)$ We first quantify the bound of $\int |\epsilon_t|^2 p(y|x_t)\,\mathrm{d}y$ when $x_t \in A_2$.

$$
\int_{A_1^c \cap A_2^c \cap A_3^c} |\epsilon_t|^2 p(y|x_t)\,\mathrm{d}y = \left[ \left| \int_{A_1^c \cap A_2^c \cap A_3^c} |\epsilon_t|^2 p(y|x_t)\,\mathrm{d}y \right|^2 \right]^{\frac{1}{2}}
$$

$$
\overset{(1)}{\leq} \left[ \int_{A_1^c \cap A_2^c \cap A_3^c} |\epsilon_t|^4 p(y|x_t)\,\mathrm{d}y \right]^{\frac{1}{2}} \tag{68}
$$

$$
\overset{(2)}{\leq} (\sigma_t B)^2,
$$

Here, (1) uses the Cauchy inequality; (2) uses the conclusion of Lemma B.8. Similarly, we can also deduce that $\int |\epsilon_t| p(y|x_t)\,\mathrm{d}y \leq \sigma_t B$ and $\int |\epsilon_t|^3 p(y|x_t)\,\mathrm{d}y \leq (\sigma_t B)^3$.

Then we can direct bound $\int |\epsilon_t|^2 p(y|x_t)\,\mathrm{d}y$ when $x_t \in A_1^c \cap A_2^c \cap A_3^c$ as follows

$$
|\nabla\bar{\epsilon}(x_t, t) \cdot \xi \otimes \xi| = |n^2 - \int (\xi^T \epsilon_t)^2 p(y|x_t)\,\mathrm{d}y + (\int (\xi^T \epsilon_t)p(y|x_t)\,\mathrm{d}y)^2|
$$

$$
\leq n^2 + n^2 \int |\epsilon_t|^2 p(y|x_t)\,\mathrm{d}y + (n \int |\epsilon_t| p(y|x_t)\,\mathrm{d}y)^2 \tag{69}
$$

$$
= n^2(1 + 2(\sigma_t B)^2)
$$

$\square$

We further bound the operator norm of $\nabla\bar{\epsilon}(x_t, t)$.

**Lemma B.14.** *The norm of $\nabla^2\bar{\epsilon}(x_t, t)$ is bounded by $\frac{8}{\sigma_t}B^3$, when $(x_t, y) \in A_1^c \cap A_2^c \cap A_3^c$.*

*Proof.* Please note that the following integral is conducted on $(x_t, y) \in A_1^c \cap A_2^c \cap A_3^c$. We first give out its analytical form:

$$2\sigma_t^2\nabla^2\bar{\epsilon}(x_t, t)$$

$$= \nabla(2\sigma_t^2 I - \iint p(y|x_t)p(y'|x_t)\alpha_t^2(y - y')(y - y')^T \, dy \, dy')$$

$$= -\nabla(\int v(x_t, t, y')p(y') \, dy')^{-2} \iint v(x_t, t, y)v(x_t, t, y')p(y)p(y')\alpha_t^2(y - y')(y - y')^T \, dy \, dy'$$

$$= -(\int v(x_t, t, y')p(y') \, dy')^{-4}[-\iint v(x_t, t, y)v(x_t, t, y')[(\frac{x_t - \alpha_t y}{\sigma_t^2} + \frac{x_t - \alpha_t y'}{\sigma_t^2})(\int v(x_t, t, y')p(y') \, dy')^2$$

$$+ 2(\int v(x_t, t, y')p(y') \, dy')\int v(x_t, t, y'')p(y'')\frac{x_t - \alpha_t y''}{\sigma_t^2} \, dy''] \otimes \alpha_t(y - y') \otimes \alpha_t(y - y')p(y)p(y') \, dy \, dy'$$

$$= \iint p(y|x_t)p(y'|x_t)[(\frac{x_t - \alpha_t y}{\sigma_t^2} + \frac{x_t - \alpha_t y'}{\sigma_t^2}) - \int p(y''|x_t)2\frac{x_t - \alpha_t y''}{\sigma_t^2} \, dy'']$$

$$\otimes \alpha_t(y - y') \otimes \alpha_t(y - y')p(y)p(y') \, dy \, dy'$$

$$= \frac{1}{\sigma_t^2}\iiint p(y|x_t)p(y'|x_t)p(y''|x_t)\alpha_t(y'' - y + y'' - y') \otimes \alpha_t(y - y') \otimes \alpha_t(y - y') \, dy \, dy' \, dy''$$

$$= \frac{1}{\sigma_t^2}\iiint p(y|x_t)p(y'|x_t)p(y''|x_t)(\epsilon_t + \epsilon_t' - 2\epsilon_t'') \otimes (\epsilon_t' - \epsilon_t) \otimes (\epsilon_t' - \epsilon_t) \, dy \, dy' \, dy''$$

$$\tag{70}$$

Denote $z_{n,t} = \int |\epsilon_t|^n p(y|x_t) \, dy$. For arbitrary $\xi \in \mathbb{R}^d$ and $|\xi| = n$, we have

$$|2\sigma_t^4\nabla\bar{\epsilon}(x_t, t) \cdot \xi \otimes \xi \otimes \xi|$$

$$= |\iiint p(y|x_t)p(y'|x_t)p(y''|x_t)\xi^T(\epsilon_t + \epsilon_t' - 2\epsilon_t'')(\xi^T(\epsilon_t' - \epsilon_t)^2 \, dy \, dy' \, dy''|$$

$$\leq n^3 \iiint p(y|x_t)p(y'|x_t)p(y''|x_t)|\epsilon_t + \epsilon_t' - 2\epsilon_t''||\epsilon_t' - \epsilon_t|^2 \, dy \, dy' \, dy''$$

$$\leq \iiint p(y|x_t)p(y'|x_t)p(y''|x_t)(|\epsilon_t| + |\epsilon_t'| + 2|\epsilon_t''|)(\epsilon_t'^2 + \epsilon_t^2 + 2|\epsilon_t||\epsilon_t'|) \, dy \, dy' \, dy''$$

$$\leq \iiint p(y|x_t)p(y'|x_t)p(y''|x_t)|\epsilon_t|^3 + |\epsilon_t'|^3 + 3|\epsilon_t|^2|\epsilon_t'| + 3|\epsilon_t||\epsilon_t'|^2 + 2|\epsilon_t''||\epsilon_t|^2 + 2|\epsilon_t''||\epsilon_t'|^2$$

$$+ 4|\epsilon_t''||\epsilon_t||\epsilon_t'| \, dy \, dy' \, dy''$$

$$= z_{3,t} + z_{3,t} + 3z_{2,t}z_{1,t} + 3z_{1,t}z_{2,t} + 2z_{1,t}z_{2,t} + 2z_{1,t}z_{2,t} + 4z_{1,t}z_{1,t}z_{1,t}$$

$$\leq 16n^3(\sigma_t B)^3$$

$$\tag{71}$$

$\square$

Similar to Lemma B.13 and B.14, higher order derivatives of $\bar{\epsilon}(x_t, t)$ w.r.t. $x_t$ is also bounded within the domain of $A_1^c \cap A_2^c \cap A_3^c$.

Let $\epsilon_t = x_t - \alpha_t y, \bar{\epsilon}_t = x_t - \alpha_t\bar{y}(x_t, t), \epsilon_s = x_s - \alpha_s y, u_t = x_t - \alpha_{t|s}x_s, \bar{\epsilon}_s = x_t - \alpha_s\bar{y}(\alpha_{t|s}x_s, t)$.

$$\int \left| \iint_{\mathbb{P}_4} p(x_s|x_t,y)p(x_t,y) - \tilde{p}(x_s|x_t,y)p(x_t,y) \, \mathrm{d}y \, \mathrm{d}x_t \right| \mathrm{d}x_s$$

$$= \int \left| \iint_{A_1^c \cap A_2^c \cap A_3^c} p(x_s|x_t,y)p(x_t,y) - \tilde{p}(x_s|x_t,y)p(x_t,y) \, \mathrm{d}y \, \mathrm{d}x_t \right| \mathrm{d}x_s$$

$$= \int \left| \iint_{A_1^c \cap A_2^c \cap A_3^c} (2\pi\sigma_{s|t}^2)^{-\frac{d}{2}} \exp\left(-\frac{|-\alpha_{s|t}u_t|^2}{2\sigma_{s|t}^2}\right) (2\pi\sigma_t^2)^{-\frac{d}{2}} \exp\left(-\frac{|\epsilon_t|^2}{2\sigma_t^2}\right) \right.$$
$$\left. \cdot \left(\exp\left(\frac{\alpha_{s|t}^2}{\sigma_s^2}u_t^T\bar{\epsilon}_t - \frac{\alpha_{s|t}^2\sigma_{s|t}^2}{2\sigma_s^4}\bar{\epsilon}_t^2\right) - \exp\left(\frac{\alpha_{s|t}^2}{\sigma_s^2}u_t^T\epsilon_t - \frac{\alpha_{s|t}^2\sigma_{s|t}^2}{2\sigma_s^4}\epsilon_t^2\right)\right) p(y) \, \mathrm{d}y \, \mathrm{d}x_t \right| \mathrm{d}x_s \tag{72}$$

$$= \int \left| \iint_{A_1^c \cap A_2^c \cap A_3^c} (2\pi\sigma_{t|s}^2)^{-\frac{d}{2}} \exp\left(-\frac{|u_t + \frac{\sigma_{t|s}^2}{\sigma_t^2}\epsilon_t|^2}{2\sigma_{t|s}^2}\right) \exp\left(-\frac{|\epsilon_s - \frac{\alpha_{s|t}\sigma_{t|s}^2}{\sigma_t^2}\epsilon_t|^2}{2\sigma_s^2}\right) \right.$$
$$\left. \cdot \left(\exp\left(\frac{\alpha_{s|t}^2}{\sigma_s^2}u_t^T\bar{\epsilon}_t - \frac{\alpha_{s|t}^2\sigma_{s|t}^2}{2\sigma_s^4}\bar{\epsilon}_t^2\right) - \exp\left(\frac{\alpha_{s|t}^2}{\sigma_s^2}u_t^T\epsilon_t - \frac{\alpha_{s|t}^2\sigma_{s|t}^2}{2\sigma_s^4}\epsilon_t^2\right)\right) p(y) \, \mathrm{d}y \, \mathrm{d}x_t \right| \mathrm{d}x_s$$

Here, the derivation is the result of Eq. 48, 49 and 47.

From Eq. (72), we can see that we sorting out two Gaussian form terms, $\exp\left(-\frac{|u_t + \frac{\sigma_{t|s}^2}{\sigma_t^2}\epsilon_t|^2}{2\sigma_{t|s}^2}\right)$

and $\exp\left(-\frac{|\epsilon_s - \frac{\alpha_{s|t}\sigma_{t|s}^2}{\sigma_t^2}\epsilon_t|^2}{2\sigma_s^2}\right)$, the difficult part is the difference of $\exp\left(\frac{\alpha_{s|t}^2}{\sigma_s^2}u_t^T\bar{\epsilon}_t - \frac{\alpha_{s|t}^2\sigma_{s|t}^2}{2\sigma_s^4}\bar{\epsilon}_t^2\right)$ and

$\exp\left(\frac{\alpha_{s|t}^2}{\sigma_s^2}u_t^T\epsilon_t - \frac{\alpha_{s|t}^2\sigma_{s|t}^2}{2\sigma_s^4}\epsilon_t^2\right)$. These two terms possess the same form, only different in $\epsilon_t$ and $\bar{\epsilon}_t$. From previous Eq. (66), these two terms should be close in the $\mathbb{P}_4$. Then we are motivated to do Taylor's expansion to quantify the error.

$|\frac{\alpha_{s|t}^2}{\sigma_s^2}u_t^T\bar{\epsilon}_t - \frac{\alpha_{s|t}^2\sigma_{s|t}^2}{2\sigma_s^4}\bar{\epsilon}_t^2| \leq \frac{\sigma_{t|s}+\sigma_{t|s}^2}{\alpha_{min}^2\sigma_{t_{min}}^2}B^2 \stackrel{\text{def}}{=} C_1$ and $|\frac{\alpha_{s|t}^2}{\sigma_s^2}u_t^T\epsilon_t - \frac{\alpha_{s|t}^2\sigma_{s|t}^2}{2\sigma_s^4}\epsilon_t^2| \leq \frac{\sigma_{t|s}+\sigma_{t|s}^2}{\alpha_{min}^2\sigma_{t_{min}}^2}B^2$ are

finite. These are due to the definition of $A_2$ and $A_3$. This ensure that the $\exp\left\{\frac{\alpha_{s|t}^2}{\sigma_s^2}u_t^T\bar{\epsilon}_t - \frac{\alpha_{s|t}^2\sigma_{s|t}^2}{2\sigma_s^4}\bar{\epsilon}_t^2\right\}$

and $\exp\left\{\frac{\alpha_{s|t}^2}{\sigma_s^2}u_t^T\epsilon_t - \frac{\alpha_{s|t}^2\sigma_{s|t}^2}{2\sigma_s^4}\epsilon_t^2\right\}$ is finite, which makes their Lagrangian remainder of Taylor's expansion is finite.

Notice that, here our constant $C \sim \mathcal{O}(B^2)$, and $B$ is defined as $\sqrt{-Cd\log\sqrt{\alpha_s}\sigma_{t|s}}$, so $C \sim \mathcal{O}(d)$. This is why our discretization error results in Theorem 3.7 has a linear dependence on $d$.

Also, the derivative of $\bar{\epsilon}_t$ is bounded as shown in Lemma B.13 and B.14. As long as they are bounded, we can do Maclaurin expansions on both of them.

Define $a = \frac{\alpha_{s|t}^2}{\sigma_s^2}$, $b = \frac{\alpha_{s|t}^2}{2\sigma_s^4}$, then $a = 2\sigma_s^2 b$.

$$\exp\left(au_t^T \bar{\epsilon}_t - b\sigma_{t|s}^2 \bar{\epsilon}_t^2\right)$$

$$= 1 + \left[au_t^T \bar{\epsilon}_t - b\sigma_{t|s}^2 \bar{\epsilon}_t^2\right] + \frac{1}{2}\left[au_t^T \bar{\epsilon}_t - b\sigma_{t|s}^2 \bar{\epsilon}_t^2\right]^2 + \frac{1}{6}\left[au_t^T \bar{\epsilon}_t - b\sigma_{t|s}^2 \bar{\epsilon}_t^2\right]^3$$

$$+ \frac{1}{24}\left[au_t^T \bar{\epsilon}_t - b\sigma_{t|s}^2 \bar{\epsilon}_t^2\right]^4 + \frac{1}{120}e^{\eta_1}\left[au_t^T \bar{\epsilon}_t - b\sigma_{t|s}^2 \bar{\epsilon}_t^2\right]^5$$

$$= 1 + au_t^T \bar{\epsilon}_t - b\sigma_{t|s}^2 \bar{\epsilon}_t^2 + \frac{1}{2}a^2(u_t^T \bar{\epsilon}_t)^2 - ab\sigma_{t|s}^2 u_t^T \bar{\epsilon}_t \bar{\epsilon}_t^2 + \frac{1}{2}b^2\sigma_{t|s}^4 \bar{\epsilon}_t^2$$

$$+ \frac{1}{6}a^3(u_t^T \bar{\epsilon}_t)^3 - \frac{1}{2}a^2 b\sigma_{t|s}^2(u_t^T \bar{\epsilon}_t)^2\bar{\epsilon}_t^2 + \frac{1}{2}ab^2\sigma_{t|s}^4 u_t^T \bar{\epsilon}_t \bar{\epsilon}_t^4 - \frac{1}{6}b^3\sigma_{t|s}^6 \bar{\epsilon}_t^6$$

$$+ \frac{1}{24}a^4(u_t^T \bar{\epsilon}_t)^4 - \frac{1}{6}a^3 b\sigma_{t|s}^2(u_t^T \bar{\epsilon}_t)^3\bar{\epsilon}^2 + \frac{1}{4}a^2 b^2\sigma_{t|s}^4(u_t^T \bar{\epsilon}_t)^2\bar{\epsilon}^4 - \frac{1}{6}ab^3\sigma_{t|s}^6 u_t^T \bar{\epsilon}_t \bar{\epsilon}^6 + \frac{1}{24}b^4\sigma_{t|s}^8 \bar{\epsilon}_t^8$$

$$+ c_1\sigma_{t|s}^4 \log\sigma_{t|s}^{-1} + \mathcal{O}(\sigma_{t|s}^5 \log^5 \sigma_{t|s})$$

$$= 1 + au_t^T \bar{\epsilon}_t - b\sigma_{t|s}^2 \bar{\epsilon}_t^2 + \frac{1}{2}a^2(u_t^T \bar{\epsilon}_t)^2 - ab\sigma_{t|s}^2 u_t^T \bar{\epsilon}_t \bar{\epsilon}_t^2 + \frac{1}{2}b^2\sigma_{t|s}^4 \bar{\epsilon}_t^4$$

$$+ \frac{1}{6}a^3(u_t^T \bar{\epsilon}_t)^3 - \frac{1}{2}a^2 b\sigma_{t|s}^2(u_t^T \bar{\epsilon}_t)^2\bar{\epsilon}_t^2 + \frac{1}{24}a^4(u_t^T \bar{\epsilon}_t)^4 + c_2\sigma_{t|s}^4 \log\sigma_{t|s}^{-1} + \mathcal{O}(\sigma_{t|s}^5 \log^5 \sigma_{t|s})$$

(73)

where $\eta_1$ is some value between 0 and before mentioned maximum of $au_t^T \bar{\epsilon}_t - b\sigma_{t|s}^2 \bar{\epsilon}_t^2$, which can be as small as we want as $\sigma_{t|s}$ gets smaller. We can do the same for $\epsilon_t$.

$$\exp\left(au_t^T \epsilon_t - b\sigma_{t|s}^2 \epsilon_t^2\right)$$

$$= 1 + au_t^T \epsilon_t - b\sigma_{t|s}^2 \epsilon_t^2 + \frac{1}{2}a^2(u_t^T \epsilon_t)^2 - ab\sigma_{t|s}^2 u_t^T \epsilon_t \epsilon_t^2 + \frac{1}{2}b^2\sigma_{t|s}^4 \epsilon_t^4$$

$$+ \frac{1}{6}a^3(u_t^T \epsilon_t)^3 - \frac{1}{2}a^2 b\sigma_{t|s}^2(u_t^T \epsilon_t)^2\epsilon_t^2 + \frac{1}{24}a^4(u_t^T \epsilon_t)^4 + c_2\sigma_{t|s}^4 \log\sigma_{t|s}^{-1} + \mathcal{O}(\sigma_{t|s}^5 \log^5 \sigma_{t|s})$$

(74)

The fifth-order term of the expansion will be integral to the fifth-order result of $\sigma_{s|t}$, due to the previous Gaussian term will be integral to 1.

For the first-order term in the expansion, we need the $\bar{\epsilon}_t$ and $\epsilon_t$ to do a cancellation. We deduce that:

$$\int\left|\iint_{A_1^c \cap A_2^c \cap A_3^c}(2\pi\sigma_{s|t}^2)^{-\frac{d}{2}}\exp\left(-\frac{|-\alpha_{s|t}u_t|^2}{2\sigma_{s|t}^2}\right)(2\pi\sigma_t^2)^{-\frac{d}{2}}\exp\left(-\frac{|\epsilon_t|^2}{2\sigma_t^2}\right)\cdot\left(au_t^T \bar{\epsilon}_t - au_t^T \epsilon_t\right)p(y)\,\mathrm{d}y\,\mathrm{d}x_t\right|\mathrm{d}x_s$$

$$= \int\left|\iint_{A_1^c \cap A_2^c \cap A_3^c}(2\pi\sigma_{s|t}^2)^{-\frac{d}{2}}\exp\left(-\frac{|-\alpha_{s|t}u_t|^2}{2\sigma_{s|t}^2}\right)\cdot\left(au_t^T \bar{\epsilon}_t - au_t^T \epsilon_t\right)p(x_t,y)\,\mathrm{d}y\,\mathrm{d}x_t\right|\mathrm{d}x_s$$

$$\overset{(1)}{\leq} \int\left|\iint_{A_1^c \cap A_2^c}(2\pi\sigma_{s|t}^2)^{-\frac{d}{2}}\exp\left(-\frac{|-\alpha_{s|t}u_t|^2}{2\sigma_{s|t}^2}\right)\cdot\left(au_t^T \bar{\epsilon}_t - au_t^T \epsilon_t\right)p(x_t,y)\,\mathrm{d}y\,\mathrm{d}x_t\right|\mathrm{d}x_s$$

$$+ \int\left|\iint_{A_1^c \cap A_2^c \cap A_3}(2\pi\sigma_{s|t}^2)^{-\frac{d}{2}}\exp\left(-\frac{|-\alpha_{s|t}u_t|^2}{2\sigma_{s|t}^2}\right)\cdot\left(au_t^T \bar{\epsilon}_t\right)p(x_t,y)\,\mathrm{d}y\,\mathrm{d}x_t\right|\mathrm{d}x_s$$

$$+ \int\left|\iint_{A_1^c \cap A_2^c \cap A_3}(2\pi\sigma_{s|t}^2)^{-\frac{d}{2}}\exp\left(-\frac{|-\alpha_{s|t}u_t|^2}{2\sigma_{s|t}^2}\right)\cdot\left(au_t^T \epsilon_t\right)p(x_t,y)\,\mathrm{d}y\,\mathrm{d}x_t\right|\mathrm{d}x_s$$

$$\overset{(1)}{\leq} 0 + \frac{\alpha_{s|t}^2}{\sigma_s^2}*\sigma_{t|s}B*\sigma_{s|t}^4 + \frac{\alpha_{s|t}^2}{\sigma_s^2}*\sigma_{t|s}B*\sigma_{s|t}^4 = \mathcal{O}(\sigma_{s|t}^4 \log^k \sigma_{s|t})$$

(75)

Here, (1) is due to the triangle inequality over space. For (2), the first inequality holds because the integral of $\epsilon$ over the whole space of $y$ is exactly $\bar{\epsilon}$; the second inequality of (2) is due to the bound in Eq. (55), bounding $\exp\left(-\frac{|-\alpha_{s|t}u_t|^2}{2\sigma_{s|t}^2}\right)$ with 1, bounding $u_t^T$ using the definition of $A_1^c$, and using $e^{-\frac{B^2}{10}}$ to absorb the $(2\pi\sigma_{s|t}^2)^{-\frac{d}{2}}$. The third inequality of (2) is similar, using $\int_{A_3}\epsilon(x_t,y)p(x_t,y)\,\mathrm{d}y\,\mathrm{d}x_t \leq \sigma_{t|s}^4 e^{-\frac{B^2}{10}}$.

For the second-order term, we can handle the $\bar{\epsilon}_t$ and $\epsilon_t$ respectively.

$$
\int \left| \iint_{A_1^c \cap A_2^c \cap A_3^c} (2\pi\sigma_{t|s}^2)^{-\frac{d}{2}} \exp\left( -\frac{|u_t + \frac{\sigma_{t|s}^2}{\sigma_t^2}\epsilon_t|^2}{2\sigma_{t|s}^2} \right) \exp\left( -\frac{|\epsilon_s - \frac{\alpha_{s|t}\sigma_{t|s}^2}{\sigma_t^2}\epsilon_t|^2}{2\sigma_s^2} \right) \right.
$$
$$
\left. \cdot \left( -b\sigma_{t|s}^2\epsilon_t^2 + \frac{1}{2}a^2(u_t^T\epsilon_t)^2 \right) p(y)\,\mathrm{d}y\,\mathrm{d}x_t \right| \,\mathrm{d}x_s
$$
$$
\overset{(1)}{=} \int \left| \iint_{A_1^c \cap A_2^c \cap A_3^c} (2\pi\sigma_{t|s}^2)^{-\frac{d}{2}} \exp\left( -\frac{\left| \frac{\sigma_t^2+\sigma_{t|s}^2}{\sigma_t^2}u_t + \frac{\sigma_{t|s}^2\alpha_{t|s}}{\sigma_t^2}\epsilon_s \right|^2}{2\sigma_{t|s}^2} \right) \exp\left( -\frac{|\epsilon_s|^2}{2\sigma_s^2} \right) \right.
$$
$$
\left. \cdot \left( -b\sigma_{t|s}^2(u_t + \alpha_{t|s}\epsilon_s)^2 + \frac{1}{2}a^2(u_t^T(u_t + \alpha_{t|s}\epsilon_s))^2 \right) p(y)\,\mathrm{d}y\,\mathrm{d}x_t \right| \,\mathrm{d}x_s + \mathcal{O}(\sigma_{t|s}^4 \log^k \sigma_{t|s})
$$
$$
\overset{(2)}{=} \int \left| \iint_{A_1^c \cap A_2^c \cap A_3^c} (2\pi\sigma_{t|s}^2)^{-\frac{d}{2}} \exp\left( -\frac{\left| \frac{\sigma_t^2+\sigma_{t|s}^2}{\sigma_t^2}u_t + \frac{\sigma_{t|s}^2\alpha_{t|s}}{\sigma_t^2} \right|^2}{2\sigma_{t|s}^2} \right) \exp\left( -\frac{|\epsilon_s|^2}{2\sigma_s^2} \right) \right.
$$
$$
\left. \cdot \left( -b\sigma_{t|s}^2\alpha_{t|s}^2\epsilon_s^2 - 2b\sigma_{t|s}^3\epsilon_s u_t + \left( \frac{1}{2}a^2\alpha_{t|s}^2\epsilon_s^2 - b\sigma_{t|s}^2 \right)u_t^2 + a^2\alpha_{t|s}\epsilon_s u_t^3 + \frac{1}{2}a^2 u_t^4 \right) p(y)\,\mathrm{d}y\,\mathrm{d}x_t \right| \,\mathrm{d}x_s + \mathcal{O}(\sigma_{t|s}^4 \log^k \sigma_{t|s})
$$
$$
\overset{(3)}{\leq} \int \left| \iint (2\pi\sigma_{t|s}^2)^{-\frac{d}{2}} \exp\left( -\frac{\left| \frac{\sigma_t^2+\sigma_{t|s}^2}{\sigma_t^2}u_t + \frac{\sigma_{t|s}^2\alpha_{t|s}}{\sigma_t^2} \right|^2}{2\sigma_{t|s}^2} \right) \exp\left( -\frac{|\epsilon_s|^2}{2\sigma_s^2} \right) \right.
$$
$$
\left. \cdot \left( -b\sigma_{t|s}^2\alpha_{t|s}^2\epsilon_s^2 - 2b\sigma_{t|s}^3\epsilon_s u_t + \left( \frac{1}{2}a^2\alpha_{t|s}^2\epsilon_s^2 - b\sigma_{t|s}^2 \right)u_t^2 + a^2\alpha_{t|s}\epsilon_s u_t^3 + \frac{1}{2}a^2 u_t^4 \right) p(y)\,\mathrm{d}y\,\mathrm{d}x_t \right| \,\mathrm{d}x_s + \mathcal{O}(\sigma_{t|s}^4 \log^k \sigma_{t|s})
$$
$$(76)$$

where (1) is sorting the $\epsilon_t$ in $\exp\left( -\frac{|\epsilon_s - \frac{\alpha_{s|t}\sigma_{t|s}^2}{\sigma_t^2}\epsilon_t|^2}{2\sigma_s^2} \right)$ to outside, because the its coefficient $\frac{\alpha_{s|t}\sigma_{t|s}^2}{\sigma_t^2}$

has been second-order to $\sigma_{t|s}^2$, which sum up with outside to fourth-order. Then the total integral can be bounded by $\mathcal{O}(\sigma_{t|s}^4)$ as follows. (1) also leverages $\epsilon_t = u_t + \alpha_{t|s}\epsilon_s$. (2) is sorting the terms. (3) make the integral to the whole $\mathbb{R}^d$ of $x_t$, the rest domain probability can be bounded by $\mathcal{O}(\sigma_{t|s}^4 \log^k \sigma_{t|s})$. The first, third order of $u_t$ in the last formulation can result in $0$ due to the Gaussian kernel, and the zero, second, and fourth orders can be bounded by $\mathcal{O}(\sigma_{t|s}^4 \log^k \sigma_{t|s})$. Thus, the whole formula can be bounded.

For higher-order terms in Eq. (74), the process is similar and results in a $\mathcal{O}(\sigma_{t|s}^4 \log^k \sigma_{t|s})$ integral; we omit the details here.

For $\exp\left( au_t^T\bar{\epsilon}_t - b\sigma_{t|s}^2\bar{\epsilon}_t^2 \right)$, this case is more complex, because we cannot simply use $\epsilon_t = u_t + \alpha_{t|s}\epsilon_s$ to split $u_t$ and $x_s$. We can only use Taylor's expansion of $\bar{\epsilon}(x_t, t)$ at $\alpha_{t|s}x_s$ to reconstruct the terms into a form where all the $x_t$ are placed in $u_t$.

$$
\bar{\epsilon}_t(x_t, t) = \bar{\epsilon}(\alpha_{t|s}x_s, t) + \frac{\partial\bar{\epsilon}(\alpha_{t|s}x_s, t)^\top}{\partial x_t}u_t + \frac{\partial^2\bar{\epsilon}(\alpha_{t|s}x_s, t)}{\partial x_t^2}\|u_t\|^2
$$
$$
+ D^3\bar{\epsilon}(\alpha_{t|s}x_s, t)\|u_t\|^3 + D^4\bar{\epsilon}(\alpha_{t|s}x_s, t)\|u_t\|^4 + \mathcal{O}(\sigma_{t|s}^4 \log^k \sigma_{t|s}) \tag{77}
$$

With this expansion, we can use the same techniques as the process of $\epsilon(x_t, t)$, which involves sorting the terms as polynomials of $u_t$.

However, previous lemmas only make sure that the derivatives like $D^2\bar{\epsilon}(\alpha_{t|s}x_s, t)$ are bounded when $\alpha_{t|s}x_s$ falls with $A_1^c \cap A_2^c \cap A_3^c$, but $\alpha_{t|s}x_s$ may fall outside this region. To bound the derivatives of $\bar{\epsilon}(\alpha_{t|s}x_s, t)$, we must leverage the Whitney's extension theorem:

**Theorem B.15.** *(Brudnyi et al., 2011, Lemma 2.18) Assume the $E$ is the extension operator, $T$ is the Taylor's polynomial operator, the following is true: (a) For every $\alpha$ with $|\alpha| \leq k+1$ and all points $x \in S$ and $y \in S^c$,*

$$\left| D_y^\alpha \left( E_k^S - T_x^k \right) \vec{f}(y) \right| \leq C\omega(\|x - y\|)$$

*for some $C = C(k, n)$;*

By assign $\bar{\epsilon}(\cdot, t)$ as $\vec{f}$ in Theorem B.15, $x_t$ as $x$ in Theorem B.15, $\alpha_{t|s}x_s$ as $y$ in Theorem B.15, we reach to the conclusion that the Taylor's polynomials of $\bar{\epsilon}(\alpha_{t|s}x_s, t)$ can be bounded because we have the soundness of $\|x_t - \alpha_{t|s}x_s\|$ by the definition of $A_1^c$. Thus, the whole integral of $\bar{\epsilon}_t$ can be bounded.

### B.5 PROOF OF LEMMA 4.3

We have

$$\tilde{p}(x_s|x_t) \sim \mathcal{N}\left( \frac{\alpha_{t|s}\sigma_s^2}{\sigma_t^2}x_t + \frac{\alpha_s\sigma_{t|s}^2}{\sigma_t^2}\bar{y}, \sigma_{s|t}^2\boldsymbol{I} \right),$$

$$p_\theta(x_s|x_t) \sim \mathcal{N}\left( \frac{\alpha_{t|s}\sigma_s^2}{\sigma_t^2}x_t + \frac{\alpha_s\sigma_{t|s}^2}{\sigma_t^2}y_\theta, \sigma_{s|t}^2\boldsymbol{I} \right) \tag{78}$$

We could directly bound the KL divergence of these two transition kernels as follows:

$$\text{KL}\left( \tilde{p}(x_s|x_t)\tilde{p}(x_t), p_\theta(x_s|x_t)\tilde{p}(x_t) \right)$$

$$= \int \tilde{p}(x_t) \left[ \int \tilde{p}(x_s|x_t) \log \frac{\tilde{p}(x_s|x_t)}{p_\theta(x_s|x_t)} dx_s \right] dx_t$$

$$= \mathcal{O}(\sigma_{t|s}^2) \int \tilde{p}(x_t) |\bar{y}(x_t, t) - y_\theta(x_t, t)|^2 dx_t$$

$$\leq \mathcal{O}(\sigma_{t|s}^2) \left[ \int p(x_t)|\bar{y}(x_t, t) - y_\theta(x_t, t)|^2 dx_t + \int |\tilde{p}(x_t) - p(x_t)||\bar{y}(x_t, t) - y_\theta(x_t, t)|^2 dx_t \right]$$

$$\leq \mathcal{O}(\sigma_{t|s}^2) \left[ \epsilon_y^2 + 4C_y^2 \int |\tilde{p}(x_t) - p(x_t)|dx_t \right]$$

$$= \mathcal{O}(t - s) \left[ \epsilon_y^2 + 4C_y^2\mathcal{O}(t - s) \right] \tag{79}$$

where the first inequality is due to the triangle inequality, and the second inequality is because of Assumption 3.3, and the last equation uses Lemma B.1.

### B.6 PROOF OF PROPOSITION D.1

*proof.*

Given that the subscript is utilized for data indices, we opt to use superscripts for vector components within the context of this proof. Let $L_i$ denote the one-hot class vector of the data $y_i$.

(1) Loss $\mathcal{L}_2$. It is a constrained optimization problem:

$$\begin{cases} \underset{a_\phi}{\arg\min} \quad \mathcal{L}_2, \\ s.t. \ \ \mathbb{1}^T a_\phi = 1, \ \ a_\theta^l \geq 0, \end{cases} \tag{80}$$

where $\mathbb{1}$ is a column vector, all of whose elements are 1s. Using the KKT condition Nocedal & Wright (1999)

$$0 = \nabla_{a_\phi(x_t,t)} \mathcal{L}_2 + \nu(\mathbb{1}^T a_\phi(x_t,t) - 1) - \mu^T a_\phi(x_t,t)$$

$$= \nabla_{a_\phi(x_t,t)} \sum_i \underbrace{\frac{1}{N}(2\pi\sigma_t^2)^{-\frac{d}{2}}}_{A_t} v_i(x_t,t)|L_i - a_\phi(x_t,t)|^2 + \nu(\mathbb{1}^T a_\phi(x_t,t) - 1) - \mu^T a_\phi(x_t,t)$$

$$= \sum_i A_t v_i(x_t,t)(a_\phi(x_t,t) - L_i) + \nu\mathbb{1} - \mu$$

$$= A_t \sum_i v_i(x_t,t)a_\phi(x_t,t) - A_t \sum_i v_i(x_t,t)L_i + \nu\mathbb{1} - \mu, \tag{81}$$

which leads to

$$a_\phi^*(x_t,t) = \frac{\sum_i v_i(x_t,t)L_i - \nu\mathbb{1}/A_t + \mu/A_t}{\sum_j v_j(x_t,t)}, \tag{82}$$

where $\mu^l \geq 0, \forall c$. Because $\mathbb{1}^T a_\phi^*(x_t,t) = 1$, we have

$$\mathbb{1}^T a_\phi^*(x_t,t) = \frac{\sum_i v_i(x_t,t)\mathbb{1}L_i - \nu L/A_t + \mathbb{1}^T\mu/A_t}{\sum_j v_j(x_t,t)} = 1 - \nu L/A_t + \mathbb{1}^T\mu/A_t, \tag{83}$$

which indicates $\nu L = \mathbb{1}^T\mu \geq 0$. Since $(a_\phi^*(x_t,t))^l \mu^l = 0, \forall l$, we have

$$(\frac{\sum_i v_i(x_t,t)L_i - \mathbb{1}\nu/A_t + \mu/A_t}{\sum_j v_j(x_t,t)})^l \mu^l = 0. \tag{84}$$

If

$$\sum_i v_i(x_t,t)L_i^l - \nu/A_t + \mu^l/A_t = 0, \tag{85}$$

then

$$\nu > \mu^l \geq 0, \tag{86}$$

which lead to the contradiction

$$\nu L > \mathbb{1}^T\mu. \tag{87}$$

As as result, $\mu = 0$ and $\nu = 0$, and

$$a_\phi^*(x_t,t) = \frac{\sum_i v_i(x_t,t)L_i - \mathbb{1}\nu/A_t + \mu/A_t}{\sum_j v_j(x_t,t)} = \sum_i w_i(x_t,t)L_i = a(x_t,t). \tag{88}$$

The Lagrange multiplier $\nu$ and $\mu$ are zero, which means we can omit the constrain $\mathbb{1}^T a_\phi(x_t,t) = 1$ and $a_\phi^l \geq 0$. As the objective function and the feasible set are all convex, the KKT condition is also sufficient.

(2) The loss $\mathcal{L}_{CE}$. Using the KKT condition (Nocedal & Wright, 1999)

$$0 = \nabla_{a_\phi(x_t,t)} \mathcal{L}_{CE} + \nu(\mathbb{1}^T a_\phi(x_t,t) - 1) - \mu^T a_\phi(x_t,t)$$

$$= \nabla_{a_\phi(x_t,t)} \sum_i \underbrace{\frac{1}{N}(2\pi\sigma_t^2)^{-\frac{d}{2}}}_{A_t} v_i(x_t,t) \sum_l -L_i^l \log(a_\theta^l(x_t,t)) + \nu(\mathbb{1}^T a_\phi(x_t,t) - 1) - \mu$$

$$= -\sum_i A_t v_i(x_t,t) \begin{bmatrix} L_i^1/a_\theta^1(x_t,t) \\ \vdots \\ L_i^L/a_\theta^L(x_t,t) \end{bmatrix} + \nu\mathbb{1} - \mu, \tag{89}$$

which leads to

$$(a_\phi^*(x_t,t))^l = \sum_i \frac{A_t v_i(x_t,t)}{\nu - \mu^l} L_i^l, \tag{90}$$

where $\mu^l \geq 0$. Since $(a_\phi^*(x_t, t))^l \mu^l = 0$, we must have $\mu^l = 0, \forall l$.

Because $\mathbb{1}^T a_\phi^*(x_t, t) = 1$, we have $\nu = A_t \sum_i v_i(x_t, t)$. Thus

$$a_\phi^*(x_t, t) = \sum_i \frac{A_t v_i(x_t, t)}{A_t \sum_j v_j(x_t, t)} L_i = \sum_i w_i(x_t, t) L_i = a(x_t, t). \tag{91}$$

In this case, the Lagrange multiplier $\nu$ is not zero, thus the constrain $\mathbb{1}^T a_\phi(x_t, t) = 1$ is essential. As the objective function and the feasible set are all convex, the KKT condition is also sufficient. $\square$

## C  HIGH-ORDER TRAINING OF DECENTRALIZED DPMs

In this section, we will first derive the analytical formulation of $da(x_t, t)$, and training regularization loss to correct it in the training of $a_\phi(x_t, t)$. With a direct application of Ito's lemma of $a(x_t, t)$ with $x_t$, we have the following formulation:

$$da(x_t, t) = \left( \frac{\partial a(x_t, t)}{\partial t} + \frac{\partial a^T(x_t, t)}{\partial x_t} \left( f_t x_t - g_t^2 s_t(x_t) \right) + \frac{1}{2} g_t^2 \Delta_{x_t} a(x_t, t) \right) dt + \frac{\partial a^T(x_t, t)}{\partial x_t} g_t dB_t \tag{92}$$

To simplify the notation, we adopt the discrete finite-dataset analogy setting of Eq. (9), where we define:

$$\hat{p}^l(x_s | x_t) = \sum_{i=1}^{N_l} (2\pi\sigma_{s|t})^{-\frac{d}{2}} u_i^l(x_t, t) \exp\{-\frac{1}{2\sigma_{s|t}^2} || x_s - \frac{\alpha_{t|s}\sigma_s^2}{\sigma_t^2} x_t - \frac{\alpha_s \sigma_{t|s}^2}{\sigma_t^2} \bar{y}^l(x_t, t) ||^2\}, \tag{93}$$

$$w_i^l(x_t, t) = \frac{\exp\left\{-\frac{||x_t - \alpha_t y_i^l||^2}{2\sigma_t^2}\right\}}{\sum_{l,j} \exp\left\{-\frac{||x_t - \alpha_t y_j^l||^2}{2\sigma_t^2}\right\}}, \quad a^l(x_t, t) = \sum_{i=1}^{N_l} w_i^l(x_t, t) \tag{94}$$

$$u_i^l(x_t, t) = \frac{w_i^l(x_t, t)}{a^l(x_t, t)}, \quad \bar{y}^l(x_t, t) = \sum_{i=1}^{N_l} u_i^l(x_t, t) y_i^l. \tag{95}$$

### C.1  ANALYTICAL FORM AND REGULARIZATION OF $\frac{\partial a(x_t, t)}{\partial t}$

Define $V_i(x_t, t) = \exp\left(-\frac{(x_t - \alpha_t y_i)^2}{2\sigma_t^2}\right)$, we have:

**Lemma C.1.**

$$\frac{\partial V_i(x_t, t)}{\partial t} = V_i(x_t, t) \frac{(x_t - \alpha_t y_i) \left[ \sigma_t y_i \frac{d\alpha_t}{dt} + (x_t - \alpha_t y_i) \frac{d\sigma_t}{dt} \right]}{\sigma_t^3} \tag{96}$$

*Proof.*

$$\frac{\partial V_i(x_t, t)}{\partial t} = -V_i(x_t, t) \frac{1}{2} \frac{\partial}{\partial t} \left( \frac{(x_t - \alpha_t y_i)^2}{\sigma_t^2} \right)$$

$$= -\frac{1}{2} V_i(x_t, t) \frac{\frac{\partial}{\partial t}(x_t - \alpha_t y_i)^2 \sigma_t^2 - 2\sigma_t \dot{\sigma}_t (x_t - \alpha_t y_i)^2}{\sigma_t^4}$$

$$= -\frac{1}{2} V_i(x_t, t) \frac{2(x_t - \alpha_t y_i)(-y_i)\frac{dx_t}{dt}\sigma_t^2 - 2\sigma_t \frac{d\sigma_t}{dt}(x_t - \alpha_t y_i)^2}{\sigma_t^4} \tag{97}$$

$$= V_i(x_t, t) \frac{(x_t - \alpha_t y_i) \left[ \sigma_t y_i \frac{d\alpha_t}{dt} + (x_t - \alpha_t y_i) \frac{d\sigma_t}{dt} \right]}{\sigma_t^3}$$

$\square$

Denote $G\left(x_t, t, y_i\right) := \frac{(x_t - \alpha_t y_i)\left[\sigma_t y_i \frac{d\alpha_t}{dt} + (x_t - \alpha_t y_i)\frac{d\sigma_t}{dt}\right]}{\sigma_t^3}$, we have:

$$\frac{\partial V_i\left(x_t, t\right)}{\partial t} \equiv V_i\left(x_t, t\right) G\left(x_t, t, y_i\right) \tag{98}$$

Now we can start to derive the analytical form of $\frac{\partial a(x_t, t)}{\partial t}$: Denote $a\left(x_t, t\right) = \sum_j w_j\left(x_t, t\right) C\left(y_j\right)$, where $C\left(y_j\right)$ is the cluster label of the data point $y_j$, we have:

$$
\begin{aligned}
\frac{\partial a\left(x_t, t\right)}{\partial t} &= \sum_j C\left(y_j\right) \frac{\partial W_j\left(x_t, t\right)}{\partial t} \\
&= \sum_j C\left(y_j\right) \frac{\partial}{\partial t}\left[\frac{V_j\left(x_t, t\right)}{\sum_k V_k\left(x_t, t\right)}\right] \\
&= \sum_j C\left(y_j\right) \frac{\frac{\partial}{\partial t} V_j\left(x_t, t\right) |\sum_k V_k\left(x_t, t\right)| - V_j\left(x_t, t\right)\sum_k \frac{\partial V_k(x_t, t)}{\partial t}}{\left(\sum_k V_k\left(x_t, t\right)\right)^2} \\
&= \sum_j C\left(y_j\right) \frac{V_j\left(x_t, t\right) G\left(x_t, y_j\right)\left(\sum_k V_k\left(x_t, t\right)\right) - V_j\left(x_t, t\right)\sum_k V_k\left(x_t, t\right) G\left(x_t, t, y_k\right)}{\left(\sum_k V_k\left(x_t, t\right)\right)^2} \\
&= \sum_j \frac{V_j\left(x_t, t\right)}{\sum_k V_k\left(x_t, t\right)} C\left(y_j\right) G\left(x_t, t, y_j\right) - \sum_j \frac{V_j\left(x_t, t\right)}{\sum_k V_k\left(x_t, t\right)} C\left(y_j\right) \sum_i \frac{V_i\left(x_t, t\right)}{\sum_k V_k\left(x_t, t\right)} G\left(x_t, t, y_k\right) \\
&= \sum_j W_j\left(x_t, t\right) C\left(y_j\right) G\left(x_t, t, y_j\right) - \sum_j W_j C\left(y_j\right) \cdot \sum_i W_i\left(x_t, t\right) G\left(x_t, t, y_i\right) \\
&= \sum_j W_j\left(x_t, t\right) C\left(y_j\right) G\left(x_t, t, y_j\right) - a_\theta\left(x_t, t\right) \cdot \sum_i W_i\left(x_t, t\right) G\left(x_t, t, y_j\right) \\
&= \sum_j W_j\left(x_t, t\right) G\left(x_t, t, y_j\right)\left[C\left(y_j\right) - a_\theta\left(x_t, t\right)\right]
\end{aligned}
\tag{99}
$$

Based on this analytical form, we can define the regularization loss of $\frac{\partial a_\phi}{\partial t}$ with:

$$\mathbb{E}_{t \sim U[0,1], x_0, x_t}\left\|\frac{\partial a_\phi(x_t, t)}{\partial t} - \frac{\alpha_t}{\sigma_t^2}\left[G(x_t, t, x_0)(C(x_0) - a_\phi(x_t, t))\right]\right\| \tag{100}$$

where $G(x_t, t, x_0) = \left(\frac{x_t - \alpha_t x_0}{\sigma_t^3}\right)^\top\left[\sigma_t x_0 \frac{d\alpha_t}{dt} + (x_t - \alpha_t x_0)\frac{d\sigma_t}{dt}\right]$ is a scalar value.

## C.2 ANALYTICAL FORM AND REGULARIZATION OF $\frac{\partial a^T\left(x_t, t\right)}{\partial x_t}$

For the matrix-value $\frac{\partial a(x_t, t)}{\partial x_t}$, we first derive its analytical form as followings:

$$
\begin{aligned}
\frac{\partial a(x_t, t)}{\partial x_t} &= \frac{\partial \sum_i W_i\left(x_t, t\right) C\left(y_i\right)}{\partial x_t} \\
&= \sum_i C\left(y_i\right) \frac{\partial \sum_i W_i\left(x_t, t\right)}{\partial x_t} \\
&= \sum_i \sum_j C\left(y_i\right) \frac{V_i\left(x_t, t\right) V_j\left(x_t, t\right)}{\left(\sum_j V_j\left(x_t, t\right)\right)^2}\left|-\frac{x_t - \alpha_t y_i}{\sigma_t^2} + \frac{x_t - \alpha_t y_j}{\sigma_t^2}\right| \\
&= \sum_i \sum_j C\left(y_i\right) \cdot \frac{V_i\left(x_t, t\right) V_j\left(x_t, t\right)}{\left(\sum_j V_j\left(x_t, t\right)\right)^2}\left[y_i - y_j\right] \cdot \frac{\alpha_t}{\sigma_t^2} \\
&= \left(\sum_i W_i\left(x_t, t\right) \cdot C\left(y_i\right) y_i - \sum_i w_i\left(x_t, t\right) C\left(y_i\right) \bar{y}\right) \frac{\alpha_t}{\sigma_t^2}
\end{aligned}
\tag{101}
$$

Based on this analytical form, we can define the regularization loss of $\frac{\partial a_\phi}{\partial x_t}$ with:

$$\mathbb{E}_{t\sim U[0,1],x_0,x_t}\left\|\frac{\partial a_\phi^\top(x_t,t)}{\partial x_t}(f_tx_t-g_t^2s(x_t))-\frac{\alpha_t}{\sigma_t^2}C(x_0)\left[x_0-y_\theta(x_t,t)\right]^\top(f_tx_t-g_t^2s(x_t))\right\| \tag{102}$$

Here we directly match the result of $(f_tx_t-g_t^2s(x_t))$ linear transformed by $\frac{\partial a_\phi^\top(x_t,t)}{\partial x_t}$.

### C.3 Analytical form and regularization of $\Delta_{x_t}a(x_t,t)$

Remember that we defined

$$V_i(x_t,t)=\exp\left(-\frac{(x_t-\alpha_ty_i)^2}{2\sigma_t^2}\right),\quad W_j(x_t,t)=\frac{V_j(x_t,t)}{\sum_k V_k(x_t,t)} \tag{103}$$

First, we have the following form of the derivative of $V_j(x_t,t)$ with respect to $x_t$:

$$\frac{\partial V_j}{\partial x_t}=-\frac{(x_t-\alpha_ty_j)}{\sigma_t^2}V_j \tag{104}$$

And we define $S=\sum_k V_k(x_t,t)$, its derivative with respect to $x_t$ is:

$$\frac{\partial S}{\partial x_t}=\sum_k\frac{\partial V_k}{\partial x_t}=-\frac{1}{\sigma_t^2}\sum_k(x_t-\alpha_ty_k)V_k \tag{105}$$

Then the derivative of $W_j(x_t,t)$ with respect to $x_t$:

$$\begin{aligned}\frac{\partial W_j}{\partial x_t}&=\frac{\frac{\partial V_j}{\partial x_t}S-V_j\frac{\partial S}{\partial x_t}}{S^2}\\&=-\frac{(x_t-\alpha_ty_j)}{\sigma_t^2}W_j+\frac{V_j}{\sigma_t^2S^2}\sum_k(x_t-\alpha_ty_k)V_k\end{aligned} \tag{106}$$

Then we have the Laplace of $W_j(x_t,t)$ with respect to $x_t$:

$$\begin{aligned}\Delta_{x_t}W_j&=\frac{\partial^2 W_j}{\partial x_t^2}\\&=\frac{\partial}{\partial x_t}\left(\frac{\partial W_j}{\partial x_t}\right)\\&=\frac{\|x_t-\alpha_ty_j\|^2-\sigma_t^2}{\sigma_t^4}W_j-\frac{W_j}{\sigma_t^4}\sum_k\frac{\|x_t-\alpha_ty_k\|^2-\sigma_t^2}{\sum_k V_k}V_k+\frac{2W_j}{\sigma_t^4}\left\|\sum_k(x_t-\alpha_ty_k)V_k\right\|^2\frac{1}{S^2}\\&=\frac{\|x_t-\alpha_ty_j\|^2}{\sigma_t^4}W_j-\frac{W_j}{\sigma_t^4}\sum_k\frac{\|x_t-\alpha_ty_k\|^2}{\sum_k V_k}V_k+\frac{2W_j}{\sigma_t^4}\left\|\sum_k(x_t-\alpha_ty_k)V_k\right\|^2\frac{1}{S^2}\\&=W_j\frac{1}{\sigma_t^4}\left\{\|x_t-\alpha_ty_j\|^2-\|x_t\|^2+2\alpha_tx_t^\top\bar{y}-T(x_t,t)+\|x_t-\alpha_t\bar{y}\|^2\right\}\\&=W_j\frac{1}{\sigma_t^4}\left\{\alpha_t^2\|\bar{y}-y_j\|^2-\|x_t\|^2+2\alpha_tx_t^\top\bar{y}-T(x_t,t)\right\}\end{aligned} \tag{107}$$

where $T(x_t,t)=\sum_k W_k\|y_k\|^2$, and can be learned via the trace matching technique. Based on this analytical form, we can define the regularization loss of $\Delta_{x_t}a_\phi(x_t,t)$ with:

$$\mathbb{E}_{t\sim U[0,1],x_0,x_t}\left\|\Delta_{x_t}a_\phi(x_t,t)-\frac{1}{\sigma_t^4}C(x_0)\left[\alpha_t^2\|y_\theta-x_0\|^2-\|x_t\|^2+2\alpha_tx_t^\top y_\theta-T(x_t,t)\right]\right\| \tag{108}$$

# D  FURTHER EXPERIMENTS FOR DECENTRALIZED DPMS

We develop the decentralized DPMs, which are concurrent to the work of (McAllister et al., 2025). We initially call this technique MoG (Mixture of Gaussians). In this section, we will introduce our training and experimental findings on MoG.

## D.1  TRAINING

The decentralized representation of the reversed transition in Eq. (8) consists of combination coefficients $a^l(x_t, t)$, referred to as the *class part*, and cluster-specific kernels $\hat{p}^l(x_s|x_t)$, referred to as the *diffusion part*. For the diffusion part, to learn $L$ cluster-specific kernels, we propose training a single conditional network $y_\theta(x_t, t, l)$ to represent them, rather than training $L$ independent networks, in order to save computational overhead. For the class part, we train the network $a_\phi(x_t, t)$ to represent it.

MoG-DPM learns the class part in a manner similar to the diffusion part. Define $L_i$ as the one-hot vector representing the class to which data point $y_i$ belongs. Then we construct $a(x_t, t)$ as:

$$a(x_t, t) = \sum_i w_i(x_t, t) L_i, \tag{109}$$

where $w_i(x_t, t)$ represents the coefficients in the ground truth transition kernel in Eq. (4)). Substituting Eq. (109) into Eq. (20) aligns with the ground truth transition kernel in Eq. (4). Given that the structure of $a(x_t, t)$ closely mirrors that of $\bar{y}(x_t, t)$, a neural network $a_\phi(x_t, t)$ can be trained in a manner analogous to the $x$-prediction networks in diffusion models. The training process to learn the class part can be formulated as:

**Proposition D.1.** *Let $L(x_0)$ denote the one-hot class vector of $x_0$, the optimal $a_\phi(x_t, t)$ for the two objective functions*

$$\mathcal{L}_2 = \mathbb{E}_{x_0, x_t \sim p(x_0, x_t), t \sim \mathbf{U}(0,1)} ||L(x_0) - a_\phi(x_t, t)||^2, \tag{110}$$

*and*

$$\mathcal{L}_{CE} = \mathbb{E}_{x_0, x_t \sim p(x_0, x_t), t \sim \mathbf{U}(0,1)} CE(L(x_0), a_\phi(x_t, t)), \tag{111}$$

*are the same and equal to $a(x_t, t)$, where CE represents the cross-entropy loss.*

Based on the analysis above, we present the algorithms for training the diffusion model $y_\theta(x_t, t, l)$ in Algorithm 1 and the label model $a_\phi(x_t, t)$ in Algorithm 2.

---

**Algorithm 1** Training process of diffusion model $y_\theta$

1: **Repeat**
2: $x_0 \sim p(x, t)|_{t=0}$
3: $t \sim \text{Uniform}(t_1, t_2, ..., t_T)$
4: $\epsilon \sim \mathcal{N}(\mathbf{0}, \boldsymbol{I})$
5: Take gradient descent step on
   $\nabla_\theta \|y_\theta(x_t, t, c(x_0)) - x_0\|^2$
6: **Until** converged

---

**Algorithm 2** Training process of label model $a_\theta$

1: **Repeat**
2: $x_0 \sim p(x, t)|_{t=0}$
3: $t \sim \text{Uniform}(t_1, t_2, ..., t_T)$
4: $\epsilon \sim \mathcal{N}(\mathbf{0}, \boldsymbol{I})$
5: Take gradient descent step on
   $\nabla_\phi \|a_\phi(x_t, t) - L(x_0)\|^2$ or
   $\nabla_\phi CE(a_\phi(x_t, t), L(x_0))$
6: **Until** converged

---

## D.2  SAMPLING

Different from the single Gaussian approximation, our method samples two random variables in each step: weight sampling in line 3 and diffusion sampling in line 5 as shown in Algorithm 3. Naive merging method only samples a class label at the beginning of sampling process as shown in line 2 in Algorithm 4.

, the MoG approximation samples two random variables in each step. These two samplings are referred to as weight sampling and diffusion sampling, respectively. The *two-stage sampling method* is encapsulated in Algorithm 3.

**Algorithm 3** Sampling process for MoG approximation

1: $x_{t_T} \sim \mathcal{N}(\mathbf{0}, \mathbf{I})$
2: **for** $i = T, ..., 1$ **do**
3:   $l \sim a_\theta(x_{t_i}, t_i)$
4:   $z \sim \mathcal{N}(\mathbf{0}, \mathbf{I})$
5:   $x_{t_{i-1}} = \frac{\alpha_{t_i|t_{i-1}}\sigma_{t_{i-1}}^2}{\sigma_{t_i}^2}x_{t_i}$
    $+\frac{\alpha_{t_{i-1}}\sigma_{t_i|t_{i-1}}^2}{\sigma_{t_i}^2}y_\theta(x_{t_i}, t_i, l) + \sigma_{t_{i-1}|t_i}z$
6: **end for**
7: **return** $x_{t_0}$

**Algorithm 4** Sampling process for fixed class approximation

1: $x_{t_T} \sim \mathcal{N}(\mathbf{0}, \mathbf{I})$
2: $l \sim (b^1, b^2, \cdots, b^L)$
3: **for** $i = T, ..., 1$ **do**
4:   $z \sim \mathcal{N}(\mathbf{0}, \mathbf{I})$
5:   $x_{t_{i-1}} = \frac{\alpha_{t_i|t_{i-1}}\sigma_{t_{i-1}}^2}{\sigma_{t_i}^2}x_{t_i}$
    $+\frac{\alpha_{t_{i-1}}\sigma_{t_i|t_{i-1}}^2}{\sigma_{t_i}^2}y_\theta(x_{t_i}, t_i, l) + \sigma_{t_{i-1}|t_i}z$
6: **end for**
7: **return** $x_{t_0}$

As the probability flow ODE keeps the single-time marginals (Song et al., 2020b), we can replace the diffusion sampling method by probability flow ODE based methods, such as DDIM (Song et al., 2020a), DPM (Lu et al., 2022b), PNDM (Liu et al., 2022) etc. We summarize this in Algorithm 5, where $ODE(x_t, t, c, z)$ represents the ODE based sampling methods.

**Algorithm 5** Sampling process for MoG approximation with ODE based methods

1: $x_{t_T} \sim \mathcal{N}(\mathbf{0}, \mathbf{I})$
2: **for** $i = T, ..., 1$ **do**
3:   $l \sim a_\theta(x_{t_i}, t_i)$
4:   $z \sim \mathcal{N}(\mathbf{0}, \mathbf{I})$
5:   $x_{t_{i-1}} = ODE(x_{t_i}, t_i, l, z)$
6: **end for**
7: **return** $x_{t_0}$

**Algorithm 6** Sampling process for fixed class approximation with ODE based methods

1: $x_{t_T} \sim \mathcal{N}(\mathbf{0}, \mathbf{I})$
2: $l \sim (b^1, b^2, \cdots, b^L)$
3: **for** $i = T, ..., 1$ **do**
4:   $z \sim \mathcal{N}(\mathbf{0}, \mathbf{I})$
5:   $x_{t_{i-1}} = ODE(x_{t_i}, t_i, l, z)$
6: **end for**
7: **return** $x_{t_0}$

Theoretically, there exists instances where the MoG kernel outperforms the single Gaussian kernel measured with KL-divergence. We present a case where: (1) The centers of the MoGs are orthogonal, i.e., $(\bar{y}^l(x_t, t))^T \bar{y}^m(x_t, t)$ for all $l, m \in 0, 1, ..., L-1$, and (2) The norms of the centers are sufficiently large: $|\bar{y}^l(x_t, t)| > \frac{\log a^l(x_t, t)}{C(a^l(x_t, t)-1)}$, where $C = \frac{\alpha_s^2 \sigma_{t|s}^2}{2\sigma_s^2 \sigma_t^2}$. Then $\text{KL}(p(x_s|x_t)|| \sum_l a^l(x_t, t)\tilde{p}^l(x_s|x_t)) \leq \text{KL}(p(x_s|x_t)||\tilde{p}(x_s|x_t))$.

We further present experimental evidence to demonstrate that training with $L > 1$ yields better results than the single Gaussian kernel case ($L = 1$).

### D.3 EXPERIMENTS

In this section, we demonstrate experimental evidence for the correctness of our proposed non-trivial merging method on a 2D synthetic toy dataset and the CIFAR-10 image dataset for SDE-based samplers. We then extend our validation by transitioning the ground truth classification labels to unsupervised clustering labels, validating the correctness of our method in a more general scenario. Finally, we provide experimental results on ODE-based samplers. We discovered that MoG-DPM outperforms DDPM and exhibits performance on par with SN-DDPM, GMS, and the naive merging method.

#### D.3.1 RESULTS ON 2D TOY DATASET

We validate our approach on five synthetic 2D datasets with varying distributions. Each dataset consists of continuous 2D points $(x, y) \in \mathbb{R}^2$, assigned class labels based on natural clustering. For each experiment, we generated 4K samples and assessed generation quality using Maximum Mean Discrepancy (MMD) with a Laplace kernel (bandwidth 0.1) Gretton et al. (2012). Each computation was repeated 8 times, and we report the average MMD value, with lower values indicating better generation quality. The evaluation results are shown in Table 3.

| #STEPS | Patterns (Number of Classes) | | | | | | | | | | | | | | |
|---|---|---|---|---|---|---|---|---|---|---|---|---|---|---|---|
| | Circles (2) | | | Moons (2) | | | Pinwheel (5) | | | CheckerBoard (8) | | | Gaussians (8) | | |
| | Uncond | Naive | MoG-DPM | Uncond | Naive | MoG-DPM | Uncond | Naive | MoG-DPM | Uncond | Naive | MoG-DPM | Uncond | Naive | MoG-DPM |
| 500 | 2.788 | 8.017 | **0.8717** | 3.935 | 7.051 | **2.590** | 3.672 | 6.658 | **3.056** | -1.301 | 6.561 | **-2.912** | 6.339 | 7.945 | **2.567** |
| 100 | 7.685 | 5.066 | **2.088** | 5.724 | 4.111 | **0.7585** | 3.891 | 9.002 | **3.789** | 0.1639 | 7.913 | **-4.804** | 6.364 | 11.80 | **5.768** |
| 50 | 12.42 | 4.764 | **2.594** | 13.21 | 5.457 | **0.08196** | 14.05 | 14.79 | **3.265** | 9.516 | 8.759 | **0.8225** | 18.93 | 13.66 | **7.536** |
| 30 | 20.82 | **10.86** | 15.55 | 21.38 | 8.021 | **3.508** | 24.37 | 17.64 | **8.619** | 33.57 | 12.18 | **7.559** | 45.70 | 8.005 | **1.572** |
| 20 | 50.96 | **18.90** | 40.57 | 40.36 | 10.44 | **10.12** | 52.51 | 20.48 | **9.817** | 115.5 | 19.88 | **10.22** | 90.04 | 6.098 | **3.801** |
| 10 | 121.0 | **33.44** | 71.94 | 149.1 | 43.29 | 44.30 | 169.3 | 21.07 | **12.32** | 494.9 | 54.94 | 90.71 | 211.1 | 22.03 | **16.62** |

Table 3: **Evaluation Results on Synthetic Datasets.** Generation quality is assessed by Maximum Mean Discrepancy (MMD) $\downarrow$. Values in the table have been rescaled by a factor of $10^{-5}$. Both the merging methods generates samples with better quality under varying data patterns and denoising timesteps.

### D.3.2 Results on Image Dataset

We evaluate our method with SDE-based samplers including DDPM (Ho et al., 2020), SN-DDPM (Bao et al., 2022) and GMS (Guo et al., 2024) respectively on CIFAR-10 dataset by generating 50k samples as per common practice and use the widely recognized Fréchet Inception Distance (FID) score (Heusel et al., 2017) as evaluation metrics.

We compare our non-trival merging method with unconditional generation and naive merging strategy, which only execute label sampling at the beginning of the sampling process. The evaluation results are presented in Tab. 4. The two merging approaches both demonstrate better generalization quality than the raw unconditional generation method across various diffusion sampling timesteps for all the three types of SDE samplers.

Although employing L2 loss and cross-entropy constraints to regularize the training of the label diffusion model $a_\theta$ are both theoretically correct as proved in Proposition D.1, we find employing cross-entropy loss leads to better performance for MoG-DPM than L2 loss in practice. This is because cross-entropy loss directly optimizes for probabilistic correctness and provides stronger gradients when the predicted probabilities are far from the true labels, which helps with the convergence of $a_\theta$.

| Schedule | Linear | | | | | | Cosine | | | | | |
|---|---|---|---|---|---|---|---|---|---|---|---|---|
| # Timesteps | 10 | 25 | 50 | 100 | 200 | 1000 | 10 | 25 | 50 | 100 | 200 | 1000 |
| DDPM (Uncond) | 43.14 | 21.63 | 15.21 | 10.94 | 8.23 | 5.11 | 34.76 | 16.18 | 11.11 | 8.38 | 6.66 | 4.92 |
| + Naive | 37.58 | 21.05 | 15.12 | 10.67 | 7.82 | **4.50** | 26.90 | 14.97 | 9.94 | 7.02 | 5.33 | **3.76** |
| + MoG-DPM | **37.50** | **20.72** | **13.81** | **9.38** | **6.66** | **4.50** | **24.28** | **13.69** | 9.44 | 7.02 | 5.82 | 4.69 |
| SN-DDPM (Uncond) | 21.87 | 6.91 | 4.58 | 3.74 | 3.34 | 3.71 | 16.33 | 6.05 | 4.19 | 3.83 | 3.72 | 4.08 |
| + Naive | 11.90 | **4.98** | **3.62** | **2.98** | **2.55** | **2.93** | 9.92 | 4.95 | 3.35 | 2.67 | 2.53 | 2.74 |
| + MoG-DPM | **11.66** | 6.39 | 4.29 | 3.40 | 2.97 | 3.30 | 10.77 | 5.87 | 4.08 | 3.63 | 3.35 | 3.51 |
| GMS (Uncond) | 17.43 | 5.96 | 4.16 | 3.26 | 3.01 | **2.76** | 13.80 | 5.48 | 4.00 | 3.46 | 3.34 | 4.23 |
| + Naive | **10.40** | **4.84** | **3.61** | **3.00** | **3.00** | 2.86 | **8.76** | 4.91 | 3.43 | 2.76 | 2.60 | 3.35 |
| + MoG-DPM | 14.54 | 5.89 | 3.82 | 3.11 | 3.01 | 2.82 | 10.80 | 6.22 | 4.53 | 3.64 | 3.34 | 4.35 |

Table 4: **Evaluation of FID $\downarrow$ on CIFAR-10 Image Dataset**. Both naive merging method (Naive) and MoG-DPM merging method showcase superior generalization quality than uncondition generation (Uncond) with various SDE-based schedulers.

### D.3.3 Unsupervised Clustering Based on K-Means

The merging method is not restricted to learn data distribution which has ground truth classification labels. As our convergence proof is independent from data division methods, our non-trivial merging method is theoritically correct with arbitrary data partition of the training data.

To experimentally validate its correctness, we have further expanded the ground truth classification labels provided by the dataset to unsupervised labels generated based on K-means clustering algorithm. We conduct comparative experiments of unconditional generation on the CelebA-HQ-256 image dataset. Initially, we compute the VAE latent vector for each image (Kingma & Welling, 2013), extract the primary dimension through principal component analysis (PCA) (Abdi & Williams, 2010), and subsequently cluster the images into 10 classes using K-Means algorithm. Experimental results in Table 5 validate the correctness of our proposed method under such scenerio.

Different data partition may lead to varying convergence speed of the training of the label model $a_\theta$. We can examine the speed of convergence for different partitions from the viewpoint of the Stochastic Gradient Descent (SGD) method. As per Theorem 5.3 in Garrigos & Gower (2023), the error bound of SGD aligns with two terms. Given the learning rate $\gamma_t$ and the initial state $x^0$, the first term is represented as $\frac{||x^0 - x^*||^2}{\sum_t \gamma_t}$, where $x^*$ is the optimal point. The second term is $2\sigma_f^* \frac{\sum_t \gamma_t^2}{\sum_t \gamma_t}$, where $\sigma_f^*$ is $\mathbb{E}||\nabla f_i(x^*) - \mathbb{E}\nabla f_i(x^*)||^2$. In our context, the choice of partition can be used to reduce $\sigma_f^*$. More specifically, when the data points $y_i^l$s of a single label are closely clustered, $\sigma_f^*$ will be small, which in turn leads to a faster convergence speed. Intuitively, the speed of convergence is quicker when each component of the partition is more compact.

However, the optimization method we employ (AdamW) is more intricate than SGD, and the speed of convergence for different partitions is not straightforward to analyze. We therefore remain a deeper discussion on this topic as future work.

| Method | # Denoising Steps | | |
| --- | --- | --- | --- |
| | 10 | 25 | 50 |
| Uncond | 35.21 | 18.60 | 14.16 |
| Naive | 30.67 | 15.73 | **12.16** |
| MoG-DPM | **30.58** | **15.46** | 12.25 |

Table 5: **FID ↓ on CelebA-HQ-256 Image Dataset.** The merging method is applicable to latent diffusion models with labels generated in an unsupervised manner, achieving better performance compared with uncondition DDPM generation.

| Method | # Denoising Steps | | | |
| --- | --- | --- | --- | --- |
| | 10 | 25 | 35 | 50 |
| DDIM | 21.31 | 10.70 | 9.12 | 7.74 |
| + Naive | **16.54** | **9.15** | 8.63 | **6.60** |
| + MoG-DPM | 19.44 | 10.19 | **8.44** | 7.33 |
| DPM Solver | 7.95 | 6.54 | 6.17 | 3.37 |
| + Naive | 5.78 | 3.49 | 3.28 | **2.90** |
| + MoG-DPM | **5.45** | **3.27** | **3.19** | 3.36 |
| DPM Solver++ | 11.11 | 4.78 | 3.95 | 3.42 |
| + Naive | 10.94 | 4.29 | 3.80 | **2.99** |
| + MoG-DPM | **10.37** | **3.69** | **3.39** | 3.34 |
| Method | 18 (# NFE=35) | | | |
| EDM | 2.00 | | | |
| + Naive | **1.78** | | | |
| + MoG-DPM | 1.95 | | | |

Table 6: **FID ↓ on CIFAR-10 with ODE-Based Samplers**.

### D.3.4 EXTENSION TO ODE-BASED METHODS

The merging method can be further extended from stochastic sampling methods to ODE-based deterministic methods. We showcase our experimental results on in Table 6, suggest that our method can indeed operate effectively with deterministic sampling methods including DDIM, DPM Solver (Lu et al., 2022b), DPM Solver++ (Lu et al., 2022c) and EDM (Karras et al., 2022).

### D.3.5 QUALITATIVE RESULTS

We provide qualitative examples of our methods' generated images in Figure 3, 4, 5, 6 and 6.

## E DISCUSSIONS

### E.1 CONVERGENCE WITH KOLMOGOROV EQUATIONS

The seminal work of (Song et al., 2020b) found that the forward process in Eq. (1) can be seen as the discretization of the following stochastic differential equation (SDE)

$$\mathrm{d}x_t = f_t x_t \, \mathrm{d}t + g_t \, \mathrm{d}B_t, \tag{112}$$

where $f(t) = \frac{\mathrm{d}\log \alpha_t}{\mathrm{d}t}$, $g^2(t) = \frac{\mathrm{d}\sigma_t^2}{\mathrm{d}t} - 2\frac{\mathrm{d}\log\alpha_t}{\mathrm{d}t}\sigma_t^2$ and $B_t$ is the standard Brownian motion. According to (Anderson, 1982), its reverse process is

$$\mathrm{d}x_t = (f_t x_t - g_t^2 \nabla_{x_t} \log p(x_t)) \, \mathrm{d}t + g_t \, \mathrm{d}\tilde{B}_t. \tag{113}$$

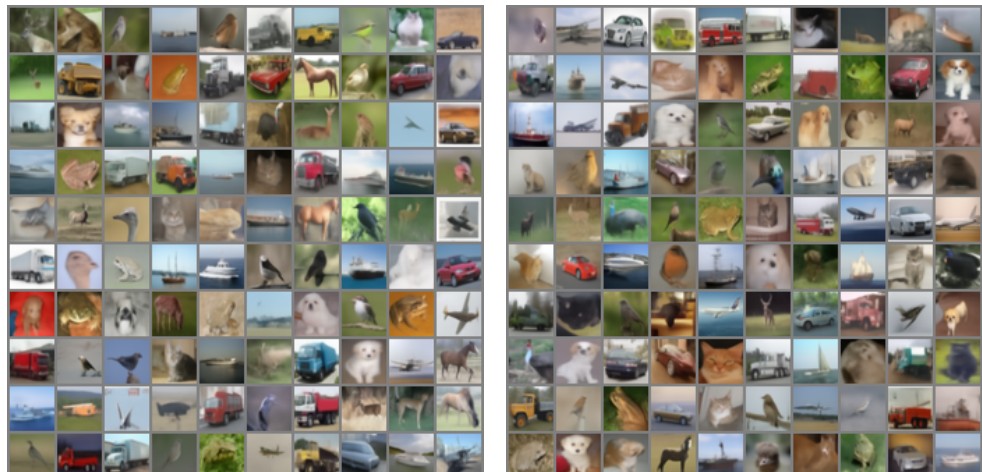

Figure 3: **DDPM + MoG-DPM on 10 Denoising Steps.**

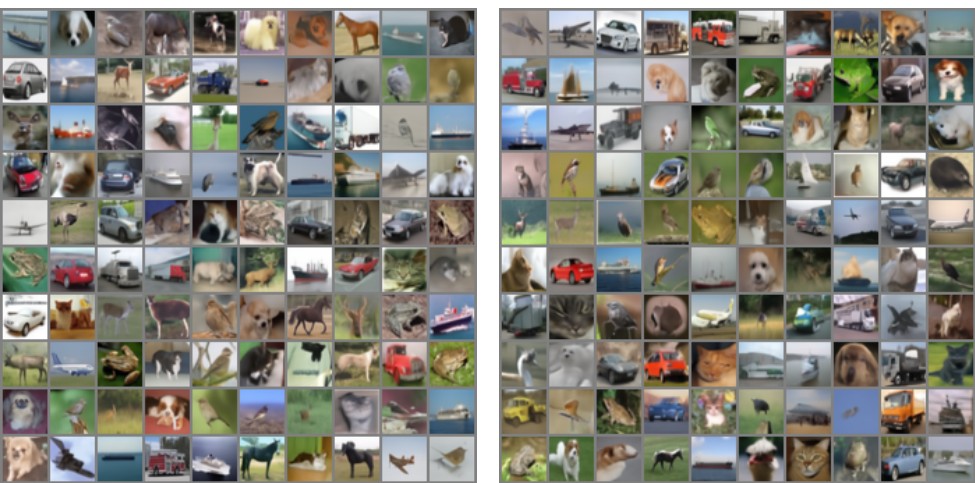

Figure 4: **DDPM + MoG-DPM on 25 Denoising Steps.**

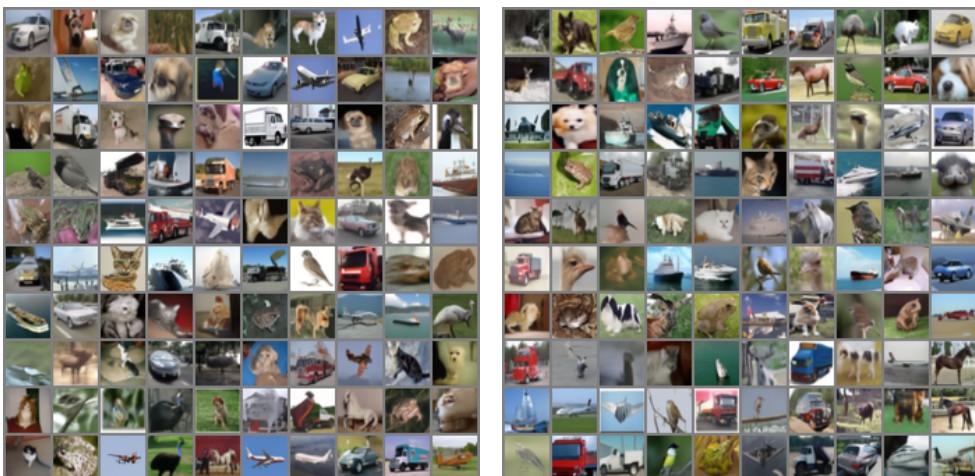

Figure 5: **DDPM + MoG-DPM on 50 Denoising Steps.**

Previous efforts to prove the convergence of DPM heavily depend on the Kolmogorov equations of Eq. (113) For instance, (Lee et al., 2022) defines the discretization approximation

$$\mathrm{d}x_t = (f_{t-}x_{t-} - g_{t-}^2 \nabla_{x_t} \log p(x_{t-})) \, \mathrm{d}t + g_t \, \mathrm{d}\tilde{B}_t, \tag{114}$$

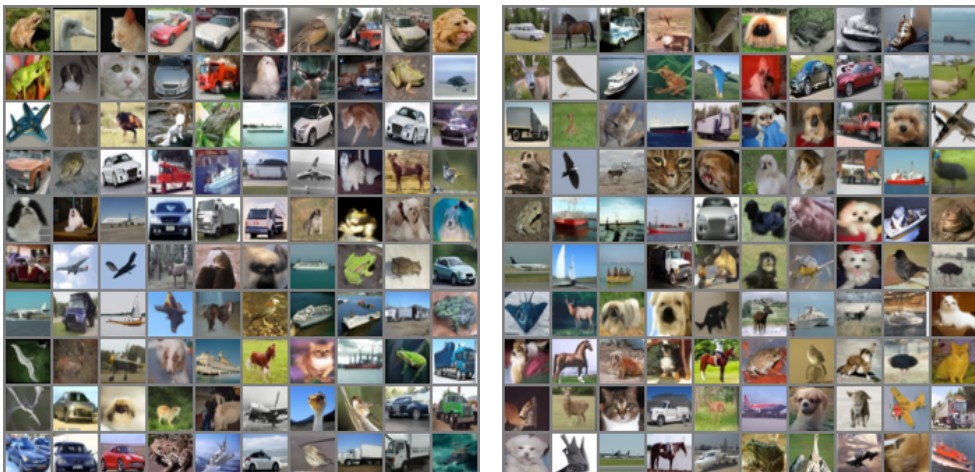

Figure 6: **DDPM + MoG-DPM on 100 Denoising Steps.**

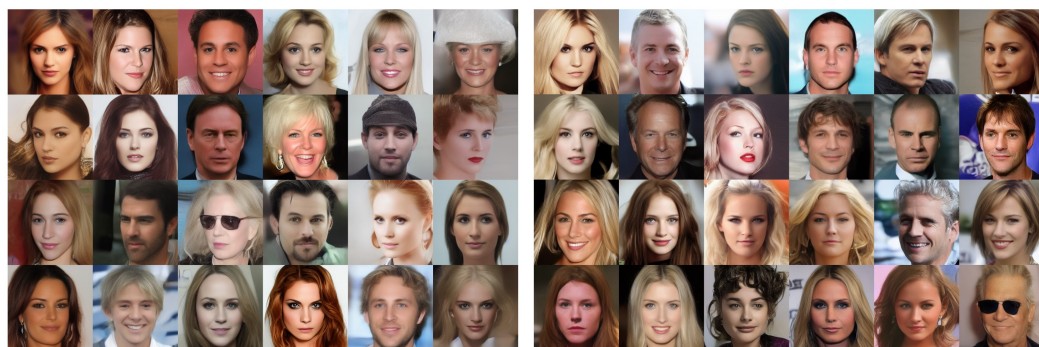

Figure 7: **Qualitative Results with ODE Samplers for Latent Diffusin Models.** Oue proposed merging method is applicable to ODE-based schedulers like DDIM with latent-space diffusion models. Left group of images are generated with 10 denoising timesteps while the right with 25 denoising timesteps

and establish the corresponding Kolmogorov forward equation for the single time marginal distribution of Eq. (114), denoted as $q(x_t)$. Here $t_-$ means the left endpoint of the stochastic integral. Ultimately, the Chi-square divergence $\chi^2(q(x_t)\|p(x_t))$ is estimated using the Kolmogorov equations. Numerous subsequent studies have embraced this configuration (Lee et al., 2023; Chen et al., 2022; 2023b;a). However, this proof has its limitations as it's based on the Kolmogorov equations. This means it cannot be applied to other types of discretizations or modified diffusion processes where constructing the Kolmogorov equations is challenging. Therefore, a proof that can be readily adapted to a wider range would be beneficial.

### E.2 THE NECESSITY OF THE PSEUDO-NON-MARKOVIAN METHOD

**Why SDE-type Methods cannot Be Extended to Decentralized DPMs** Taking Chen et al. (2022) as an example, the authors first state Girsanov's theorem in Theorem 8 and then use it to analytically express the stepwise discretization KL error in the integral form shown in Eq. (5.5). They further bound the integral length by $\Delta t$ and the integrand by $\Delta t$ in Eq. (5.6). With some extra derivation, Chen et al. (2022) can deduce the following KL stepwise discretization error between $p_{s|t}(x_s|x_t)$ and $\hat{p}(x_s|x_t)$:

$$\mathbb{E}_{p_t} KL(p_{s|t}(x_s|x_t), \hat{p}(x_s|x_t)) = \mathcal{O}(\Delta t^2).$$

If we follow Chen et al. (2022) and attempt to convert the stepwise error to TV distance, we must apply Pinsker's inequality, which gives:

$$\mathbb{E}_{p_t} TV(p_{s|t}(x_s|x_t), \hat{p}(x_s|x_t)) = \mathcal{O}(\Delta t).$$

However, this order is insufficient: a stepwise TV discretization error of order $\Delta t$ cannot guarantee global convergence. Summing across all steps yields a global TV discretization error of order $\mathcal{O}(1)$. This bound cannot vanish as $\Delta t \to 0$, making it an unconvergent bound.

The core gap in applying previous SDE-type analyses to decentralized DPMs is that they inherently yield only $\mathcal{O}(\Delta t)$ stepwise TV error. This gap persists not only in Chen et al. (2022) but also in the SDE-type analyses of Lee et al. (2022; 2023); Benton et al. (2024); Chen et al. (2023a), because all of these works first bound the stepwise discretization error in KL and then convert it to TV using Pinsker's inequality. This conversion inevitably makes the TV bound too loosely.

This is precisely the gap addressed by our method: instead of relying on KL, we directly use the pseudo-non-Markov technique to analyze the stepwise discretization error in TV distance and derive a sharper $\mathcal{O}(\Delta t^2)$ TV stepwise error. Thus, our pseudo-non-Markovian method is necessary to obtain the SOTA convergence result for decentralized DPMs.

**Why Auxiliary-Chain–Type Methods cannot Be Extended to Decentralized DPMs**    In auxiliary-chain–type analyses such as Li & Yan (2024), the framework does not involve bounding the stepwise TV term:

$$\mathbb{E}_{p_t} TV(p_{s|t}(x_s|x_t), \hat{p}(x_s|x_t))$$

, and thus their methodology cannot support our augment.

More specifically, auxiliary-chain–type methods derive an error bound between the true state $X_t$ and a constructed intermediate auxiliary state $\bar{Y}_t$. However, the definition of this auxiliary state $\bar{Y}_t$ relies on historical information of the sampling trajectory and therefore cannot be determined in a purely stepwise manner. Moreover, how to define such an auxiliary state in the setting of decentralized DPM sampling remains an open and non-trivial problem.

In conclusion, such auxiliary-chain–type methods cannot be used to characterize the pure stepwise TV error, and consequently cannot be directly extended to the convergence analysis of decentralized DPMs.

