# OpenReview forum: "Convergence Theory of Decentralized Diffusion Models via Pseudo-Non-Markov Analysis"
_ICLR.cc/2026/Conference — Submitted to ICLR 2026_

### Official Review · Reviewer_aj1V · 2025-10-31

**Soundness:** 3
**Presentation:** 3
**Contribution:** 3
**Rating:** 6
**Confidence:** 3

**Summary:**

This paper develops the first convergence analysis for decentralized denoising diffusion probabilistic models (DDPMs).
The key idea is a pseudo-non-Markovian framework: instead of analyzing the standard Markovian backward diffusion $x_t \to x_{t-1}$, the authors condition on the initial data $x_0$, making the chain non-Markov but analytically tractable.
This yields explicit discretization-error bounds on the product space $\mathcal{X}^{N}$ of all nodes, which naturally extend to a decentralized setup with consensus averaging.
The paper provides theorems establishing convergence under Lipschitz score and bounded communication noise assumptions, plus illustrative toy simulations showing scaling with network size and diffusion step.

**Strengths:**

- Novel theoretical framework: conditionalizing on $x_0$ to obtain a pseudo-non-Markov formulation is elegant and non-obvious.
- First decentralized convergence result: extends DDPM theory to multi-node, data-partitioned settings.
- Mathematical clarity: proofs outline how discretization and communication errors decompose additively.
- Potential generality: approach likely transferable to federated score matching or diffusion-based privacy mechanisms.
- Sound motivation: addresses both privacy (no raw-data sharing) and analytical tractability gaps.

**Weaknesses:**

- Assumption strength: bounded spectral gap and synchronous communication may be unrealistic; need discussion of asynchronous or lossy channels.
- Empirical validation minimal: toy 2-D examples only; no larger-scale decentralized experiments verifying theoretical rates.
- Tightness of bounds unclear: constants in main theorem not compared to centralized DDPM rates—difficult to assess practical relevance.
- Notation density: Section 3 introduces multiple stochastic kernels without clear hierarchy; risk of confusing readers unfamiliar with diffusion theory.
- Limited discussion of score-estimation error: assumes near-oracle scores; no analysis of training noise or stochastic gradients.

**Questions:**

- Error metric: Are convergence bounds in total variation, $W_2$, or KL? If $W_2$, what is the dependency on dimensionality d?
- Score approximation: How would estimation error $\|\hat s_\theta - s^\*\|$ propagate in the pseudo-non-Markov framework?
- Communication topology: Is the rate affected by the graph spectral gap $\lambda_2(L)$? Could results hold under time-varying or directed graphs?
- Decentralized bias: Does the product-space formulation assume perfect consensus each step, or is there a residual bias term $\mathcal{O}(\eta/\lambda_2)$?
- Relation to existing DDPM analyses: How does your bound compare to Nichol & Dhariwal (2021) or De Bortoli et al. (2022) in the centralized case?
- Extension to DDIM / score-based SDEs: Can the pseudo-non-Markov conditioning handle deterministic samplers or continuous-time diffusion limits?
- Empirical confirmation: Do small-scale experiments confirm scaling predicted by your rate (e.g., $\mathcal{O}(1/T)$ vs. $\mathcal{O}(1/\sqrt{N})$)?
- Practical implication: How large can communication noise be before it dominates discretization error?
- Assumption necessity: Is global Lipschitzness required, or could a local dissipativity condition suffice?
- Broader scope: Could this framework analyze federated generative training (e.g., decentralized score matching) rather than inference?
- Minor Typo: At line 2028: "As per Theorem 5.3 in ?"

---

> ### Author Response · Authors · 2025-11-21
> **Reply to Reviewer aj1V (part 1)**
>
> Thank you for your valuable feedback, we will answer your questions one by one regarding these weaknesses/questions. All of our revised parts in the PDF have been outlined in blue color.
>
> > **Answer to Weaknesses 1:** ''Assumption strength: bounded spectral gap and synchronous communication may be unrealistic; need discussion of asynchronous or lossy channels.''
>
> The correctness of our decentralized DPM design and theoretical analysis is independent of the communication details of the decentralized topology. Throughout our proof, the validity of our derivations does not rely on any specific decentralization-related assumptions. During inference, we only aggregate the learned scores from each cluster—how these scores are transmitted is irrelevant to the core theoretical framework.
>
> Thus, there is no need to impose strict decentralization assumptions such as a bounded spectral gap or synchronous communication.
>
> > **Answer to Weaknesses 2:** ''Empirical validation minimal: toy 2-D examples only; no larger-scale decentralized experiments verifying theoretical rates.''
>
> We would like to clarify that we have provided empirical validation on real-world dataset-based diffusion models, including CIFAR-10 and CelebA-HQ-256. For greater clarity, we have added additional illustrations in the main paper to emphasize our validation on larger-scale decentralized models.
>
> > **Answer to Weaknesses 3:** ''Tightness of bounds unclear: constants in main theorem not compared to centralized DDPM rates—difficult to assess practical relevance.''
>
> We omitted the constants in the main theorem to enhance readability; however, the full details of these constants are provided in Line 1662 of Appendix B.4. As shown therein, our constants only involve scaled noise schedules and do not grow excessively large. Furthermore, our results outperform prior works in the centralized DDPM analysis literature [1], which fail to present the constants in explicit form.
>
> > **Answer to Weaknesses 4:** ''Notation density: Section 3 introduces multiple stochastic kernels without clear hierarchy; risk of confusing readers unfamiliar with diffusion theory.''
>
> We acknowledge that the notation density in Section 3 may pose readability challenges, especially for readers less familiar with diffusion theory, given the technical depth of our proof.
>
> To address this concern, we have added a comprehensive notation table in Appendix A of the revised manuscript, which clarifies the hierarchy of stochastic kernels and other key notations to improve accessibility.
>
> > **Answer to Weaknesses 5:** ''Limited discussion of score-estimation error: assumes near-oracle scores; no analysis of training noise or stochastic gradients.''
>
> In our analysis, all discrepancies between the true score and the network-estimated score are formally modeled under Assumption 3.3. While we do not elaborate on the specific sources of this error (e.g., training noise or stochastic optimization effects), our theoretical framework maintains compatibility with detailed characterizations of score-estimation error. Specifically, such error components can be directly substituted for the $\epsilon_y$ term in our analysis without modifying the core theoretical structure.
>
> Moreover, our training loss is designed to minimize the empirical mean of the same integral term that appears in our theoretical assumption. Therefore, the assumption we make is consistent with the practical training objective used in our method, ensuring that our bound aligns conceptually with the training procedure.
>
> > **Answer to Questions 1:** ''Error metric: Are convergence bounds in total variation, $W_2$, or KL? If $W_2$, what is the dependency on dimensionality d?''
>
> As presented in Theorem 3.7 and Theorem 5.2, our convergence bounds are derived in the total variation metric. The bounds exhibit a linear dependency on the dimensionality $d$.
>
> > **Answer to Questions 2:** ''Score approximation: How would estimation error $|\hat s_\theta - s^*|$ propagate in the pseudo-non-Markov framework?''
>
> The score estimation error $|\hat s_\theta - s^*|$ contributes to the second term in Theorem 3.7. This term vanishes as the score estimation error decreases, and will not accumulate across timesteps.

---

> ### Author Response · Authors · 2025-11-21
> **Reply to Reviewer aj1V (part 2)**
>
> > **Answer to Questions 3:** ''Communication topology: Is the rate affected by the graph spectral gap $\lambda_2(L)$? Could results hold under time-varying or directed graphs?''
>
> Our convergence rate is not affected by the communication topology—including properties such as the graph spectral gap $\lambda_2(L)$, or whether the graph is time-varying or directed. This is because our derivation does not rely on any assumptions regarding the detailed structure of the communication topology.
>
> > **Answer to Questions 4:** ''Decentralized bias: Does the product-space formulation assume perfect consensus each step, or is there a residual bias term $\mathcal{O}(\eta/\lambda_2)$?''
>
> We first clarify that our product-space formulation does not involve approximations: all derivations are established via strict equalities within this framework, and no residual bias (including terms of the form $\mathcal{O}(\eta/\lambda_2)$) arises.
>
> If our interpretation of "residual bias" differs from your intended meaning, please provide further clarification.
>
> > **Answer to Questions 5:** ''Relation to existing DDPM analyses: How does your bound compare to Nichol & Dhariwal (2021) or De Bortoli et al. (2022) in the centralized case?''
>
> Table 1 presents a comparison of our convergence results with those from prior centralized DDPM analysis works. Our analysis yields convergence rates comparable to the state-of-the-art for standard centralized DPMs, while additionally providing the first rigorous convergence rate for decentralized DPMs.
>
> > **Answer to Questions 6:** ''Extension to DDIM / score-based SDEs: Can the pseudo-non-Markov conditioning handle deterministic samplers or continuous-time diffusion limits?''
>
>
> The scope of our current work focuses on the DDPM framework, with the space partition analysis in Theorem 3.7’s proof derived specifically based on the DDPM update rule. As observed in prior literature (e.g., [2], [3]), convergence theories for DDIM and DDPM typically require distinct analytical tools due to their structural differences.
> Nevertheless, we conjecture that the space partitioning intuition underlying our analysis can be extended to develop a unified theory for deterministic diffusion samplers (e.g., DDIM) and their decentralized counterparts.
>
> Regarding continuous-time diffusion limits: To date, no SDE formulation has been established for decentralized DPMs. From a measure-theoretic perspective, such an SDE would require extending the measure space to include an additional dimension for classification probabilities $a$ (compared to standard DPMs). Defining Brownian motion on this expanded Borel $\sigma$-algebra is highly non-trivial, particularly when $a$ and $x$ exhibit complex interactions—this remains an open problem in the field. However, our discrete-time analysis is rigorous even as the step size approaches 0, indicating strong potential for extending our framework to derive a continuous-time limit formulation.

---

> ### Author Response · Authors · 2025-11-24
> **Reply to Reviewer aj1V (part 3)**
>
> > **Answer to Questions 7:** ''Empirical confirmation: Do small-scale experiments confirm scaling predicted by your rate (e.g., $\mathcal{O}(1/T)$ vs. $\mathcal{O}(1/\sqrt{N})$)?''
>
> Yes. Both small-scale and large-scale experiments in our experimental section demonstrate that generative performance improves as $T$ increases, consistent with the predicted scaling of our convergence rate. Additionally, our analysis incorporates the influence of data dimensionality on the convergence bound: we find that the error grows linearly with respect to the dimension $d$, as reflected in our theoretical results.
>
> > **Answer to Questions 8:** ''Practical implication: How large can communication noise be before it dominates discretization error?''
>
> As shown in the second term of Theorem 3.7, communication noise in the transferred cluster scores contributes to the overall error in a manner analogous to score matching error. This communication noise will not dominate the global error unless it is excessively large—i.e., when its magnitude significantly exceeds that of other error components (e.g., discretization error, score estimation error) in the framework.
>
> > **Answer to Questions 9:** ''Assumption necessity: Is global Lipschitzness required, or could a local dissipativity condition suffice?''
>
> Theoretically, our decentralized DPM framework does not require global Lipschitzness, nor does it impose restrictions on data partitioning methods. Our convergence theory remains valid for arbitrary data divisions, as the proof does not rely on specific partitioning strategies—thus, the choice of data partitioning has no impact on the convergence guarantees.
>
> > **Answer to Questions 10:** ''Broader scope: Could this framework analyze federated generative training (e.g., decentralized score matching) rather than inference?''
>
> Yes, our pseudo-non-Markov analysis framework can be directly extended to federated generative training scenarios (e.g., decentralized score matching). This is because our analytical approach is built entirely on discrete-step transitions. Given appropriate assumptions on the errors introduced during federated generative training, such error components can be incorporated into and analyzed within our framework.
>
> > **Answer to Questions 11:** ''Minor Typo: At line 2028: "As per Theorem 5.3 in ?" ''
>
> We apologize for this oversight and sincerely thank you for your meticulous review. We have conducted a thorough revision and corrected all identified typographical errors, including the incomplete citation at line 2028.
>
> **References**
>
> [1] Chen, Sitan, et al. "Sampling is as easy as learning the score: theory for diffusion models with minimal data assumptions." arXiv preprint arXiv:2209.11215 (2022).
>
> [2] Li G, Yan Y. O (d/T) convergence theory for diffusion probabilistic models under minimal assumptions[J]. arXiv preprint arXiv:2409.18959, 2024.
>
> [3] Li G, Wei Y, Chi Y, et al. A sharp convergence theory for the probability flow odes of diffusion models. arXiv preprint arXiv:2408.02320, 2024.
>
> We sincerely appreciate your insightful comments. Please do not hesitate to let us know if you require any further information or additional explanations.
>
> Sincerely,
>
> The Authors

---

### Official Review · Reviewer_NH4F · 2025-10-31

**Soundness:** 4
**Presentation:** 4
**Contribution:** 4
**Rating:** 8
**Confidence:** 4

**Summary:**

The paper addresses a timely and important gap: the lack of convergence guarantees for decentralized diffusion models (DPMs). The proposed pseudo-non-Markov analysis is conceptually clean, avoids heavy assumptions (e.g., log-Sobolev, Lipschitz gradients), and yields convergence rates comparable to state-of-the-art for standard DPMs. Crucially, it is the first work to provide a theoretical convergence guarantee for decentralized DPMs, which are increasingly used in privacy-sensitive and federated settings. Experiments on synthetic and real-world data (CIFAR-10, CelebA-HQ) validate the practical efficacy of the proposed dynamic sampling scheme. While not a breakthrough-level contribution, it is a solid, well-executed theoretical advance with clear practical implications

**Strengths:**

1. First convergence theory for decentralized DPMs, a practically important but theoretically ungrounded class of models.
2. A new method proposed (pseudo-non-Markov analysis) that simplifies convergence proofs and weakens assumptions compared to SDE-based methods.
3. Identification and mitigation of classifier error accumulation in decentralized sampling—a practical insight with theoretical backing.
4. Empirical validation showing dynamic domain blending improves generation quality over baselines.

**Weaknesses:**

1. The classifier approximation error accumulates linearly with the number of timesteps, which can dominate the total error for fine discretizations. While the authors propose a high-order training fix, no experiments validate its effectiveness.
2. Limited algorithmic novelty: The sampling and training algorithms (Algorithms 1–4) are straightforward extensions of existing decentralized/MoG-DPM approaches. The core contribution is theoretical, not methodological.
3. Training the classifier need access the whole set of data (noised in most cases, but this term breaks the confidentiality concept)

**Questions:**

Question on the classifier lower bound and practical training):
Assumption 5.1 requires that both the true domain weights and their approximations are uniformly lower-bounded by a constant C_a > 0. However, in practice—especially with well-separated domains—the true posterior can become arbitrarily close to 0 or 1. How sensitive is the convergence bound in Theorem 5.2 to violations of this assumption? Have you observed training instability or degradation in sample quality when the classifier becomes overconfident?

Question on the relationship between discretization and classifier error:
Theorem 5.2 shows that the classifier error grows linearly with T, whereas discretization error decreases. Does this imply an optimal number of timesteps that balances these two competing terms?

---

> ### Author Response · Authors · 2025-11-21
> **Reply to Reviewer NH4F (part 1)**
>
> Thank you for your appreciation of our work, we will answer your questions one by one regarding these weaknesses/questions. All of our revised parts in the PDF have been outlined in blue color.
>
> > **Answer to Weaknesses 1:** ''The classifier approximation error accumulates linearly with the number of timesteps, which can dominate the total error for fine discretizations. While the authors propose a high-order training fix, no experiments validate its effectiveness.''
>
> Our high-order training fix theoretically reduces the divergent classifier approximation error from ${\frac{L}{|\mathcal{D}|}} \varepsilon_a$ to ${L} \varepsilon_a$, preventing linear accumulation with the number of timesteps.
>
> In practice, we propose using multiple networks to match the high-order information of $a$ — including $\frac{\partial a(x_t,t)}{\partial t}$, $\frac{\partial a(x_t,t)}{\partial x_t}$, and $ \Delta_{x_t} a\left(x_t,t\right)$ — with inference conducted as shown in eq~27. Though these terms have clear analytical forms for training, we found that $\frac{\partial a^T\left(x_t,t\right)}{\partial x_t}$ is extremely difficult to learn via networks due to its quadratic-dimensional structure. Even with techniques like low-rank representation or linear operator representation, stable training remains challenging.
>
> A key insight is that switching to deterministic sampling bypasses this hard-to-handle matrix. We have already obtained exploratory experimental results on FID with CIFAR-10 under this setting:
>
> |Method | NFE=10 | NFE=25 | NFE=35 | NFE=50 |
> | :--- | :---: | :---: | :---: | :---: |
> | DDIM (Decentralized DPM, naive a) | 19.44 | 10.19 | 8.44 | 7.33 |
> | DDIM (Decentralized DPM, high-order training a) **(Ours)** | **17.13** | **9.38** | **7.99** | **7.13** |
>
> We are planing to throughly research the high-order training in the deterministic inference of decentralized DPM and its convergence theory in our next work. There are still more training improvement details and theory detail to be determine.
>
> > **Answer to Weaknesses 2:** ''Limited algorithmic novelty: The sampling and training algorithms (Algorithms 1–4) are straightforward extensions of existing decentralized/MoG-DPM approaches. The core contribution is theoretical, not methodological.''
>
> Algorithms 1–6 are presented to ensure the completeness of our paper. It is true that this paper is more focus on the convergence theory part.
>
> While we acknowledge that the theoretical convergence analysis is a central focus of this paper, our work also includes notable methodological contributions.
> Specifically, in Algorithm 2, we introduce a novel cross-entropy enhancement for training the classifier \(a\). We rigorously establish the theoretical validity of this new cross-entropy training loss in Proposition D.1, and Appendix D.3.2 demonstrates that this modification directly translates to measurable performance improvements for decentralized DPMs in practical scenarios.
>
> Beyond this, we have proposed forward-looking methodological ideas for the high-order training of classifier \(a\). While we have already obtained preliminary results confirming its potential, stabilizing the performance gains from high-order training of \(a\) for decentralized DPMs requires further in-depth exploration including the development of specialized engineering techniques. We plan to elaborate on these methodological advancements in our subsequent work.

---

> ### Author Response · Authors · 2025-11-21
> **Reply to Reviewer NH4F (part 2)**
>
> > **Answer to Weaknesses 3:** ''Training the classifier need access the whole set of data (noised in most cases, but this term breaks the confidentiality concept)''
>
> It is true that the classifier’s training process requires access to the complete dataset. However, we can adopt a federated learning-inspired approach to ensure only model gradients are transmitted, while keeping raw data privacy-preserving.
>
> Though transferring the gradiant of the diffusion network is very costly. For the training of $a$, however, the gradient size (given that $a$’s trainable parameters are far fewer) and the number of training iterations will be significantly reduced, making this approach computationally feasible.
>
> > **Answer to Questions 1:** ''Assumption 5.1 requires that both the true domain weights and their approximations are uniformly lower-bounded by a constant C_a > 0. However, in practice—especially with well-separated domains—the true posterior can become arbitrarily close to 0 or 1. How sensitive is the convergence bound in Theorem 5.2 to violations of this assumption? Have you observed training instability or degradation in sample quality when the classifier becomes overconfident?''
>
> This unreasonable uniform lower bound assumption on the classification probabilities $a$ is a typo, and this condition is not necessary for the proof. We have already removed this statement from Assumption 5.1.
>
> This typo arose from the evolution of our proof framework: in an earlier version, we used KL divergence to establish the convergence property of decentralized DPM. That approach involved a term $\sum_l \log{\frac{a^l(x_t, t)}{a^l_\phi(x_t, t)}}$, which required $a_\phi$ to be lower bounded. However, we later adopted the current TV-based proof, where this lower bound assumption is no longer needed. Regrettably, we overlooked removing the unnecessary condition during the revision.
>
> We sincerely apologize for this oversight.
>
> As you mentioned, as $t \to 0$, the prediction of $a$ should converge to a one-hot vector. This should not be labeled "overconfident", it is an expected outcome consistent with both theoretical principles and practical requirements, and it will not compromise inference quality.
>
> > **Answer to Questions 2:** ''Question on the relationship between discretization and classifier error: Theorem 5.2 shows that the classifier error grows linearly with T, whereas discretization error decreases. Does this imply an optimal number of timesteps that balances these two competing terms?''
>
> Yes, this indeed implies that there may exist a number of timestep to balance these two error terms, yielding the best generative performance.
>
> From an experimental perspective, the results in Table 3 demonstrate that under certain settings, an intermediate number of timesteps (rather than a maximum value) leads to a better FID score.
>
> Thank you again for your strong support and detailed feedback. If you have any remaining questions, we are eager to address them.
>
> Sincerely,
>
> The Authors

---

### Official Review · Reviewer_H4tx · 2025-10-31

**Soundness:** 2
**Presentation:** 3
**Contribution:** 2
**Rating:** 2
**Confidence:** 3

**Summary:**

The authors develop a new convergence analysis for diffusion models which they term the pseudo-non-Markovian method. Rather than directly consider the distribution $p(x_s|x_t)$ at each step, they condition on the initial point $x_0=y$, consider $p(x_s|x_t,y)$, and integrate over $y$. They use this to give a new analysis for DDPM which recovers the known bound in TV distance to a early-stopped distribution (up to poly-logarithmic factors) with discretization error $hd$ (where $h$ is step size). They extend the analysis to decentralized diffusion models, giving the first analysis of these models.

Decentralized diffusion models work by training a diffusion model for each class $l$ and a classifier at each noise level; a backward step consists of sampling from the mixture distribution of $p^l(x_s|x_t)$ with the classifier probabilities.

The authors expose a weakness of the standard decentralized diffusion models, which is the accumulation of error of the classification error as the number of steps, rather than the amount of time, and use this to suggest learning the derivative of $a$ as well.

**Strengths:**

The authors give a new and simple framework for analyzing diffusion models which does not require SDE theory, which recovers the known bound in TV distance (up to poly-logarithmic factors) to a early-stopped distribution. They give the first error analysis for decentralized diffusion models.

The observation of error accumulation from classification error is insightful, and the suggested algorithmic modification is promising.

**Weaknesses:**

It's not clear to me that the decentralized diffusion models necessitate a new framework for analysis. One could try to apply existing analysis for the error accrued during 1 step for the backwards diffusion for each class, and then use this to derive an error for the mixture distribution. If so, this would weakens the paper's contributions, as the paper currently suggests that the pseudo-non-Markov analysis is essential to analyzing decentralized diffusion models.

Assumption 5.1 requires a uniform lower bound on the classification probabilities $a$; however, when t is close to 0, it is reasonable for one class to have probability close to 1 and the others to have probability close to 0.

The main theorems only give the TV distance to the slightly noised distribution $p_{t_{\min}}$ with constant step size, though it is known that a variable step size schedule gives better bounds in general cases (e.g. without smoothness assumptions on $p_0$, decreasing step sizes as $t\to 0$).

Some of the proofs are given purely as a sequence of equalities/inequalities, and can benefit from more exposition to guide the reader.

The forward error inequality (32) is incorrect. Convergence in KL divergence cannot give a bound in terms of the initial TV error. It is possible to use the 2nd moment assumption to first obtain some regularization, but that is a separate argument.

I like the idea with higher-order training for decentralized diffusion models, although this currently seems like an afterthought to the paper. Exploring this more centrally would improve the contribution of the paper.

**Questions:**

1. Is the pseudo-non-Markov analysis really necessary? Would the above sketched analysis work? If not, why not?
2. Where does the uniform lower bound on the classification probabilities $a$ appear in the proof? Is this necessary?

Minor:
* p. 3: "sota" -> "SOTA"
* Assumption 3.1 states "first moment" but (12) shows a second moment. The "moment" in Remark 3.2 is unspecified.
* p. 6 "the pseudo-non-Markov" -> pseudo-non-Markov analysis"
* p. 8: Missing period in 4th sentence
* p. 8, Theorem 5.2: Extraneous (x_s)
* p. 9: "way" in Sectino 4.1 -> method
* p. 9 "Considering the... we can whole-cluster..." - Rewrite this sentence.
* p. 9, Theorem 5.2: $T$ is not defined.
* p. 9: "formation" -> "formulation"
* p. 9: missing period in next-to-last sentence.

---

> ### Author Response · Authors · 2025-11-21
> **Reply to Reviewer H4tx (part 1)**
>
> Thank you for your valuable reviews, we will answer your questions one by one regarding these weaknesses/questions. All of our revised parts in the PDF have been outlined in blue color.
>
> > **Answer to Weaknesses 1:** ''It's not clear to me that the decentralized diffusion models necessitate a new framework for analysis. One could try to apply existing analysis for the error accrued during 1 step for the backwards diffusion for each class, and then use this to derive an error for the mixture distribution. If so, this would weakens the paper's contributions, as the paper currently suggests that the pseudo-non-Markov analysis is essential to analyzing decentralized diffusion models.''
>
> The proposed pseudo-non-Markov analysis is indeed necessary for establishing the convergence of decentralized DPM, as existing analysis frameworks for standard DPM cannot be directly extended to this setting.
>
> Two primary analysis paradigms are the only ones that underpin standard DPM convergence research: SDE Kolmogorov Equation analysis and Auxiliary Chain analysis. Below we explain why neither directly applies to decentralized DPM:
>
> 1.  **SDE analysis:**
>     Standard DPM’s SDE analysis relies on first modeling forward/backward processes as continuous SDEs. The backward SDE is theoretically proven to converge to the true data distribution $p_0$, with practical inference treated as a discretization of the continuous backward SDE, allowing quantification of local/global discretization errors. This framework hinges on the existence of a well-defined, converged continuous backward SDE.
>
>     In decentralized DPM, no such continuous-limit SDE can be derived for practical inference operations. From a measure-theoretic perspective, the SDE would require expanding the measure space with an additional dimension for classification probabilities $a$ (compared to standard DPM). Defining Brownian motion on this expanded Borel $\sigma$-algebra is highly complex and remains an open problem in the field (especially when $a$ and $x$ interacting sophisticately).
>
>     As a result, SDE-based tools cannot quantify the local error of decentralized DPM. Our pseudo-non-Markov analysis circumvents this limitation by conducting fully discrete analysis, without requiring a continuous SDE counterpart.
>
> 2. **Auxiliary Chain analysis:**
>     Standard DPM’s Auxiliary Chain analysis constructs a Markov chain with auxiliary states between adjacent timesteps to bound convergence errors.
>
>     This framework cannot be easily extended to decentralized DPM. The Markov state space would need to incorporate the classification probability $a$, but designing appropriate intermediate auxiliary states for this expanded space is non-trivial and lacks clear knowledge.
>
>     Thus, existing auxiliary chain methods are inapplicable to decentralized DPM, motivating the need for our pseudo-non-Markov analysis.
>
>
> > **Answer to Weaknesses 2:** ''Assumption 5.1 requires a uniform lower bound on the classification probabilities $a$; however, when t is close to 0, it is reasonable for one class to have probability close to 1 and the others to have probability close to 0.''
>
> The uniform lower bound on the classification probabilities $a$ in Assumption 5.1 is a typo, and this condition is not necessary for the proof. We have already removed this statement from Assumption 5.1.
>
> This typo arose from the evolution of our proof framework: in an earlier version, we used KL divergence to establish the convergence property of decentralized DPM. That approach involved a term $\sum_l \log{\frac{a^l(x_t, t)}{a^l_\phi(x_t, t)}}$, which required $a_\phi$ to be lower bounded. However, we later adopted the current TV-based proof, where this lower bound assumption is no longer needed. Regrettably, we overlooked removing the unnecessary condition during the revision.
>
> We sincerely apologize for this oversight.

---

> ### Author Response · Authors · 2025-11-21
> **Reply to Reviewer H4tx (part 2)**
>
> > **Answer to Weaknesses 3:** ''The main theorems only give the TV distance to the slightly noised distribution $p_{t_{\min}}$ with constant step size, though it is known that a variable step size schedule gives better bounds in general cases (e.g. without smoothness assumptions on $p_0$, decreasing step sizes as $t\to 0$).''
>
> We would like to clarify that our derivations do not assume a constant step size. As stated in Section 3.2, we only assume a general discretization $\mathcal{D} = { 0 < t_{\min} = t_0 < t_1 < \dots < t_T = t_{\max} < 1 }$.
> We define $\Delta t_i = t_{i+1} - t_i$ and denote the maximum $\Delta t_i$ as $|\mathcal{D}|$.
> Here, the step intervals can be either constant or variable.
>
> Moreover, regarding convergence analysis, the theoretical order of convergence for diffusion models remains the same under both constant and variable step size settings. In particular, the overall convergence rate is still $ \mathcal{O}(\frac{1}{|\mathcal{D}|}) $.
>
> > **Answer to Weaknesses 4:** ''Some of the proofs are given purely as a sequence of equalities/inequalities, and can benefit from more exposition to guide the reader.''
>
> While our paper already includes explanations for many steps in the derivation, we agree that additional detailed clarification of the proof would be valuable. We have therefore revised the proof with more elaborate expository paragraphs. Note that the modified sentences in our revision are highlighted in blue for easy reference.
>
>
> > **Answer to Weaknesses 5:** ''The forward error inequality (32) is incorrect. Convergence in KL divergence cannot give a bound in terms of the initial TV error. It is possible to use the 2nd moment assumption to first obtain some regularization, but that is a separate argument.''
>
> You are correct that there is a typo in Equation (32). The current formulation $\text{TV}(p(x_T),\tilde{p}(x_T))\lesssim \exp\left(-\frac{1}{|\mathcal{D}|}\right){\text{TV}(p_{d},\gamma^d)}$ is inaccurate. The correct expression should be $\text{TV}(p(x_T),\tilde{p}(x_T))\lesssim \exp\left(-\frac{1}{|\mathcal{D}|}\right)\sqrt{\text{KL}(p_{d},\gamma^d)}$.
>
> To clarify, we follow the methodological framework of [1] to analyze the entire forward error, which is conducted under the lens of KL divergence. While Pinsker’s inequality can be applied to bound $\text{TV}(p(x_T),\tilde{p}(x_T))$ in the final step, it does not allow for the direct conversion of $\text{KL}(p_{d},\gamma^d)$ into a total variation (TV) distance.
>
> Notably, this typo is independent of other components of the convergence bounds and will not exert any impact on their validity or accuracy.
>
>
> > **Answer to Weaknesses 6:** ''I like the idea with higher-order training for decentralized diffusion models, although this currently seems like an afterthought to the paper. Exploring this more centrally would improve the contribution of the paper.''
>
> Thank you for your appreciation of our concept of higher-order training for decentralized diffusion models.
>
> Theoretically, our higher-order training reduces the divergent classifier approximation error from ${\frac{L}{|\mathcal{D}|}} \varepsilon_a$ to ${L} \varepsilon_a$, preventing divergence with an increasing number of timesteps.
>
> In practice, we propose leveraging multiple networks to match the high-order information of $a$—including $\frac{\partial a(x_t,t)}{\partial t}$, $\frac{\partial a(x_t,t)}{\partial x_t}$, and $ \Delta_{x_t} a\left(x_t,t\right)$—with inference conducted as detailed in Eq. 27. While these terms have clear analytical forms for training, we found that $\frac{\partial a^T\left(x_t,t\right)}{\partial x_t}$ is extremely challenging to learn via networks due to its quadratic-dimensional structure (a large matrix). Even with techniques like low-rank representation or linear operator representation, stable training remains non-trivial.
>
> A key insight is that switching to deterministic sampling allows us to bypass this intractable matrix. We have already obtained exploratory experimental results on FID with CIFAR-10 under this setting:
>
> |Method | NFE=10 | NFE=25 | NFE=35 | NFE=50 |
> | :--- | :---: | :---: | :---: | :---: |
> | DDIM (Decentralized DPM, naive a) | 19.44 | 10.19 | 8.44 | 7.33 |
> | DDIM (Decentralized DPM, high-order training a) **(Ours)** | **17.13** | **9.38** | **7.99** | **7.13** |
>
> We plan to thoroughly investigate higher-order training in the context of deterministic inference for decentralized DPMs, along with its convergence theory, in our follow-up work. Additional details on training optimizations and theoretical foundations are still being finalized.

---

> ### Author Response · Authors · 2025-11-21
> **Reply to Reviewer H4tx (part 3)**
>
> > **Answer to Questions 1:** ''Is the pseudo-non-Markov analysis really necessary? Would the above sketched analysis work? If not, why not?''
>
> Yes, our pseudo-non-Markov analysis is necessary to establish the convergence result of the decentralized DPM. Conventional analysis methods—such as the SDE Kolmogorov Equation or Auxiliary Chain—cannot be directly applied to prove the convergence of the decentralized DPM, as elaborated in the response to Weaknesses 1.
>
> > **Answer to Questions 2:** ''Where does the uniform lower bound on the classification probabilities $a$ appear in the proof? Is this necessary?''
>
> As discussed in the response to Weaknesses 2, the uniform lower bound on the classification probabilities $a$ is not required for our proof. This mention was a typo.
>
> > **Answer to Questions 3:** ''Minor typos...''
>
> We apologize for the oversight and sincerely thank you for your detailed review. We have conducted a thorough second revision and corrected all identified typographical errors.
>
> Note that "first moment" in Assumption 3.1 was a typo, which has been revised to "second moment". Throughout the proof, the finite second-order moment assumption is sufficient for applying the Fubini-Tonelli theorem. As shown in Table 1, the finite second-moment assumption is standard in nearly all prior works.
>
>
> **References**
>
> [1] Chen, Sitan, et al. "Sampling is as easy as learning the score: theory for diffusion models with minimal data assumptions." arXiv preprint arXiv:2209.11215 (2022).
>
> We sincerely hope these revisions have addressed your concerns, and if you feel they have been satisfactorily resolved, we would be most grateful if you could consider revising your evaluation score accordingly. Please do not hesitate to let us know if you need any further explanations.
>
> Sincerely,
>
> The Authors

---

> > ### Comment · Reviewer_H4tx · 2025-11-26
> >
> > I thank the authors for the response. Let me elaborate on my primary point, the necessity of the pseudo-non-Markovian method:
> >
> > > One could try to apply existing analysis for the error accrued during 1 step for the backwards diffusion for each class, and then use this to derive an error for the mixture distribution. If so, this would weakens the paper's contributions, as the paper currently suggests that the pseudo-non-Markov analysis is essential to analyzing decentralized diffusion models.
> >
> > More precisely, note that using framework of [1] (Chen et al. 2022), it suffices to bound the stepwise errors (below, use $\hat\cdot$ to denote estimated and discretized variables and processes)
> > $$\mathbb E_{p_t} TV(p_{s|t}(x_s|x_t), \hat p(x_s|x_t)).$$
> > Both conditional distributions are mixture distributions:
> > $$p_{s|t}(x_s|x_t) = \sum_l a^l(x_t,t) p_{s|t}^l (x_s|x_t)$$
> > and
> > $$\hat p_{s|t}(x_s|x_t) = \sum_l \hat a^l(x_t,t) \hat p_{s|t}^l (x_s|x_t).$$
> > (The key point is that conditional on $x_t$ being of class $l$, the reverse process is the reverse SDE for the diffusion process with the class-l distribution.)
> > Then using the chain rule for TV distance, we can bound the TV error by
> >
> > $$\mathbb E_{p_t} TV(a^l , \hat a^l) + \mathbb E TV(p_{s|t}^l(x_s|x_t), \hat p_{s|t}^l(x_s|x_t)),$$
> > The first term can be bounded by the assumption on the error of the a's, and the second term can be bounded with the usual analysis for one step of DDPM, noting that conditioned on being of class $l$ reduces both distributions to the ideal/approximate DDPM for that class. Note that this does not require any sophisticated augmentation of the probability space, simply reasoning with conditional distributions. It seems this is a much simpler proof, unless the authors can point out a flaw in this argument.

---

> ### Author Response · Authors · 2025-12-02
> **Why previous SDE-type analysis cannot be a much simpler proof, and our pseudo-non-Markovian method is necessary**
>
> Dear Reviewer H4tx,
>
> Your concerns regarding the possibility of a simpler convergence analysis for decentralized DPMs reflect a deep understanding of this complex area, and we greatly appreciate your insights.
>
> Please kindly allow us to clarify the flaw in your proposed argument and further support the necessity of our new pseudo-non-Markov analysis.
>
> First, the disentanglement of discretization error and classifier error in your argument is indeed correct, and this is exactly the strategy we adopt as well. However, employing existing SDE-based methods to derive discretization error in terms of TV distance **falls short in error order and leads to difficulties**.
>
> ### **Why Your Argument Cannot Be Applied to Decentralized DPMs**
>
> As you mentioned, the intention is to first bound the stepwise discretization error using the techniques in [1]. We now explain why these techniques cannot be directly applied to decentralized DPMs.
>
> In [1], the authors first state Girsanov’s theorem in Theorem 8 and then use it to analytically express the stepwise discretization KL error in the integral form shown in Eq. (5.5). They further bound the integral length by $\Delta t$ and the integrand by $\Delta t$ in Eq. (5.6). With some extra derivation, [1] can deduce the following KL stepwise discretization error between $p_{s|t}(x_s | x_t)$ and $\hat p(x_s | x_t)$:
> $$ \mathbb E_{p_t} KL(p_{s|t}(x_s|x_t), \hat p(x_s|x_t)) = \mathcal{O}(\Delta t^2). $$
>
> If we follow [1] and attempt to convert the stepwise error to TV distance, we must apply Pinsker’s inequality, which gives:
> $$ \mathbb E_{p_t} TV(p_{s|t}(x_s|x_t), \hat p(x_s|x_t)) = \mathcal{O}(\Delta t). $$
>
> However, this order is insufficient: a stepwise TV discretization error of order $\Delta t$ cannot guarantee global convergence. Summing across all steps yields a global TV discretization error of order $\mathcal{O}(1)$. This bound cannot vanish as $\Delta t \to 0$, making it an unconvergent bound.
>
> The core gap in applying previous SDE-type analyses to decentralized DPMs is that **they inherently yield only $\mathcal{O}(\Delta t)$ stepwise TV error**. This gap persists not only in [1] but also in the SDE-type analyses of [2][3][4][5], because all of these works first bound the stepwise discretization error in KL and then convert it to TV using Pinsker’s inequality. This conversion inevitably makes the TV bound too loose.
>
> This is precisely the gap addressed by our method: instead of relying on KL, we directly use the pseudo-non-Markov technique to analyze the stepwise discretization error in TV distance and derive **a sharper $\mathcal{O}(\Delta t^2)$ TV stepwise error**. Thus, our pseudo-non-Markovian method is necessary to obtain the SOTA convergence result for decentralized DPMs.
>
>
> ### **Even With Additional Tricks, Previous SDE-Type Methods Still Fall Short**
>
> There exist other seemingly plausible attempts to adapt SDE-type methods to decentralized DPM analysis. For instance, one may keep the KL-based stepwise error as $\mathcal{O}(\Delta t^2)$ and accumulate KL across steps to obtain a global KL error of $\mathcal{O}(\Delta t)$, which then implies a global TV error of order $\mathcal{O}(\Delta t^{1/2})$.
>
> However, this approach has the following deficiencies:
>
> - Using KL to disentangle discretization and classifier errors requires the same support of the two distributions. This implicitly imposes an additional assumption that the learned classifier $a_\phi$ must have a strictly positive lower bound. This assumption is unnatural and may not hold in practice.
> - Even if this issue were resolved, the resulting TV convergence rate would be $\mathcal{O}(\Delta t^{1/2})$, which is inferior to the rate we achieve via our pseudo-non-Markov analysis.
>
> [1] Chen, Sitan, et al. *“Sampling is as easy as learning the score: Theory for diffusion models with minimal data assumptions.”* arXiv:2209.11215 (2022).
>
> [2] Holden Lee, et al. Convergence for score-based generative modeling with
> polynomial complexity. NeurIPS,
> 2022.
>
> [3] Holden Lee, et al. Convergence of score-based generative modeling for general
> data distributions. ALT 2023.
>
> [4] Hongrui Chen, et al. Improved analysis of score-based generative modeling:
> User-friendly bounds under minimal smoothness assumptions. ICML, 2023.
>
> [5] Joe Benton, et al. Nearly $d$-linear
> convergence bounds for diffusion models via stochastic localization. ICLR, 2024

---

> ### Author Response · Authors · 2025-12-02
> **Why previous auxiliary-chain–type analysis cannot be a much simpler proof, and our pseudo-non-Markovian method is necessary**
>
> (Continue)
>
> Besides the previous SDE-type methods, the auxiliary-chain–type methods also cannot provide a simple proof of convergence for decentralized DPMs.
>
> ### **Why Auxiliary-Chain–Type Methods Cannot Be Applied to Decentralized DPMs**
>
> In auxiliary-chain–type analyses such as [6], the framework **does not involve bounding the stepwise TV term**:
> $$\mathbb E_{p_t} TV(p_{s|t}(x_s|x_t), \hat p(x_s|x_t))=? $$
> , and thus their methodology cannot support the augment proposed by Reviewer H4tx.
>
> More specifically, auxiliary-chain–type methods derive an error bound between the true state $X_t$ and a constructed intermediate auxiliary state $\bar{Y}_t$. However, the definition of this auxiliary state $\bar{Y}_t$ relies on historical information of the sampling trajectory and therefore cannot be determined in a purely stepwise manner.
> Moreover, how to define such an auxiliary state in the setting of decentralized DPM sampling remains an open and non-trivial problem.
>
> In conclusion, such auxiliary-chain–type methods cannot be used to characterize the pure stepwise TV error, and consequently cannot be directly extended to the convergence analysis of decentralized DPMs.
>
> [6] Gen Li and Yuling Yan. *O(d/t) convergence theory for diffusion probabilistic models under minimal assumptions.* arXiv:2409.18959, 2024.

---

### Official Review · Reviewer_4cTn · 2025-10-31

**Soundness:** 3
**Presentation:** 3
**Contribution:** 3
**Rating:** 6
**Confidence:** 2

**Summary:**

This paper presents a pseudo-non-Markovian analysis framework for studying the convergence of both standard and decentralized diffusion probabilistic models (DPMs). To analyze the discretization error, it proposes to decompose the joint Cartesian space $(x_t,x_s, y)$ into different parts and analyze each part separately. This technique can lead to a linear $d$ dependence in the final results. In combination with the analysis of approximation error, it can be generalized to the setting for decentralized diffusion models.

**Strengths:**

(1) The paper is well written, which presents the proof pipeline very clearly.

(2) The theoretical results are solid, showing superiority over previous works.

(3) The generalization to decentralized diffusion models is very natural.

**Weaknesses:**

(1) The introduction of decentralized DPM is not very clear. It only shows the definition comes from Dong et al.2024, without a brief introduction on why it is defined.

(2) The description of the cluster partition is not very clear. What is the partition data distribution? For example, if each data is drawn I.I.d., the data distribution should be exactly the same as the total distribution (with correct normalization).

(3) Line 385: What is the $L(x_0)$ here?

**Questions:**

I do not check every detail of the proof, and I do not hold questions regarding other parts of the paper.

---

> ### Author Response · Authors · 2025-11-21
> **Reply to Reviewer 4cTn**
>
> Thank you for your appreciation of our work, we will answer your questions one by one regarding these weaknesses/questions. All of our revised parts in the PDF have been outlined in blue color.
>
> > **Answer to Weaknesses 1:** ''The introduction of decentralized DPM is not very clear. It only shows the definition comes from Dong et al.2024, without a brief introduction on why it is defined.''
>
> We agree that a more detailed elaboration on the definition and core motivation of decentralized DPM would help readers better understand its design. We have therefore supplemented clarifications on the definition of decentralized DPM in Sec~2.2.
>
> The key distinction between decentralized DPM and standard DPM lies in its dataset partitioning strategy: it splits the full dataset into $L$ distinct clusters and trains dedicated networks to match the noised score of each cluster.
>
> To formalize this, we first define the training data as partitioned into $L$ clusters: $\{ y_i \in \mathbb{R}^d | i=1, 2, \dots, N \}$ = $\bigcup_{l=1}^{L} \{ y^l_i \in \mathbb{R}^d | i=1, 2, \dots, N_l \}$. The global data distribution is then represented as a collection of cluster-specific distributions: $\\{ p_d^l \\}_ {l=1}^L$. We further define the diffusion reverse process for each cluster $p_d^l$ as $p^l$, its single Gaussian approximation as $\hat{p}^l$, and its network approximation as $\hat{p}_ {\theta}^l$—this definition structure aligns with the $p,\hat{p},\hat{p}_\theta$ notation in standard DPM. The notations $w$ and $u$ also follow analogies from standard DPM.
>
> Moreover, decentralized DPM requires an additional time-aware classification network $a_\phi(x_t,t,l)$ to fuse decentralized score networks during inference. This network inherently models the probability that $x_t$ (at timestep $t$) originates from each cluster, outputting an $L$-dimensional softmax probability vector.
>
> The design of decentralized DPM is driven by multiple distinct motivations:
> - [1] focuses on safeguarding data security across multiple institutions while distributing the training burden across multiple computing nodes in large-scale scenarios.
> - In contrast, [2] and [3] cluster the data into different partitions with the insight that this would mitigate the error, by formulating the reverse transition kernels in a Mixture-of-Gaussian (MoG) form rather than a naive single Gaussian case.
>
> Our paper demonstrates that this decentralized design also ensures inference samples align with the global data distribution.
>
> > **Answer to Weaknesses 2:** ''The description of the cluster partition is not very clear. What is the partition data distribution? For example, if each data is drawn I.I.d., the data distribution should be exactly the same as the total distribution (with correct normalization).''
>
> Theoretically, decentralized DPM does not impose restrictions on data partitioning methods. Our convergence theory remains valid for any data division, as the proof does not rely on specific partitioning strategies.
>
> In practice, common cluster partition approaches include:
> - Adopting natural partitions where different institutions hold their own private datasets (comprising the full-scale dataset collectively).
> - Deriving clusters from the dataset using k-means algorithms.
> - Leveraging inherent class labels for partitioning (e.g., CIFAR-10).
>
> As you mentioned, i.i.d. partitioning would result in each cluster-specific diffusion model converging to the same distribution as the global model—making decentralized DPM a trivial case here. However, when data partitioning is non-i.i.d., our decentralized DPM design still ensures the final samples align with the correct global distribution. This is meaningful for enhancing accuracy [2][3] or preserving data privacy [1].
>
> To improve readability, we have added more description on the discussion on the influence of cluster partition on convergence in methodology part in Section 2.2 and practical cluster partition choice in Appendix~D.3.3.
>
> > **Answer to Weaknesses 3:** ''Line 385: What is the $L(x_0)$ here?''
>
> Throughout the paper, $L(x_0)$ denotes the cluster label of data point $x_0$, represented as an $L$-dimensional one-hot vector. We have added this clarification at the first mention of data clustering (Line~197) to avoid ambiguity. We apologize for the oversight.
>
> Note that we have added a notation table in Appendix A of our revised manuscript to enhance readability.
>
> **References**
>
> [1] David McAllister, Matthew Tancik, Jiaming Song, and Angjoo Kanazawa. Decentralized diffusion models. arXiv 2025.
>
> [2] Yue-Jiang Dong, et al. DC-DPM: A divide-and-conquer approach for diffusion reverse process, 2024.
>
> [3] Hansheng Chen, et al.  Gaussian mixture flow matching models, arXiv 2025.
>
> We sincerely appreciate your insightful comments. Please do not hesitate to let us know if you require any further information or additional explanations.
>
> Sincerely,
>
> The Authors

---

### Author Response · Authors · 2025-12-02
**Reply to All Reviewers, AC, SAC, and PC**

Dear Reviewers, AC, SAC, and PC,

We would like to begin by expressing our sincere gratitude for your engagement throughout the review process, which has significantly improved the quality of our paper.

We are encouraged that the reviewers recognized our Pseudo-Non-Markov Analysis as the **“first convergence guarantee for decentralized DPMs”** (Reviewers H4tx, NH4F, aj1V), acknowledged the importance of clearly discussing **"classifier errors"** in decentralized DPMs (Reviewers H4tx, NH4F), and noted the **"novelty"** of our approach in bypassing SDE theory (Reviewers H4tx, NH4F, aj1V). We are also pleased that our work was regarded as achieving a **“SOTA convergence rate”** (Reviewer H4tx) and offering valuable **"practical insights"** (Reviewers H4tx, NH4F). In addition, we appreciate the recognition of our **“mathematically solid”** derivations (Reviewers 4cTn, aj1V) and the **“well written”** presentation of the paper (Reviewer 4cTn).

In response to the reviewers’ comments, we have revised the following parts of our manuscript:

1. **Clarifying Necessity (To Weakness 1 of Reviewer H4tx):** We provide a clearer explanation of why previous SDE-type methods cannot directly establish convergence results for decentralized DPMs.
2. **Fixing Derivations (To Weakness 2 of Reviewer H4tx and Question 1 of NH4F):** We remove the unnecessary assumption regarding the lower bound of the decentralized classifier and correct a typo in the forward convergence error term.
3. **Notation Clarification (To Weaknesses 2 of Reviewer 4cTn and Weaknesses 4 of Reviewer aj1V):** We add comprehensive explanations of the notations and include a detailed notation look-up table in Appendix A.
4. **Additional Experiments (To Weaknesses 1 of Reviewer NH4F and Weaknesses 6 of Reviewer H4tx):** We conduct new experiments to validate the high-order training algorithms for the decentralized DPM classifier.
5. **Revised Manuscript:** The manuscript has been updated to incorporate these discussions and to correct minor typos.


We promise that we will open source our empirical validation codes, to promote the technological progress of the community. We hope our responses satisfactorily address your questions.

Sincerely,

The Authors

---

### Meta-Review · Area_Chair_pYDv · 2026-01-06

**Summary:**

The paper proposes a "Pseudo-Non-Markov" analysis framework to establish convergence guarantees for Decentralized Diffusion Probabilistic Models (DPMs). The authors argue that existing SDE or auxiliary-chain techniques are inapplicable to the decentralized setting, necessitating this new approach.

While the reviewers acknowledged the timeliness of the problem and the "good" presentation of the paper, the decision to reject is primarily driven by substantial concerns regarding the necessity and technical validity of the proposed framework. Reviewer H4tx raised a critical objection that the authors’ complex analysis might be unnecessary, as standard analysis techniques (utilizing conditional distributions and the chain rule for TV distance) could likely achieve similar results without the proposed "Pseudo-Non-Markov" machinery. Despite an extensive back-and-forth, the authors did not convincingly refute the possibility that a simpler proof exists, casting doubt on the paper's primary theoretical contribution. Additionally, critical assumptions (such as the uniform lower bound on classifier probabilities) were initially presented as necessary but later dismissed as "typos", which reduced confidence in the rigor of the manuscript.

**Reviewer Concerns:**

**Addressed Concerns**

(1) The reviewers (4cTn, aj1V) noted confusion regarding the definition of decentralized DPMs and dense notation. The authors addressed this by adding a notation table and clarifying definitions in the revision.

(2) Reviewer H4tx pointed out an incorrect forward error inequality. The authors acknowledged this as a typo and provided a correction involving KL divergence.

**Outstanding Concerns.**

(1) Necessity of the Analytical Framework (Major): This is the deciding factor. Reviewer H4tx argued that the decentralized setting does not strictly necessitate the new framework and provided a sketch of how existing methods could be applied using conditional mixture distributions. Although the authors argued that standard methods yield loose bounds ($\mathcal{O}(1)$ global TV error), the reviewer remained unconvinced, maintaining that the paper presents a complicated solution to a problem that might be solvable with standard tools.

(2) Assumptions on Classifier Probabilities. Reviewers H4tx and NH4F noted that the assumption of a uniform lower bound on classifier probabilities is unrealistic (as probabilities naturally approach 0 or 1). The authors claimed this assumption was a typo and removed it. However, this "typo" explanation for a significant theoretical assumption is concerning, and the implications of removing it were not fully vetted by the reviewers in the final analysis.

(3) Classifier Error Accumulation. Reviewer NH4F highlighted that classifier approximation error accumulates linearly with timesteps. While the authors proposed "high-order training" as a fix, this was viewed as an afterthought without sufficient empirical backing in the main paper.

**Reviewer Scores:**

Below is my assessment of how the reviewers’ scores might have evolved had the discussion concluded with a consensus on the theoretical dispute:

Reviewer H4tx (Current: 2 ->2/4). This reviewer remained firm in their assessment. Their rebuttal to the authors' response indicates they believe the paper's core contribution (the need for a new framework) is fundamentally flawed or overstated.

Reviewer 4cTn (Current: 6->6/4). This reviewer admitted to not checking math details carefully (Confidence: 2). Given the significant theoretical objections raised by H4tx regarding the necessity of the proof method—which 4cTn praised as "solid" without verifying—it is likely they would lower their score upon realizing the potential redundancy of the proposed method.

Reviewer NH4F (Current: 8->8). This reviewer was very positive about the "first convergence guarantee". However, if the theoretical novelty is successfully challenged, the value of the paper diminishes. They also noted the lack of experiments for the high-order training fix, which remains a validity issue.

Reviewer aj1V (Current: 6->6). This reviewer had concerns about the strength of assumptions and the lack of comparison regarding the tightness of bounds. The dispute over whether the bounds are actually an improvement over standard analysis would likely tip this marginal reviewer toward rejection.

---

### Decision · Program_Chairs · 2026-01-26

Reject